# Nellie: automated organelle segmentation, tracking and hierarchical feature extraction in 2D/3D live-cell microscopy

Austin E. Y. T. Lefebvre [1] ✉, Gabriel Sturm[1,2], Ting-Yu Lin[1], Emily Stoops [1], Magdalena Preciado López[1], Benjamin Kaufmann-Malaga[1] & Kayley Hake [1]

Cellular organelles undergo constant morphological changes and dynamic interactions that are fundamental to cell homeostasis, stress responses and disease progression. Despite their importance, quantifying organelle morphology and motility remains challenging due to their complex architectures, rapid movements and the technical limitations of existing analysis tools. Here we introduce Nellie, an automated and unbiased pipeline for segmentation, tracking and feature extraction of diverse intracellular structures. Nellie adapts to image metadata and employs hierarchical segmentation to resolve sub-organellar regions, while its radius-adaptive pattern matching enables precise motion tracking. Through a user-friendly Napari-based interface, Nellie enables comprehensive organelle analysis without coding expertise. We demonstrate Nellie's versatility by unmixing multiple organelles from single-channel data, quantifying mitochondrial responses to ionomycin via graph autoencoders and characterizing endoplasmic reticulum networks across cell types and time points. This tool addresses a critical need in cell biology by providing accessible, automated analysis of organelle dynamics.

The complex weave and elaborate dance of organelles lies at the center of cellular physiology and pathology. For example, the alterations and balance of mitochondrial dynamics coregulate its turnover, quality control, mitochondrial DNA organization and bioenergetic output[1–5]. Notably, organelles can also form contact sites between one another, allowing for the exchange of metabolites, ions and proteins and promote autophagic turnover, where dysfunctions in any of these have been correlated with aging and other various diseases[6–9].

The topic of organelles as drivers of physiological dysfunction is clearly important; however, the dynamic morphology and motility of these organelles, coupled with limitations inherent to microscopy such as acquisition speed, the diffraction limit and tradeoffs between signal and phototoxicity, pose substantial challenges in extracting this information. This results in manually involved or organelle-specific pipelines that do not generalize well to broader datasets. Consequently,

there is a pressing need for a widely accessible analytical tool capable of providing detailed extraction of spatial and temporal features at multiple organellar scales, but that remains independent of the tool's user or the organelle in question.

Many tools exist for intracellular structural segmentation and tracking[10–29]; however, the pipelines either rely heavily on manual or semi-automated techniques, which are time-consuming, prone to subjective bias and often infeasible for large and/or spatially 3D datasets. Automated methods, which offer improvements in speed and objectivity, frequently struggle with the complexity and variability inherent in biological imaging data. Common issues of existing tools include the inability to effectively handle multiscale structures present between or even within datasets, insufficient segmentation accuracy for dim or small objects and limitations in tracking algorithms, particularly in dense and dynamically complex cellular environments.

[1]Calico Life Sciences LLC, South San Francisco, CA, USA. [2]Department of Biochemistry and Biophysics, University of California San Francisco, San Francisco, CA, USA. ✉e-mail: austin.e.lefebvre+nellie@gmail.com

Additionally, most tools rely on the assumption that an organelle is a single and temporally consistent entity with only occasional and specifically defined merging or splitting events, which limits the quantification of these phenomena to arbitrary metrics. Additionally, deep-learning (DL)-based microscopy methods are rapidly advancing, with state-of-the-art (SOTA) segmentation, tracking and feature extraction tools constantly being released[30]. Most of these tools, however, either lack three-dimensional (3D) functionality (mainly due to the complexity of manual 3D annotation leading to the lack of ground-truth data for the training or fine-tuning of models), organelle generalizability or are specifically tailored to electron microscopy organelle datasets[28,31–45]. Furthermore, these DL models inherently contain 'black box' predictions with unexplainable results that are generally hard to interpret[46]. Thus, there remains a substantial gap in the development of a comprehensive, automated and organelle-agnostic pipeline capable of efficiently and accurately processing large-scale and multidimensional fluorescence microscopy datasets.

In this paper, we introduce Nellie (short for organellometer), a novel, easy-to-use GUI-based, point-and-click image analysis pipeline designed specifically to address these challenges. By incorporating multiscale, structure-enhancing preprocessing methods, Nellie is able to segment and hierarchically divide organelles into logical subcomponents. These subcomponents are interrogated to produce motion-capture (mocap) markers that are compared via local, variable-range feature and pattern matching to create linkages between adjacent frames. These linkages act as beacons for novel temporal interpolation algorithms to provide subvoxel tracking capabilities. We incorporate and introduce a multitude of both standard and advanced quantification techniques to extract a hierarchical pool of descriptive multilevel spatial and temporal features to choose from.

The whole pipeline can be run on a CPU or accelerated on a GPU, either of which computationally outperforms current SOTA organelle segmentation and tracking tools (Supplementary Note 1 and Extended Data Fig. 1). Running the whole pipeline and analysis of its outputs is made easily accessible via a point-and-click Napari GUI[47]. The Napari GUI prompts the user to select their file or folder to analyze (Fig. 1a). A metadata validation module ensures that Nellie has detected proper dimension order and resolutions, and allows the user to correct these parameters if automatic detection has failed (Fig. 1b). If the data contain multiple temporal or channel dimensions, the user can specify which slice(s) to run through Nellie (Fig. 1b). The GUI then prompts the user to run the pipeline and allows for visualization of intermediate images and tracks (Fig. 1c). Finally, the GUI allows for quick visualization of extracted features (Fig. 1d). Notably, Nellie allows the user to run single frame datasets (without a temporal dimension) for morphology-only analysis or multiframe datasets for additional motility quantification, and works for both two-dimensional (2D) (single-plane) datasets and 3D (volumetric) datasets. We also allow Nellie to find compatible plugins via the 'nellie.plugins' Python entry point to allow scientists to integrate their tailored code and pipelines into Nellie's ecosystem.

To showcase the broad range of potential uses that Nellie and its extracted features offer, we present three use cases that we hope will serve to inspire more advanced developments on Nellie's extracted features. First, we show how one can use Nellie's outputs to unmix multiple organelle types from a single channel of a fluorescence time-lapse. Second, using Nellie's outputs we develop a novel multi-mesh approach to organelle graph construction, inspired by DeepMind's recent GraphCast paper[48]. We use this multi-mesh to train an unsupervised graph autoencoder, and use the model to compare mitochondrial networks across a complex feature-space. Third, we demonstrate Nellie's capabilities in characterizing and comparing endoplasmic reticulum (ER) networks between different cell types and temporal frames, showcasing its ability to perform in-depth analyses of organelle morphology, motility and network topology. This case study highlights Nellie's power in extracting meaningful differences between cell types while maintaining consistency within temporal sequences. We hope that this work not only provides a valuable tool for cellular biologists but that it also sets a new standard for automated image analysis as a whole, enabling researchers to gain deeper insights into the complex world of intracellular organization and dynamics.

## Results

### Multiscale adaptive filters enhance structural features

Laser and dye properties can cause the signal-to-noise of organelles to fluctuate widely both between and within datasets. In the preprocessing stage of our pipeline, we account for these fluctuations by implementing a modified version of a multiscale Frangi filter to enhance the inherent structural contrast of organelles, allowing for segmentation based on local structure rather than fluorescence intensity (Fig. 1e and Supplementary Fig. 1)[49]. Our filter is empirically optimized for structures in the size range of typical organelles, and automatically adjusts the filter's effective range based on voxel dimensions to adapt to various magnifications and anisotropies. We modify the traditional Frangi filter to make it generalizable to both tubular and nontubular structures, which we further make more broadly generalizable via an adaptive and fully automated parameter calculation on a scale-by-scale basis to enhance structures of multiple sizes (Supplementary Fig. 2). Our pipeline contrasts with the current SOTA of traditional intracellular segmentation pipelines, which are not adaptive to intrinsic image metadata, and if included, use the same filter parameters across all scales of structural enhancement, limiting the robustness of the filter and subsequent segmentation of variable-scale objects[12,14]. Once filtered, this preprocessed image is subject to standard thresholding techniques for semantic segmentation. Additional details on Nellie's preprocessing pipeline can be found in Supplementary Notes 2 and 3. In this regard, we find that Nellie surpasses current SOTA methods in a diverse range of segmentation tasks across simulated datasets ranging from small round objects to large round objects to small tubular objects to large tubular objects and everything in between across a variety of noise levels (Supplementary Note 4 and Extended Data Fig. 2). Furthermore, we show that Nellie generalizes well to various datasets, including those from different microscopes and across various organelles, as compared to custom pre-trained Swin UNETR DL models (Extended Data Fig. 3 and Supplementary Note 5)[50].

### Hierarchical deconstruction for multilevel segmentation

Before answering how an organelle changes, we must first ask ourselves how an organelle should be represented. Individual organelles are rarely ever individual organelles at all, but rather belong to a complex and continuously evolving organellar landscape. In this regard, it is useful to instead think of and represent organelles as a hierarchical collection of objects at independent frames; the organellar landscape of a cell at a single frame is made up of spatially disconnected organelles, which is in turn made up of numerous subcompartments, which can be broken down into nodes, voxels or even subvoxel regions. To capture this representation, Nellie performs several steps to deconstruct our organellar landscape. First, we employ a Minotri threshold on the preprocessed image generating a semantic segmentation mask, our organellar landscape (Supplementary Note 3). We then perform a simple connected-components-based labeling scheme to generate instance segmentations of individual objects, our spatially disconnected organelles (Fig. 1f). We then skeletonize these segmentations and use the skeleton to identify network junction nodes (branching points) within individual components, allowing us to deconstruct the organelle network into individually labeled branches, our organelle subcompartments (Fig. 1g). We further break down these subcompartments into individual skeleton nodes, which hold properties of their radius-dependent surrounding voxels. To maintain a continuous linkage across all levels of our hierarchy, we can generate adjacency maps by iteratively reassigning the semantic segmentation mask voxels via a

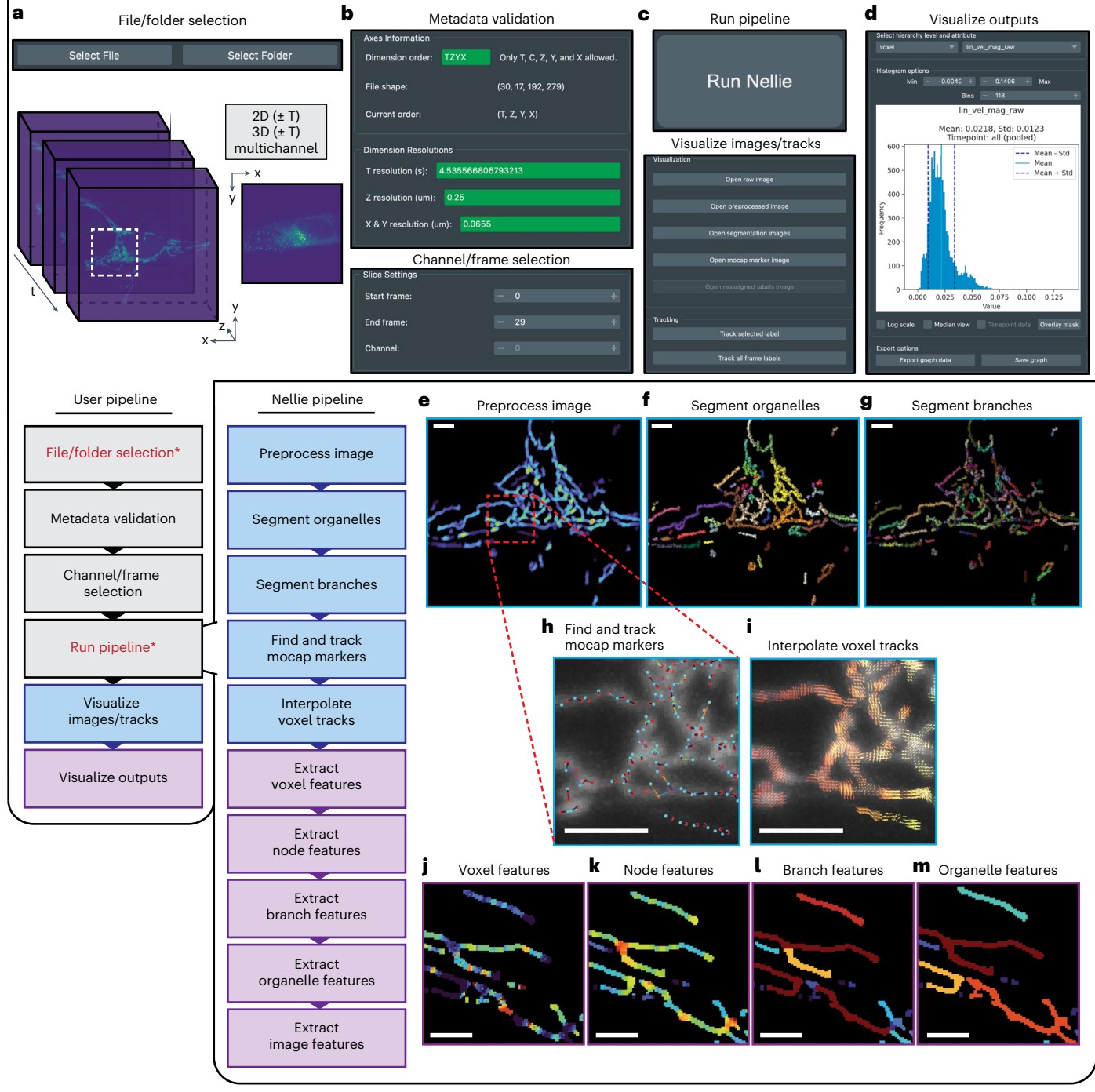

**Fig. 1 | Nellie's user workflow and internal workflow. a–d**, The user pipeline requires (red text, asterisk) the selection and confirmation of data to process and optionally (black) allows for correction of file metadata, data slice selection and visualization of intermediate images and extracted features. The file/folder selection menu, capable of accepting single-plane (2D, *xy*) or volumetric (3D, *xyz*) data with or without multiple time points (*T*) and channels (*Z*) (**a**). The validation menu, which automatically populates dimension order and dimension resolutions based on the file's metadata (top) and allows the user to select a specific channel or a range of temporal frames to run (bottom) (**b**). The processing tab (top) starts Nellie's pipeline, and the visualization tab (bottom) allows the user to concurrently visualize image outputs and tracks (**e–i**) during the pipeline's run (**c**). The analysis tab, which allows the user to export and visualize specific extracted features for the different hierarchical levels and overlay those features on the original image (**j–m**) (**d**). **e–m**, After data confirmation, the Nellie pipeline runs through preprocessing (**e**), segmentation of organelles (**f**) and branches (**g**), mocap marker detection and tracking (**h**), voxel-level track interpolation (**i**), and extraction of features at the hierarchical voxel (**j**), node (**k**), branch (**l**) and organelle (**m**) levels. Scale bars, 5 μm.

*k*-dimensional (*k*-d) tree of the skeleton nodes and their branch labels, enabling the graph-like traversal of our organellar landscape between any hierarchical level[51]. Additional details on Nellie's segmentation pipeline can be found in Supplementary Note 6.

## Motion-capture markers are generated for downstream tracking

The consistency of object-based segmentations and skeleton networks are notoriously temperamental between time points, which causes

linkage problems when using center of mass or skeleton-based tracking approaches. To avoid these problems, we instead generate mocap markers within our organelles, completely independently of our labels, and use these as a basis for linking variable-radius regions of our image across time frames (Fig. 1h). These mocap markers do not intrinsically contain any biological significance, but instead act as guideposts for downstream flow interpolation. This is in line with representing the organellar landscape as a dynamic entity, rather than tracking specific instance segmentations. Additional details on Nellie's mocap marker generation pipeline can be found in Supplementary Notes 7 and 8. Using these mocap markers to generate tracks, we find that Nellie generalizes well to various datasets, including those from different microscopes and across various organelles, and surpasses current SOTA methods in a diverse range of tracking tasks in simulated datasets (Extended Data Figs. 3 and 4 and Supplementary Note 9).

### Features for each motion-capture marker are gathered via variable-range queries
To temporally link these mocap markers, a comprehensive feature vector is constructed to encapsulate critical aspects of the organelles' local characteristics and dynamics. At each mocap marker, the distance-transformed value, representing the organelle's radius at that point, is multiplied by two, providing the dimensions for the bounding box of each marker (Fig. 2a). Within these bounding boxes, the mean and variance are computed for both the raw image and the preprocessed image. These statistics are collectively termed as the 'stats vector' (Fig. 2b). Furthermore, the first six 2D Hu moment invariants of the raw intensity image and the preprocessed image within the mocap marker's bounding box regions are computed to generate translation, scale and rotation-invariant comparison metrics[52]. In 3D, these 2D Hu moment invariants are calculated for $xy$, $xz$ and $yz$ projections of the 3D bounding box region, resulting in what we term as the 'Hu vectors' (Fig. 2c). Finally, to link markers between adjacent frames, a multifaceted cost matrix is constructed (Fig. 2d). Additional details on Nellie's feature-based cost matrix mocap marker-linkage pipeline can be found in Supplementary Note 10.

### Tracking mocap markers for guiding flow interpolation
Rather than solving the linear assignment problem by minimizing the global cost of mocap marker linkages, markers from frame $T$ are simply assigned to their best-matched markers in frame $T + 1$ and vice versa, allowing for 1-to-1 matching, 1-to-$n$ matching and $n$-to-1 matching (Fig. 2e). Each assignment results in a vector pointing from a marker at time $T$ to another marker at $T + 1$, and vice versa, with an associated cost to that linkage. It is important to emphasize that these motion capture (mocap) markers and their linkages do not represent the final organelle tracks but rather serve as the beacons that point in the direction of local motion to inform the subsequent subvoxel flow interpolations that serve as tracks. Once markers are assigned, any arbitrary coordinate or coordinates of interest (CoIs) can have its flow vector interpolated, meaning the motion of an entire organellar object's collection of voxels, or a single branch's collection of voxels, or even a single voxel or subvoxel point within an organelle can be interrogated.

For interpolating flow vectors of CoIs forward in time from frame $t$, a $k$-d tree is first constructed using the coordinates of markers at frame $t$, facilitating efficient nearest-neighbor searches. All nearby marker coordinates within the maximum travel distance from the CoI are then identified, and the distances between these detected markers and the CoI are calculated by querying the $k$-d tree. Each detected marker's flow vector for interpolation of the CoI's flow vector is weighted based on a preference for vectors that are closer and have a lower assignment cost value, indicating a better match during mocap marker linkage. The final interpolated vector from frame $T$ to $T + 1$ is the sum of these weighted vectors (Fig. 2f). A similar process for markers at frame $T − 1$ to frame $T$ is performed for interpolation of flow vectors backward in time. This interpolation process is efficiently executed in parallel for all CoIs, resulting in a list of flow vectors that represent the interpolated motion of the CoIs to the adjacent frame (Figs. 1i and 2g).

Of note, these flow vectors can be used to match voxels between time points, meaning one can determine the fate of each individual voxel coming from an organelle, or any part of its segmentation hierarchy, across all frames of a timelapse (Extended Data Fig. 5 and Supplementary Note 11). Additionally, we note that this novel tracking and interpolation method is not limited to organelles or fluorescence microscopy and can easily be adapted to other types of images, including those segmented via, for example, Meta's Segment Anything Model (Extended Data Fig. 6)[53].

### Features are calculated across multiple hierarchical levels
Though segmentation and tracking of organelles is useful for data exploration and visualization, objective interpretation only becomes possible when quantifiable features are available. To this end, Nellie allows for the calculation and export of a plethora of features specific to each level of the segmentation hierarchy, as well as the statistical investigation of inter-level features, such as an organelle's mean branch feature values, or a branch nodes' mean feature values.

Nellie begins with the calculation of features at the single-voxel level, such as fluorescence and structural intensity values from the raw and preprocessed images, and uses flow interpolation of nearby mocap markers to extract motility metrics at each voxel's coordinates (Figs. 1j and 3a). Next, at the single-node level, which are centered around individual skeleton voxels, Nellie calculates various local morphology and voxel flow patterns (Figs. 1k and 3b). At the single-branch level, Nellie calculates both skeleton-specific and branch-specific morphology features (Figs. 1l and 3c and Supplementary Note 12). Finally, at the single-organelle level, Nellie calculates morphology features of the whole organelle (Figs. 1m and 3d). For all of these metrics, Nellie also outputs a final aggregate dataset, allowing for interpretation of image-wide averages, maximum values, minimum values, variability, etc. of the entire organellar landscape (Fig. 3e). Nellie also calculates statistics of aggregate values for features calculated at lower hierarchical levels. For example, a mean aggregation of linear velocities coming from voxels within a node or organelle can be calculated for each node and organelle as a whole (Fig. 3f). A variety of aggregation metrics can be calculated, from those as convoluted as a branch's nodes' mean linear velocity vector magnitude variability, to those as simple as summing the lengths of all the branches within an organelle (Fig. 3g). The values for each of these level-specific and aggregated features can additionally be overlaid as a colormap for each voxel, node, branch or organelle label, and viewed with the features' corresponding histogram in Nellie's Napari plugin, allowing for easy data exploration and visualization. Additional details on Nellie's hierarchical feature extraction pipeline can be found in Supplementary Note 12, and a full list of exportable features can be found in Supplementary Table 1.

### Case 1 on unmixing multiple organelles in a single channel
In cellular microscopy, imaging more than a few organelle types in live cells within a single time series is a formidable challenge, often constrained by the limited availability of imaging channels, dye or fluorophore specificity, and the necessity to minimize phototoxicity and photobleaching. We introduce an innovative methodology that synergizes the advanced feature extraction capabilities of our pipeline, Nellie, with machine-learning classification techniques. This approach enables the post hoc de-mixing of organelles in single-channel images, effectively addressing a critical bottleneck in cellular imaging.

Utilizing multichannel timelapse fluorescence microscopy, we separately captured images of Golgi apparatus and mitochondria (Fig. 4a). These independent channels were processed through Nellie to extract organelle-specific features. For validation, we generated combined organelle images, comprising both mitochondria and Golgi

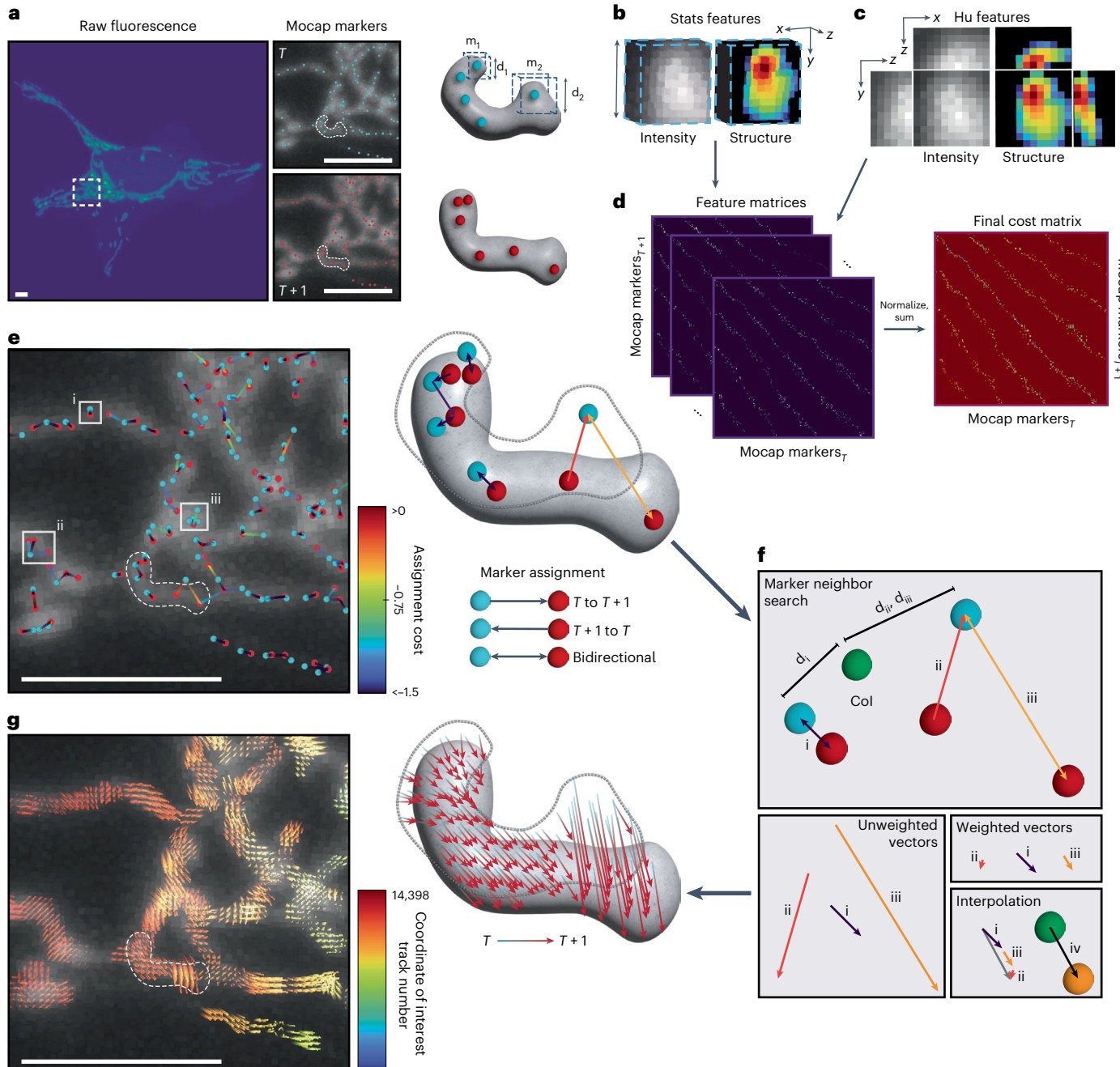

**Fig. 2 | Linking motion-capture markers and interpolating subvoxel movements of organelles. a**, Fluorescently labeled mitochondria (left) and their respective mocap markers for time $T$ (top right, blue dots) and $T+1$ (bottom right, red dots). Feature search bounding boxes are marked by dashed lines for two mocap markers (m1 and m2) with different radii (d1 and d2) at time $T$. **b**, 3D search bounding box raw intensity values (left) and post-structural enhancement values (right) of m2, which are used for calculating stats features. **c**, 2D orthogonal max projections in $xy$, $xz$ and $yz$ of raw intensity values (left) and post-structural enhancement (right) of mocap marker 2, which are used for calculating Hu features. **d**, Difference matrices of distance, Hu features and stats features are calculated between mocap markers in frame $T$ (columns) and $T+1$ (rows) to create weighted feature matrices (left), which are then $z$-score

normalized and summed to create the final cost matrix for marker linkage (right). **e**, Mocap markers from frame $T$ (blue dots) are linked to their best mocap marker match in $T+1$ (red dots) based on assignment cost (line colors) and vice versa for $T+1$ to $T$, resulting in 1-to-1 (i), 1-to-$n$ (ii) or $n$-to-1 (iii) matches. **f**, A CoI (green sphere) not corresponding to a mocap marker (red, blue spheres) has a flow vector interpolated via distance-weighted and cost-weighted vector summation of nearby assigned mocap marker linkages to a new coordinate at $T+1$ (orange sphere). **g**, Interpolated flow vectors for all voxels in the image (left) and flow vector representations (arrows) for all voxels within a mitochondrion (right) between time $T$ (blue) and $T+1$ (red). Images represent a typical U2OS cell, with 11 independent timelapses of fields of individual cells run as an experimental validation. Scale bars, 5 μm.

in a single channel, from the maximum intensity channel-projections of the two channels, and ran this projection image through Nellie to generate instance segmentations and extract their corresponding spatial and temporal features (Fig. 4b). To establish ground truth for

the combined-channel timelapse, we quantified the overlap of mask voxels between the multichannel and the combined-channel masks. Organelles were labeled as mitochondria or Golgi based on the predominant overlapping channel.

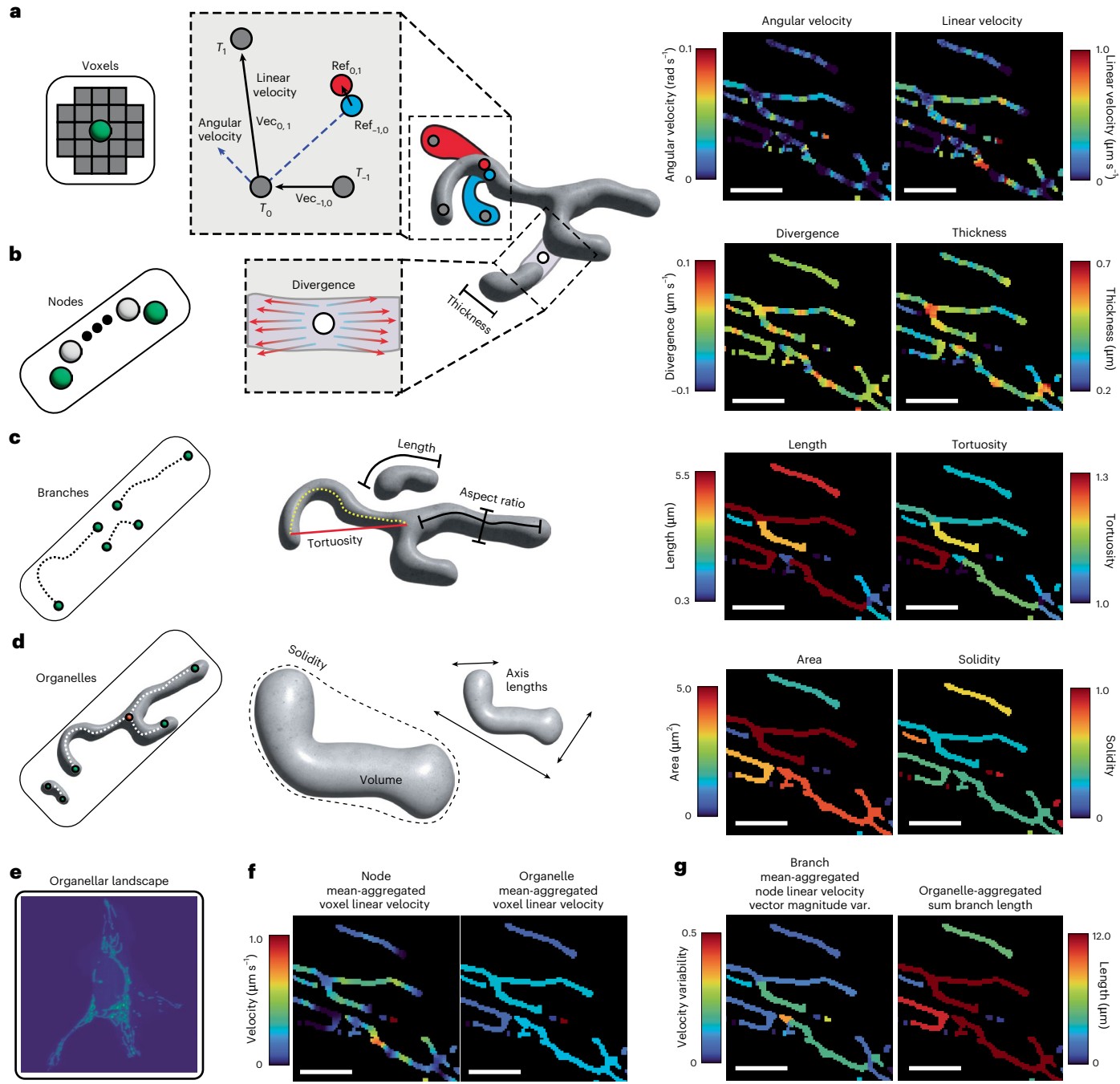

**Fig. 3 | Extraction of spatial and temporal features of organelles at multiple hierarchical levels. a**, Individual voxels represent the lowest hierarchical resolution of our organelle, containing feature information such as voxel intensities and motility metrics. Shown here is a subset of motility features extracted for one tracked voxel (gray dots) through time $T-1$ (blue structure) to $T+1$ (red structure). The tracked reference point for each time point (blue dot, red dot) is shown, and is used to calculate reference-adjusted linear and angular movements. **b**, Individual skeleton nodes represent the next level of the organellar hierarchy, encapsulating voxels within a radius corresponding to that node's border distance, containing features such as divergence/convergence of surrounding flow vectors, thickness and more. **c**, Individual branches represent

the next level of the organellar hierarchy, containing information on the curviness (tortuosity), length, aspect ratio and more. **d**, Organelles represent the next level of the organellar hierarchy, spatially disconnected components in the image, containing information about volume, solidity, axis lengths and more. **e**, The organellar landscape as a whole represents the highest level of our hierarchy, containing aggregate information from all levels below it. **f**, Each hierarchical level can aggregate metrics from its lower level's components, such as a node's or organelle's voxels' mean linear velocity. **g**, Other metrics as complex as a branch's nodes' mean linear velocity vector magnitude variability, or as simple as an organelle's branches' sum branch length can be calculated as well. All images were color-mapped via Nellie's Napari plugin. Scale bars, 5 μm.

Employing the multichannel features from Nellie, we developed three random forest classifier models, each trained on either motility features alone, morphology features alone or a combination of both[54]. The selection of features was grounded both empirically and through the analysis of features showing the most significant fold differences

between mitochondria and Golgi (Fig. 4c). We intentionally exclude voxel intensity metrics in the models as intensity is not inherently dependent on structure. The models' efficacies were tested by comparing the models' predicted organelle types to the ground-truth organelle types. We captured timelapses of 11 cells and used each cell

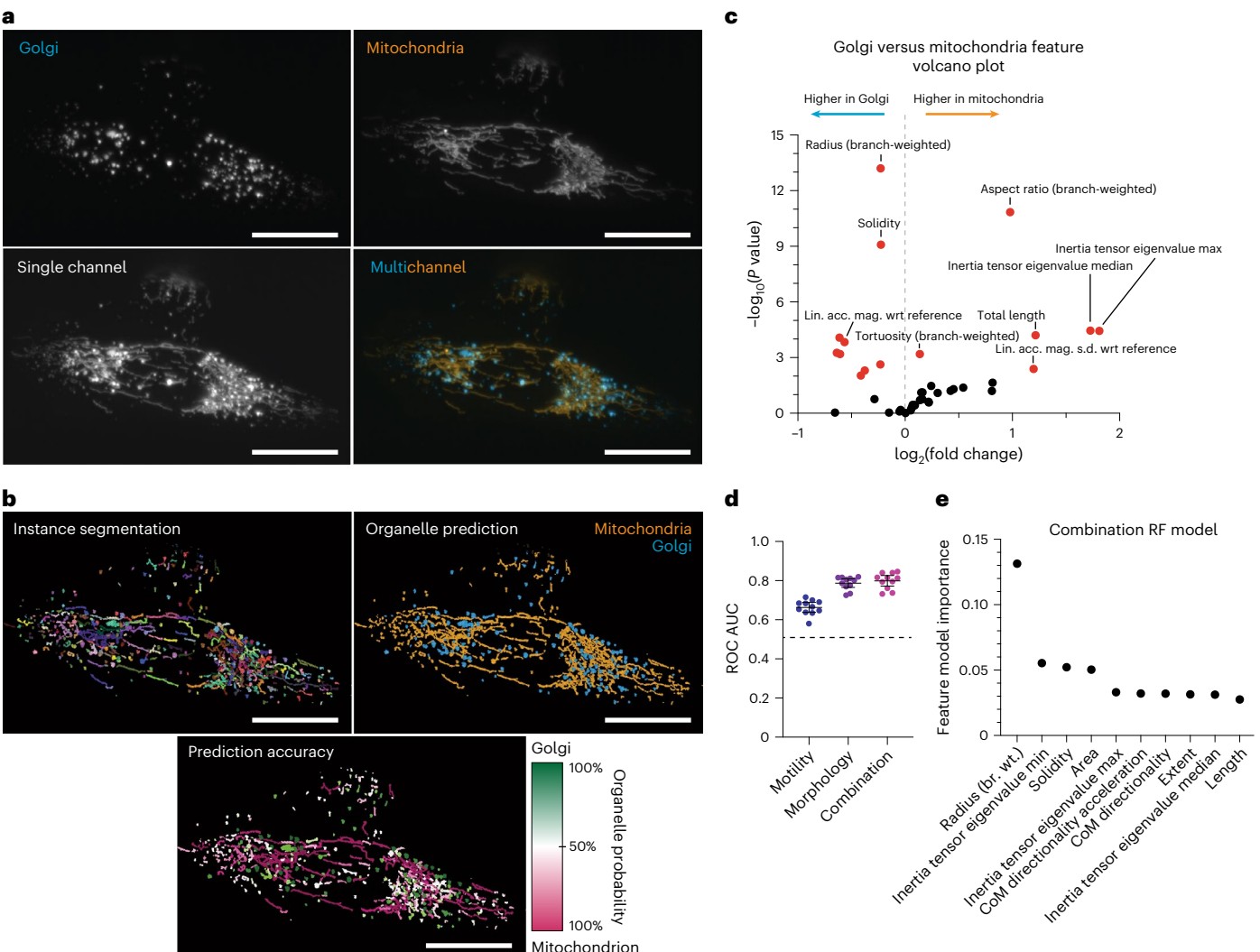

**Fig. 4 | Single-channel multi-organelle unmixing using features extracted by Nellie. a**, Raw intensity images of fluorescently labeled Golgi and mitochondria (top). A single-channel max-intensity projection (bottom left) over the channel dimension, combining the fluorescence signal from both channels (bottom right) into one. **b**, The branch-based instance segmentation output derived from Nellie of the single-channel image (top left). The binary organelle prediction from a trained random forest classifier (top right) and their corresponding probabilities (bottom). **c**, A volcano plot showing features upregulated (positive) or downregulated (negative) in mitochondria compared to Golgi,

with significantly different ($P < 0.05$) features colored in red. Feature $P$ value calculated via unpaired two-tailed $t$-tests. **d**, Areas under the ROC curve for $n = 11$ leave-one-out random forest classifier models with only motility features, only morphology features or a combination of both. Bars are mean ±s.d. **e**, Random forest model feature-importance for the classification of the representative image's organelles. Scale bars, 50 μm. Images are representative and showcase 1 of the 11 independent U2OS timelapses of fields of individual cells run as the experimental sample size. br. wt., branch-weighted.

in an 11-fold leave-one-out cross-validation, leaving one timelapse out of the training set for testing for each cross-validation to evaluate each model, reflecting a realistic experimental scenario with limited sample sizes. Our validation results were promising, with all models surpassing the 0.50 threshold indicative of random guessing in the area under the curve (AUC) of the receiver operating characteristic (ROC) curve. The combined model achieved an average AUC of 0.80, followed by 0.79 for morphology-only features and 0.66 for motility-only features (Fig. 4d). Additional performance metrics also follow similar trends (Supplementary Fig. 7). Moreover, the model allowed us to identify the most impactful features contributing to its performance. These features include a higher aspect ratio and length in mitochondria, owing to its more networked morphology, and a larger radius and solidity in the Golgi, owing to its more spherical shape (Fig. 4e).

This study demonstrates the potential of Nellie in advancing cellular organelle microscopy, especially under conventional imaging constraints. Leveraging standard random forest classification models

from features extracted by Nellie, our method adeptly distinguishes complex biological structures in single-channel images, even with a limited dataset size, creating a useful tool in the field of cellular imaging.

## Case 2 on representation learning of organelle graphs
In cellular microscopy, the intricate task of analyzing organelle networks demands innovative approaches, particularly when examining dynamic alterations in organelle organization. Here, we introduce a novel method that employs graph-based latent space representations to interpret changes in organellar networks. By transforming skeletonized networks of organelle segmentation masks into graph structures and utilizing an attention-based graph autoencoder to transform Nellie's extensive feature outputs into a comparable representation, we decode subtle shifts in organellar arrangements.

We first define the nodes of our graph as skeleton voxels underlying the organelle segmentation masks, with each node encapsulating features of the adjacent organelle voxels. Utilizing the

distance-transformed image, we determine the radius representative of each node. The features of surrounding voxels within this radius and the semantic segmentation mask, including raw intensity, structural enhancement and motility features, are aggregated to form a comprehensive feature set for each node.

Inspired by DeepMind's GraphCast multi-mesh, our method constructs a multilevel graph network to efficiently facilitate message passing at multiple distances within the organelle graph[48]. Intuitively, this graph represents intraorganellar feature dependence, where each node depends, in part, on all other nodes' features within the organelle. The adjacency matrix's construction begins with the selection of a tip node, connecting nodes at increasing powers of two to establish a multilevel mesh (Fig. 5a). A detailed explanation of the multi-mesh creation scheme can be found in Supplementary Note 13.

The graph autoencoder architecture is central to our methodology (Fig. 5b)[55]. We use an initial multilayer perceptron (MLP) to transform the inputs from our original feature set dimension to 512 dimensions, followed by a sigmoid linear unit or Swish activation function, and layer normalization[56]. The encoder uses an independently weighted 16-layer graph neural network (GNN) with a linear MLP, again followed by Swish activation and layer normalization, with a mean aggregation message passing step across the multi-mesh and a final residual connection for each layer[57,58]. The decoder is similarly composed of 16 independently weighted layers, but instead uses a graph attention network operator with a 20% dropout, followed again by a Swish activation function, layer normalization and a residual connection[59]. Finally, we transform the 512 features back to the original feature set's dimensions. A more detailed explanation of the model can be found in Supplementary Note 14.

Our case study focuses on examining mitochondrial networks in cells treated with ionomycin, a calcium channel depolarizer known to induce fission-like events in mitochondria[60]. Before treatment, we capture 20 volumes of fluorescently tagged mitochondria at a frequency of 1 Hz (1 volume per second), leading to 18 pre-treatment graph embeddings (Fig. 5c). We then treat the cells with 4 μM ionomycin and begin imaging for up to 120 volumes, again at 1 Hz. We run the dataset through feature extraction with Nellie, construct its multi-mesh graph and normalize the nodes' features to zero mean and unit s.d. To train the model, we use a 70–30 train–test split of our data and use a mean squared error (MSE) loss to compare the reconstructed features to the original normalized features during training. We run the training with an Adam optimizer and a learning rate of 0.01 until validation loss stops decreasing for more than ten epochs[61]. We use the model with the lowest validation loss as our final model, which was achieved after only 40 epochs.

Post-training, we deployed the model's encoder to obtain latent space representations of each node across different timepoints. These embeddings allowed us to geometrically compare graphs between temporal frames via cosine distances, revealing distinct phases in the mitochondrial network's response to ionomycin treatment. A control graph embedding was established by averaging the 18 temporal frames' pre-treatment graph embeddings, serving as a baseline for comparison. The cosine distance to the control embedding delineated a consistent period of alteration and gradual recovery post-treatment (Fig. 5d). We see that, compared to the control graph, ionomycin shows a quick rise (~60 s to the peak) followed by a recovery nicely modeled by an exponential decay (tau of ~63 s) (Fig. 5e). Using $t$-distributed stochastic neighbor embedding ($t$-SNE) for dimensionality reduction, we visualize the divergence and eventual convergence of post-treatment graph embeddings toward the pre-treatment group, whose 18 points are essentially in the same position on the $t$-SNE plot (Fig. 5f)[62]. Of note, we identify oscillatory patterns in the graph embeddings post-treatment, absent in pre-treatment embeddings (Fig. 5g). These oscillations, discerned via frequency-based bandpass filtering and Fourier analysis, suggest notable mitochondrial responses to ionomycin treatment that are ripe for exploration, but whose analyses and interpretations lie outside the scope of this paper.

This study demonstrates a new method for analyzing organellar organization and motility using graph-based latent space representations. The approach offers rich, exploratory insights into cellular dynamics, akin to Cell-Painting strategies in drug discovery. The potential applications of this technique are vast, ranging from rare event detection in single organelles, to detailed organelle network studies, to broader systemic analyses in various model systems, and can be expanded to graph-representations of other structures at both larger and smaller resolutions. We hope that this case study will inspire further innovative research in organellar microscopy, leveraging the power of graph-based analyses to uncover new dimensions in cellular biology.

## Case 3 on characterization of ER features and network topology

In cellular biology, characterizing and comparing complex organelle networks is crucial for understanding their structure and function; however, this task often presents substantial challenges due to the intricate and dynamic nature of these networks. Here, we demonstrate how Nellie's segmentation and feature extraction capabilities can be leveraged to perform detailed analyses of ER networks, enabling both comprehensive characterization and nuanced comparisons between different cells.

To showcase Nellie's ability to characterize complex organelle networks, we analyzed a 3D lightsheet volume of the full ER network in a primary human fibroblast (hFB) cell (Fig. 6a,b). Using Nellie's segmentation algorithms, we isolated the largest connected network component and constructed a graph representation, with the component's network's skeleton branch junctions serving as nodes and branches as edges between nodes (Fig. 6c). This graph-based approach allowed us to apply concepts from graph theory to investigate the ER network topology. We examined features such as node degree (the number of other nodes connected to each node) and node betweenness centrality, which is how frequently a node is traversed when traveling between two nodes via their shortest path (Fig. 6a,d,e). We found that the distribution of node degrees in our ER network closely aligns with existing data from ER-specific quantification tools such as ERnet, with a majority of nodes (56.83%) having a degree of 3 (Fig. 6f)[33].

Beyond characterization, researchers often need to compare organelle networks between different conditions, such as wild-type versus treated cells, cells derived from different patients and distinct cell lines. To demonstrate Nellie's comparative capabilities, we analyzed three sequential frames from both an hFB cell and a U2OS cell line. We compared ER network graph topologies between the two cell types and across temporal frames. As expected, we found minimal variability in average node degree, betweenness centrality and normalized cyclomatic number between the three frames within each cell. Notably, however, significant differences were observed between the hFB and U2OS cell lines for these metrics (Fig. 6g–i). This suggests that while the ER network topology remains relatively stable over short time periods within a cell, there are distinct differences in ER organization between these two cells.

Nellie's branch-segmentation capabilities allowed us to compare ER branch morphology and motility between cells and across frames. We examined features such as branch length and linear velocities. Similar to the topology analysis, we observed little variability in these features between frames of the same cell line, but significant differences between the hFB and U2OS cells (Fig. 6k,l).

To perform a more comprehensive comparison, we employed tensor decomposition techniques. This approach allowed us to extract weights of different features, cells and temporal frames for the first component of the decomposition (Fig. 6m–o). The results provide a holistic view of the most influential factors distinguishing the ER networks between cell types and across time.

To further validate the distinctiveness of ER characteristics between cells and the consistency within temporal frames, we built random forest

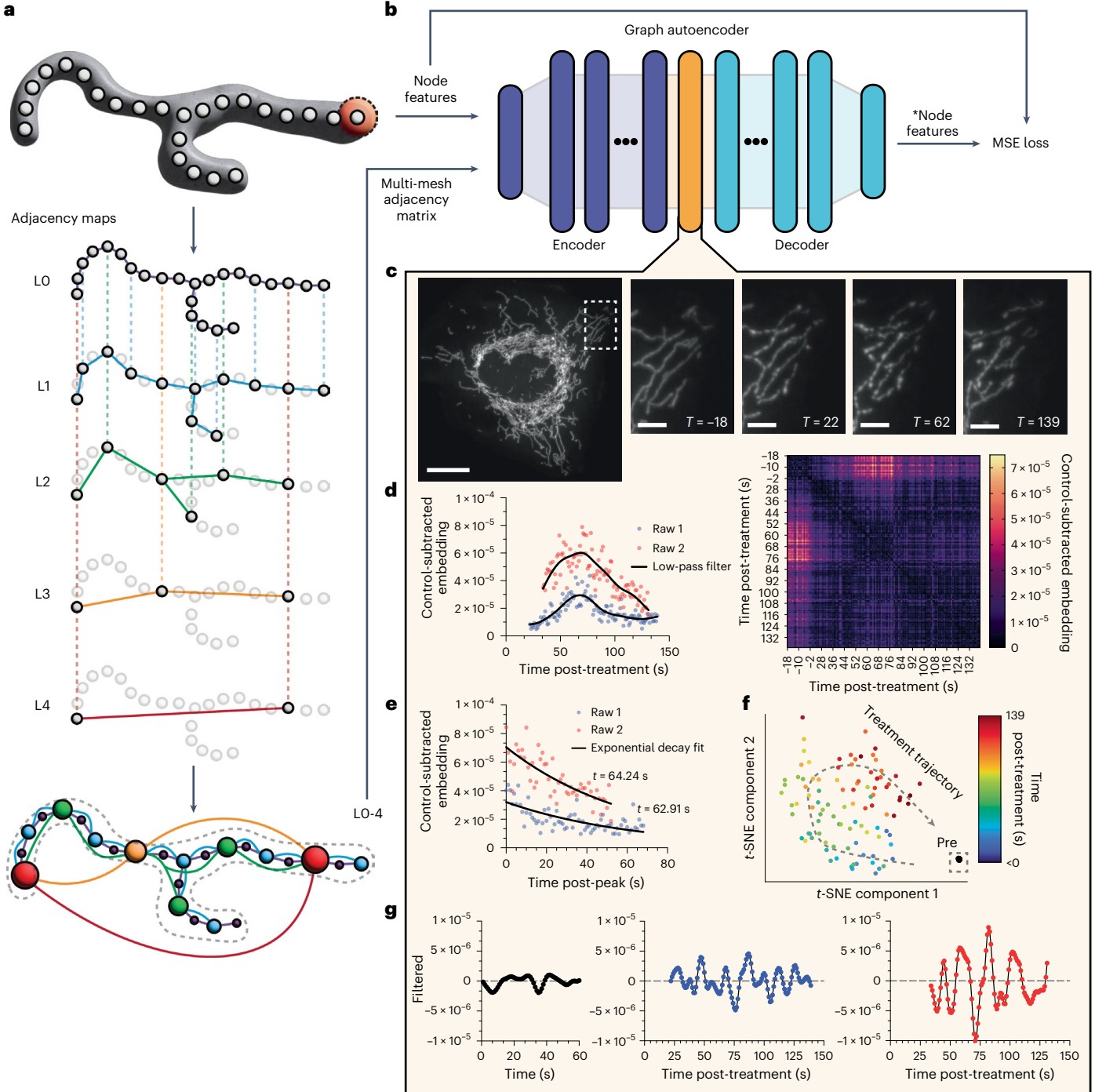

**Fig. 5 | Quantification of the evolution of ionomycin-treated mitochondria multi-mesh graphs via comparisons of graph autoencoder latent space embeddings. a**, Example construction of a multi-mesh adjacency network of a mitochondrion, where each level (L) corresponds to the power-of-2 node-jump distance used for node linkage at that level, allowing for efficient message passing at variable ranges within one organelle. **b**, The multi-mesh adjacency matrix (node edges) and node features are used as the inputs to a graph autoencoder. The GNN uses an MSE loss derived from the comparisons on the differences of the input and output node features for unsupervised training. After training, the latent space representation outputs (orange) from the encoder can be used for vector-based similarity comparisons. **c**, Fluorescently labeled mitochondria with representative images throughout their responses to ionomycin treatment. **d**, Cosine distances of latent space embeddings to the

average of the 18 pre-treatment latent space embeddings for two independent samples (left), or to all latent space embeddings (right) of mitochondria multi-mesh graphs after ionomycin treatment. Raw 1 corresponds to the representative images in **c**. **e**, Cosine distances of latent space embeddings to the average of the 18 pre-treatment latent space embeddings post-peak, fit to exponential decay curves. **f**, t-SNE dimensionality reduction of latent space embeddings colored by time post-treatment. **g**, Bandpass filtered cosine distances of latent space embeddings of untreated mitochondria to the first frame's mitochondrial latent space embeddings (left) or of ionomycin-treated mitochondria to the first pre-treatment frame's mitochondrial latent space embeddings (middle and right). Scale bars, 50 μm for the full-field image and 10 μm in length for the zoomed-in images. Images represent a typical U2OS cell, with two independent timelapses of fields of individual cells run as the experimental sample size.

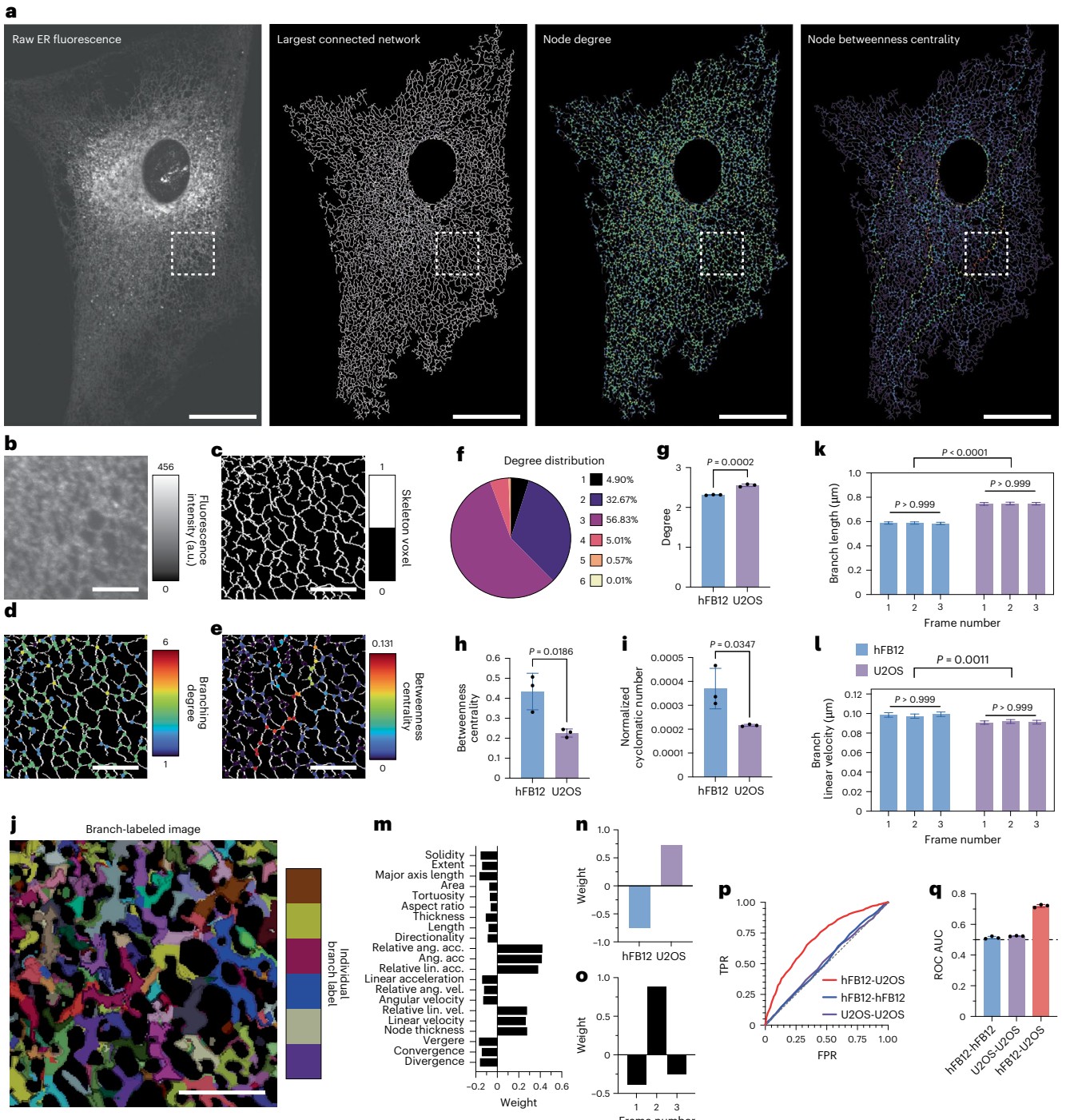

**Fig. 6 | Nellie allows for in-depth characterization and comparisons of ER morphology, motility and network topology. a**, Raw fluorescence of a 3D ER network from a primary human fibroblast cell (hFB12) labeled with CellLight ER. Shown also is the ER network's largest connected component skeleton and its corresponding connectivity graph, with nodes color coded based on node degree and node betweenness centrality. **b–e**, Also shown are close-ups of the regions denoted by the white bounding boxes in the raw (**b**), skeleton (**c**), degree color coded (**d**) and betweenness centrality color coded (**e**) images. **f**, The distribution of node degrees in the ER network from **a**. **g–i**, The mean degree (**g**), betweenness centrality (**h**) and normalized cyclomatic number (**i**) in three sequential 2D temporal frames of either a hFB12 or U2OS cell. Bars are mean ± s.d. *P* values were calculated via an unpaired two-tailed *t*-test. *n* = 3 temporal frames. **j**, The instance segmentation of individual ER branches in the same close-up region as **b–e**, randomly color coded by label number. **k,l**, Branch lengths (**k**) and linear velocities (**l**) of three sequential 2D temporal frames of either a hFB12 (blue) or U2OS (purple) cell. Bars are mean ± 95% CI.

Intra-cell *P* value calculated via a Kruskal–Wallis test with a corrected Dunn's multiple comparisons test. *n* = 6,109, 6,242 and 6,229 branches for hFB12 frames 1–3, respectively, and *n* = 6,319, 6,507 and 6,526 branches for U2OS frames 1–3 respectively. Inter-cell *P* values were calculated via an unpaired two-tailed *t*-test. *n* = 18,590 and 19,352 hFB12 and U2OS branches, respectively. **m–o**, Tensor decomposition weights of the first component for branch features (**m**), cell types (**n**) and temporal frame number (**o**). **p,q**, ROC (**p**) and corresponding AUC of the ROC (**q**) for random forest models to classify between ER branches between hFB12 and U2OS cells (red), two separate temporal frames of hFB12 cells (blue) or of U2OS cells (purple). *n* = 3 random forest classifiers for each group. Bars are mean ± s.d. Scale bars, 50 μm for the full-field images and 10 μm for the zoomed-in images. Images represent a typical hFB12 cell, with ten independent timelapses of cells run as a validation sample size. Relative ang. acc., relative angular acceleration; ang. acc, angular acceleration; relative lin. acc., relative linear acceleration; relative ang. vel., relative angular velocity; relative lin. vel., relative linear velocity.

classifiers to predict the origin of a branch object. When classifying branches between hFB and U2OS cells, the model achieved high accuracy with an ROC AUC of 0.72 (Fig. 6p,q). In contrast, when attempting to classify branches between two separate temporal frames from the same cell, the model performed essentially no better than random chance, with an AUC of 0.51 and 0.52 for hFB and U2OS cells, respectively.

These classification results further underscore the intrinsic differences in ER branch features between the two cells, while confirming the consistency of Nellie's outputs between temporal frames of the same cell. This consistency is crucial, as we would expect similarity between close temporal frames of the same cell under stable conditions.

In conclusion, this case study demonstrates Nellie's powerful capabilities in characterizing and comparing complex organelle networks. By leveraging advanced segmentation, feature extraction and analytical techniques, Nellie enables researchers to gain deep insights into organelle structure and dynamics, even for complex endomembrane organelles such as the ER. The ability to perform robust comparisons between different cell types or conditions opens up new avenues for understanding the role of organelle organization in cellular function and disease states.

## Discussion

Nellie is an unbiased and efficient pipeline for accurately and automatically analyzing spatial and temporal features of organelles, while staying agnostic to the organelle or substructure in question. We use intracellularly optimized adaptive preprocessing to allow for multiscale segmentation and adaptive local peak detection to generate multiscale markers for tracking. Our mocap marker-linkage method uses the most important regions in our data as waypoints, independent of instance segmentations, while our flow interpolation method allows for tracking of subvoxel coordinates. Finally, our multilevel feature extraction allows cross-sectional and hierarchical analyses both within and between scales of interest. This allows users to interpret their data from a subvoxel level, all the way up to an image-wide level. We fully automate parameter selection for every step of the pipeline in a way that adapts to the image context to ensure no tailoring of features is required by the user. Conveniently, we package our methods in a Napari plugin GUI to allow for visualization and point-and-click functionality for ease of use, while keeping the codebase modular to allow flexibility and extensibility for more advanced users.

The case studies we presented demonstrate Nellie's broad range of applications across multiple aspects of cellular biology. First, we demonstrated Nellie's ability to capture extensive metrics in both Golgi and mitochondria by using object and branch-based morphology and motility features to train random forest models and predict organelle type.

Second, we constructed multilevel mesh-like graph networks of mitochondria, which we used to train an unsupervised graph autoencoder. We used this to compare latent space embeddings of the graph nodes to investigate the effect of ionomycin on mitochondrial networks, such as intrinsic feature oscillations and post-treatment effect-and-recovery dynamics. To our knowledge, this case study also establishes the first organelle-based graph autoencoder, a model that would not be possible without high quality segmentation and tracking, and a diverse and comparable feature set, both of which were only possible with Nellie. We predict this organelle graph autoencoder type of model to have a broad and useful range of applications, from treatment-based clustering and comparisons of organelles, reminiscent of Cell-Painting methods, to local morphology and motility predictions of organelles, reminiscent of GraphCast's weather prediction, but at intracellular scales[48,63].

Third, we used Nellie to characterize and compare ER networks across different cell types and temporal frames. By leveraging Nellie's segmentation and skeletonization techniques, we performed in-depth analyses of ER morphology, motility and network topology.

While Nellie performs well across diverse datasets without parameter tuning, it may underperform compared to specialized software when analyzing specific organelles. For example, dedicated tools such as ERnet and AnalyzER for the ER discriminate between tubular and sheet-like structures[29,33]. Similarly, MitoSegNet specializes in mitochondrial analysis in 2D images, and the Allen Institute's segmentation pipeline offers specialized tools for each organelle[28,32]. We attempted to compare Nellie with these models but faced challenges: ERnet had installation issues and is limited to 2D images; MitoSegNet failed to detect structures in our images and also operates only on 2D images; and the AICS Segmenter, while providing 2D and 3D models, requires extensive manual fine-tuning, making it impractical for comparison.

While such tools may offer superior performance for their target organelles, Nellie's strength lies in its generalizability across organelles, without any parameter tuning or annotated datasets; however, Nellie does not perform optimally for organelles with large minor axes, such as nuclei, as it tends to segment the edges rather than the entire structure. This limitation suggests that for certain applications, integration with specialized methods may be necessary. As DL advances, generalized foundation models trained on extensive ground-truth data may eventually surpass Nellie, but until then, Nellie remains a valuable tool for researchers needing a flexible, organelle-agnostic pipeline for spatial and temporal analysis. In the future, we plan to build out more plugins to extend Nellie's outputs toward specific organelles. We also plan to incorporate multichannel analysis tools to natively analyze multi-organelle interactions.

We hope Nellie will promote imaging-based approaches for analyzing organelles and their perturbation-mediated disruptions. Additionally, we hope Nellie's ease of use will encourage open-access science by providing a simple way to share intracellular feature data in a comparable manner. We present Nellie as a catalyst for a new wave of scientific inquiry, where the complex weave and elaborate dance of organelles is not just observed, but deeply understood, and where the mysteries of the cell are unlocked, one pixel at a time.

## Online content

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

## Methods

### Cell culture

U2OS osteosarcoma cells used in the representative figures throughout the main text, and in the multi-mesh GNN case study were cultured following standard procedures at 37 °C with 5% $CO_2$ in DMEM (Thermo, 10567014) supplemented with 10% fetal bovine serum (Gibco, 26140-079). Cell passaging occurred every 3–5 days and downstream assays were performed before reaching 20 passages of growth. For the organelle unmixing case study, U2OS cells were cultured in DMEM supplemented with 10% FBS, 1× antibiotic-antimycotic solution and 1× GlutaMAX (Gibco).

Primary human fibroblasts (male, LifeLine Cell Technologies, FC-0024, 03099) were cultured in DMEM with 5.5 mM glucose (Thermo Fisher, 10567022) supplemented with 10% FBS. Fibroblasts were split every 7 days and all experiments were performed between 3–5 passages after thawing.

### Fluorescent labels

To establish a stable cell line expressing a fluorescently labeled version of COX8A, a mitochondrial matrix protein, for representative images throughout the main text and the multi-mesh GNN case study, U2OS cells at passage no. 8 (1 million cells) were transfected with 1 µg DNA (Davidson-COX8A-mEmerald construct, Addgene, 54160) using the SE Cell Line Nucleofector kit (Lonza, V4XC-1032) following the manufacturer's protocols. Transfected cells were cultured on collagen-coated plates to facilitate the recovery process. After 2 days, cells were selected with 1 mg ml$^{-1}$ Geneticin Selective Antibiotic (G418 Sulfate, Thermo Fisher, 10131035) for 1 week. Fluorescence signals were continuously monitored during the selection process. Upon recovery, G418 at a concentration of 0.5 mg ml$^{-1}$ was used for stable cell line maintenance. FACS-sorting for moderate-expression cells was performed to enhance the homogeneity of cells containing labeled mitochondria. Lysosome staining was performed via a 30 min incubation of 1 µM SiR-Lysosome (Cytoskeleton, CY-SC012) and washed out 2× with warm medium.

To generate U2OS cells expressing three genetically encoded fluorescent markers targeted to organelles for the organelle unmixing case study, a plasmid was generated containing mEGFP targeted to the mitochondrial matrix with a human COX8 presequence, ECFP-tagged H2B (not used in this paper), and the first 82 residues of B4GALT1 tagged with mScarlet. MluI and AsiSI sites were used to insert left and right homology arms, respectively, for the CLYBL safe harbor locus flanking the coding sequence. Two homozygous knock-in clones were generated in U2OS cells (ATCC, HTB96) using CRISPR editing. Clones were validated using Sanger sequencing of genomic DNA.

To fluorescently label the ER, fibroblasts and U2OS cells were plated in eight-well glass-bottom chamber slides (CellVis, C8-1.5H-N), pre-coated with 1:100 dilution fibronectin (Sigma, F1141). After 4 h of plating cells were transfected overnight with 10 ppc of CellLight ER-RFP, BacMam v.2.0 (Thermo Fisher, C10591), as per the manufacturer's protocol. No other transfection reagent was used due to the active viral capacity of the CellLight formula. Fresh medium was added to cells an hour before imaging.

### Image acquisition

Cells were plated on fibronectin-coated (Sigma, F1141-1mg) eight-well glass-bottom chamber slides (CellVis, C8-1.5H-N) and incubated for 4 h (main text figures and multi-mesh GNN case study) or 24 h (organelle unmixing case study and ER network imaging). Imaging was then performed on an in-house single objective light sheet microscope with a stage-top incubator maintaining a temperature of 37 °C with 5% $CO_2$ throughout[64]. Videos were acquired of individual cells at a frequency of one 3D volume per second using 488 nm laser at 5% power and 1-ms exposure, calculated to be 20.67 µJ per volume on the sample.

Imaging of the ER was performed on a Nikon W1 spinning-disk confocal system with a Plan Apo VC ×100/1.4 oil objective, using a 561 nm excitation laser line and zET405/488/561/635 m quad filter, and a stage-top incubator set at 37 C with 5% $CO_2$. Full 3D z-stacks were acquired with 250 nm slices for a total of 5.25 µm depth across the cell. Imaging 2D time series videos were collected at 1.6 frames s$^{-1}$.

### Treatments

For calcium-ionophore treatment experiments, ionomycin (4 µM, Thermo, I24222) in dimethylsulfoxide (DMSO) was manually injected into the medium of the cell imaging dish.

### Random forest classifier

For our random forest classifier in Case Study 1, we used the same hyperparameters across all three models (motility only, morphology only and motility and morphology combined) to discriminate against mitochondria and Golgi within our organelle de-mixing case study. The random forest classifier was implemented via scikit-learn's RandomForestClassifier class, with the following hyperparameters: n_estimators = 300; criterion = 'gini'; max_depth = None; min_samples_split = 2; min_samples_leaf = 1; min_weight_fraction_leaf = 0.0; max_features = 'sqrt'; max_leaf_nodes = None; min_impurity_decrease = 0.0; bootstrap = True; oob_score = False; n_jobs = −1; random_state = 42; verbose = 0; warm_start = False; class_weight = None; ccp_alpha = 0.0; max_samples = None; and monotonic_cst = None.

For the motility-only model, the median, maximum, minimum and s.d. values of the following features were used as inputs: the angular velocity magnitude with regard to the branch's pivot point between t1 and t2 (rel_ang_vel_mag_12), the angular acceleration magnitude with regard to the branch's pivot point (rel_ang_acc_mag), the linear velocity magnitude with regard to the branch's pivot point between t1 and t2 (rel_lin_vel_mag_12), the linear acceleration magnitude with regard to the branch's pivot point (rel_lin_acc_mag), the linear velocity magnitude of the branch's pivot point between t1 and t2 (ref_lin_vel_mag_12), the linear acceleration magnitude of the branch's pivot point (ref_lin_acc_mag), the directionality of the branch with regard to the center of mass of the fluorescence intensity between t1 and t2 (com_directionality_12) and the rate of change of the directionality with regard to the center of mass of the fluorescence intensity (com_directionality_acceleration).

For the morphology-only model, the following features were used as inputs: area; extent; solidity; the minimum median and maximum of the object's inertia tensor eigenvalues (inertia_tensor_eig_sorted_min; inertia_tensor_eig_sorted_mid; and intertia_tensor_eig_sorted_max); the total length (branch_lengths); the average radius (branch_radius); the average tortuosity (branch_tortuosity); and the average aspect ratio (branch_aspect_ratio).

For the morphology and motility combined model, all the features from both the morphology and motility models were aggregated.

For both training and testing, all feature columns were first standardized (mean of 0, s.d. of 1), then used as inputs to the random forest classifier.

### Reporting summary

Further information on research design is available in the Nature Portfolio Reporting Summary linked to this article.

## Data availability

All data supporting the findings of this study are available in the article and its supplementary information files. Raw .tif files are available upon request only, due to the large size and number of .tif files present throughout the paper. Example images are provided within GitHub for testing at https://github.com/aelefebv/nellie/tree/main/sample_data. Source data are provided with this paper.

## Code availability

The Nellie pipeline and its Napari-based plugin is fully written in Python. The Python code and plugin are freely available online via

GitHub at https://github.com/aelefebv/nellie. Supplemental materials for non-pipeline-related code can be found via GitHub at https://github.com/aelefebv/nellie-supplemental. A template for creating Nellie plugins can be found via GitHub at https://github.com/aelefebv/nellie-plugin-example.

## References

64. Millett-Sikking, A. & York, A. High NA single-objective light-sheet. *Zenodo* https://doi.org/10.5281/zenodo.3376243 (2019).

## Acknowledgements

We thank A. G. York for helpful discussions on the pipeline, A. Millett-Sikking for advice and guidance on microscopy with the SOLS and E. Salinas for insights on fluid dynamics and its applications to this pipeline. We also thank B. Feng, M. Onsum and C. Ledogar for reviewing the paper content and M. Kane and C. Sanford for reviewing the paper for its wording and grammar. We also thank Calico Life Sciences for supporting this work.

## Author contributions

A.E.Y.T.L. conceived of and designed Nellie and its pipeline, wrote the paper, created the figures, developed and wrote the code, performed the data analysis and supervised the study. G.S. and K.H. provided experimental design ideas. A.E.Y.T.L. and G.S. α-tested the code, and performed cell-based maintenance and experiments, as well as microscopy experiments. T.-Y.L., E.S and M.P.L. provided cells and reagents. A.E.Y.T.L., G.S., K.H. and B.K.-M. provided helpful feedback on experimental design, data interpretation and edited the paper. A.E.Y.T.L. and G.S. performed experiments for the revision of the paper.

## Competing interests

The authors declare no competing interests.

## Additional information

**Extended data** is available for this paper at https://doi.org/10.1038/s41592-025-02612-7.

**Correspondence and requests for materials** should be addressed to Austin E. Y. T. Lefebvre.

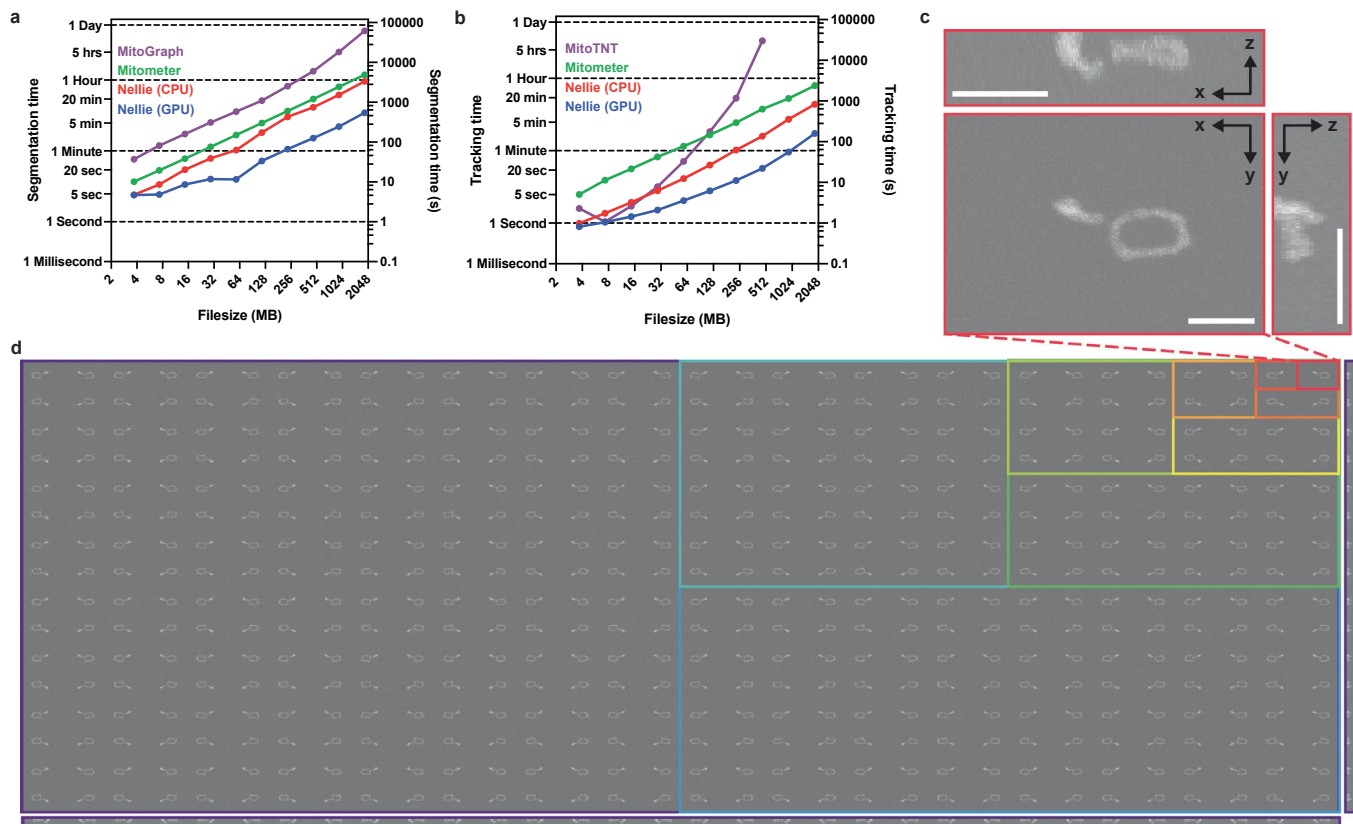

**Extended Data Fig. 1 | Runtime comparisons. a**, Segmentation time comparisons for various tools with datasets of increasing size: MitoGraph (purple), Mitometer (green), Nellie running on CPU (red), and Nellie running on GPU (blue). **b**, Tracking time comparisons for different tools: MitoTNT (purple), Mitometer (green), Nellie on CPU (red), and Nellie on GPU (blue), across datasets of increasing size. **c**, Visualization of the first of two timepoints from the smallest dataset (3.7 MB). **d**, Visualization of the first of two timepoints from the largest dataset (1.87 GB), with intermediate dataset sizes marked by different colored borders. Scale bars are 5 um in length.

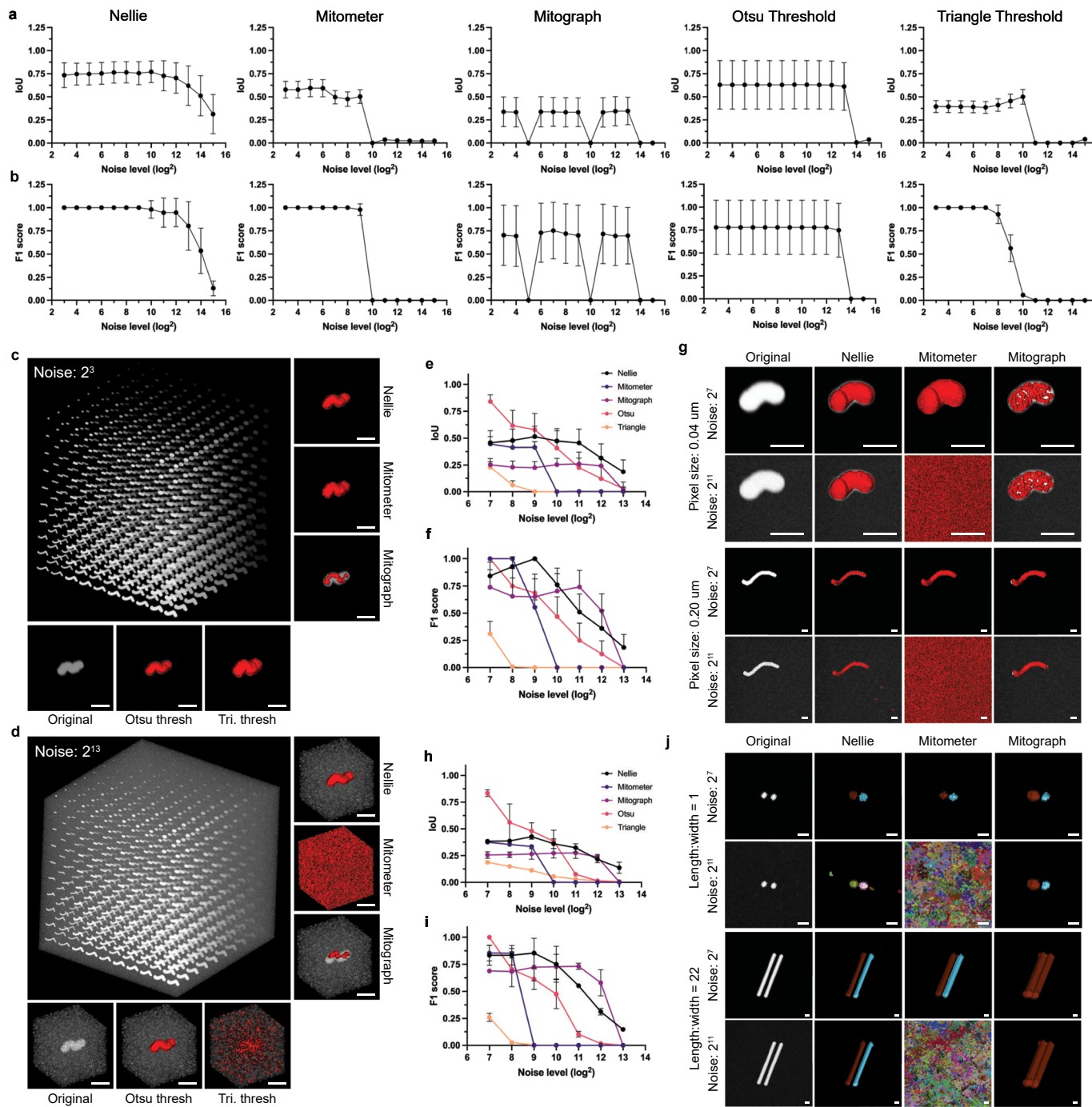

**Extended Data Fig. 2 | Segmentation comparison and benchmarking against state-of-the-art organelle segmentation algorithms.** Quantification of segmentation quality of Nellie, Mitometer, Mitograph, a simple Otsu threshold, and a simple triangle threshold from intersection over the union (IoU) (**a**), and F1 scores (**b**) via comparisons of the algorithms' outputs against generated ground truth data. N = 10, where each point is the mean score at a different intensity value. Generated ground truth consists of 1000 simulated organelle objects at low (**c**) to high (**d**) noise levels. Images of a single object's segmentation are shown for each method. **e**, **f**, IoU and F1 scores at increasing noise levels for objects of various lengths and thicknesses generated at varying pixel resolutions. N = 32, where each point is the mean score at a different pixel resolution.

**g**, Representative images of generated objects at low and high noise levels at high and low pixel resolutions. **h**, **i**, IoU and F1 scores at increasing noise levels for objects of various lengths and thicknesses generated at varying pixel resolutions at a separation distance of at least 2 times the objects' thicknesses. N = 3, where each point is the mean score at a different distance. **j**, Representative images of generated objects at low and high noise levels at low and high length:width ratios. All data points are mean ±95% confidence intervals. Scale bars are 2 um in length.

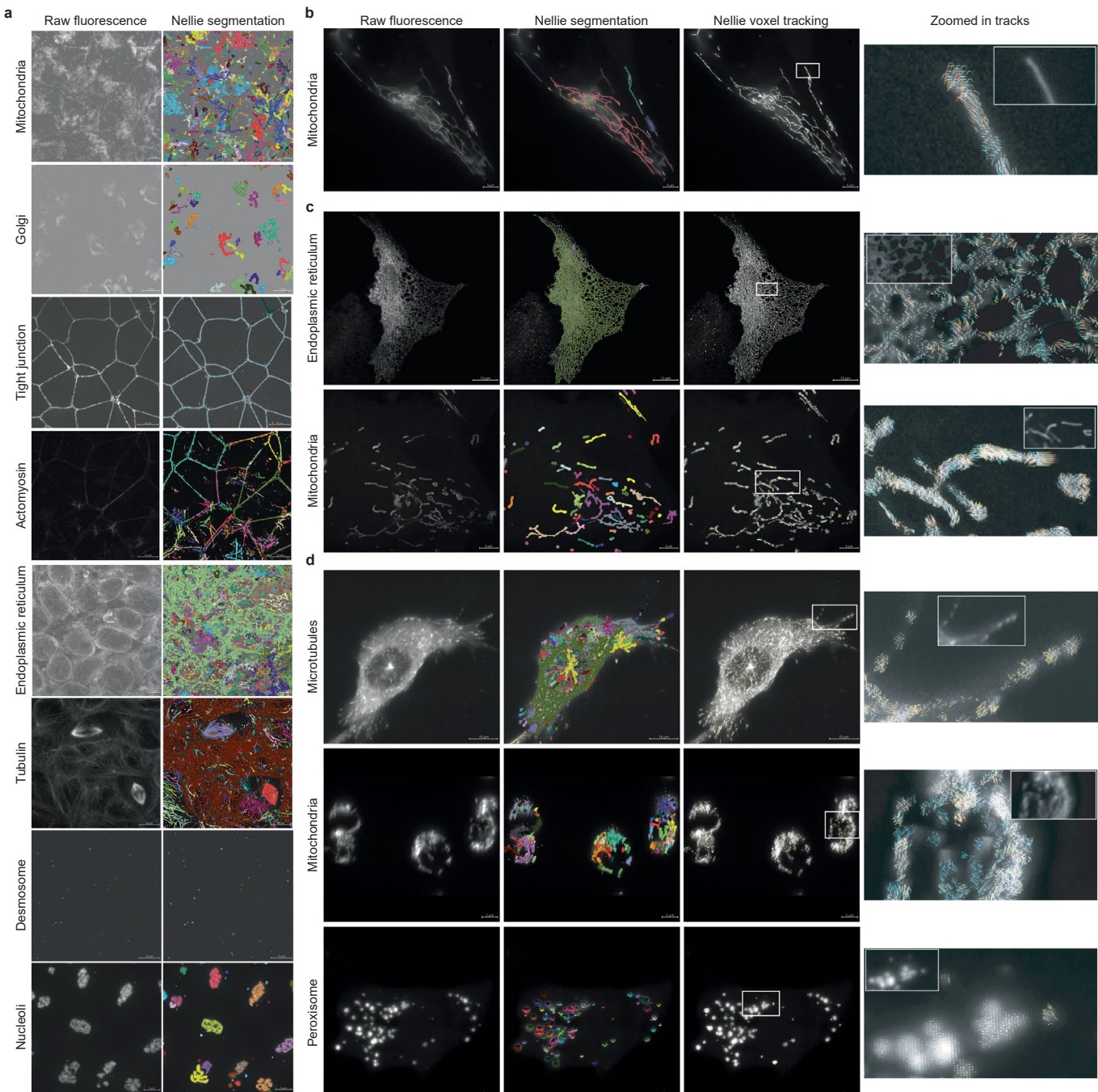

**Extended Data Fig. 3 | Segmentation and tracking showcase across organelles and imaging modalities. a**, Examples of Nellie's segmentation algorithm outputs on the Allen Institute of Cell Science's 3D single-frame confocal microscopy datasets of various fluorescently labeled structures including their raw fluorescence intensity images on the left. **b**, Example of Nellie's segmentation and tracking algorithm outputs on a 2D timelapse widefield dataset of fluorescently labeled mitochondria, including its raw fluorescence intensity image on the left. **c**, Examples of Nellie's segmentation and tracking algorithm outputs on 2D timelapse confocal spinning disk microscopy datasets of fluorescently labeled endoplasmic reticulum and mitochondria, including their raw fluorescence intensity images on the left. **d**, Examples of Nellie's segmentation and tracking algorithm outputs on 3D timelapse lightsheet microscopy datasets of fluorescently labeled microtubules (EB2), mitochondria, and peroxisomes, including their raw fluorescence intensity images on the left. Zoomed in tracks correspond to supplementary videos.

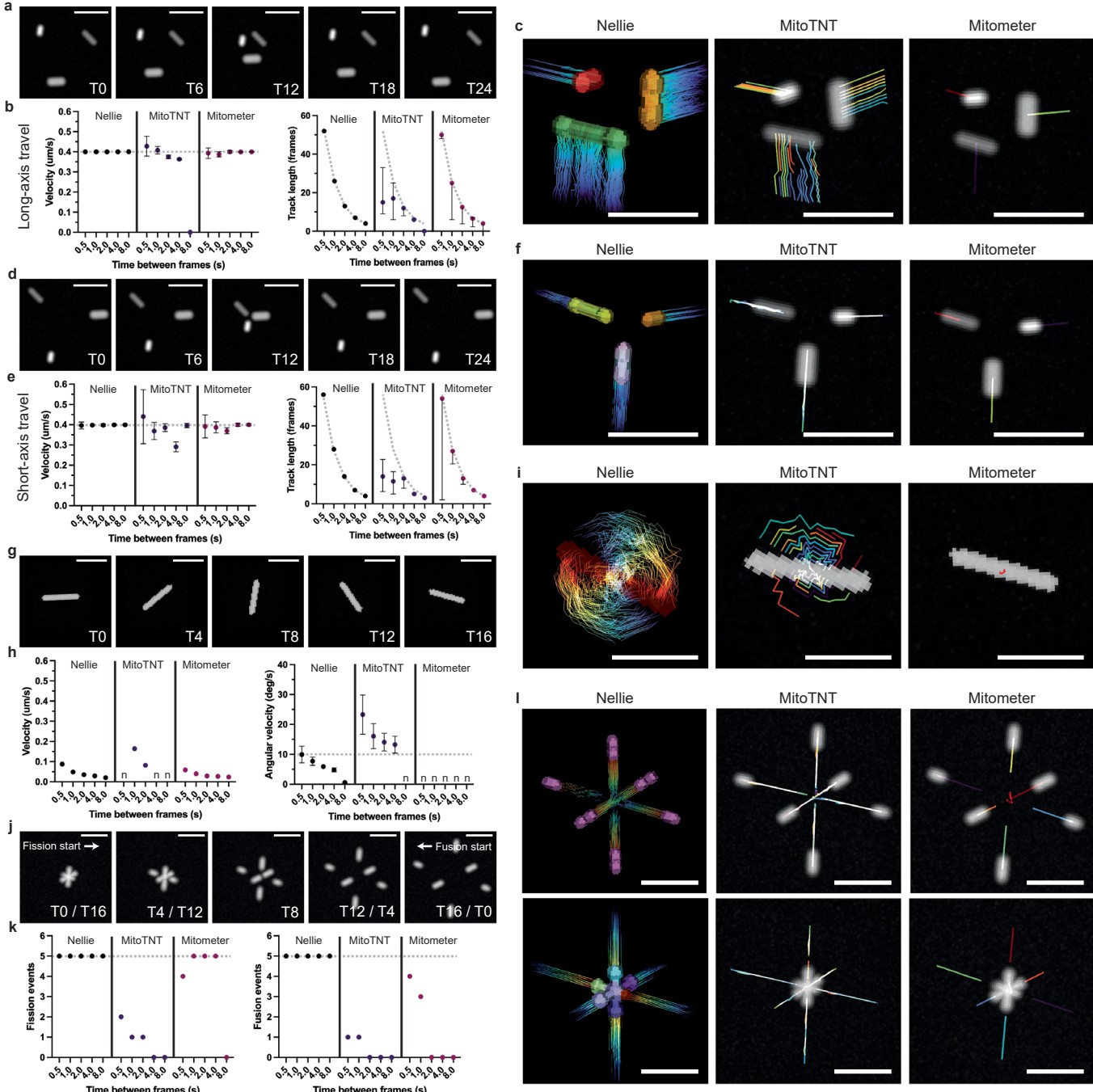

**Extended Data Fig. 4 | Tracking comparison and benchmarking against state-of-the-art organelle tracking algorithms.** Simulated 3D organelle objects of varying characteristics traveling along their long (**a**) or short (**d**) axis, in three orthogonal directions. Linear velocity (left) and track length (right) quantification outputs (**b,e**), and representative branch segmentation (Nellie) and track visualizations for the 1 s/frame temporal resolution (**c, f**) for Nellie, MitoTNT, and Mitometer. Points are mean ±95% CI (left) and median ±upper/lower quartiles (right) of one simulation run. Dotted line is ground truth. **g**, Simulated 3D organelle object rotating along its non-short axis. Linear velocity (left) and angular velocity (right) quantification outputs (**h**), and branch segmentation (Nellie) and track visualizations for the 1 s/frame temporal resolution (**i**) for Nellie, MitoTNT, and Mitometer. Points are single values (left) and mean ±95% CI (right) of one simulation run. Dotted line is ground truth. **j**, Simulated 3D organelle objects of varying lengths undergoing fission events

(left to right) or fusion events (right to left). Number of quantified fission (left) and fusion (right) events (**k**) and branch segmentation (Nellie) and track visualizations for the 1 s/frame temporal resolution (**l**) for Nellie, MitoTNT, and Mitometer. Points are single values of one simulation run. Dotted line is ground truth. For **b**, **e**, and **h** left, for Nellie and Mitometer, N are detected objects, for MitoTNT, N are detected nodes, each for one simulation run. For **h** right, n are detected nodes for all methods. **b**: n = 3, 19, 3; 3, 23, 3; 3, 27, 3; 3, 31, 3; 3, 0, 3 for 0.5, 1.0, 2.0, 4.0, 8.0 time between frames, for Nellie, MitoTNT, and Mitometer, respectively. **e**: n = 3, 10, 3; 3, 19, 3; 3, 23, 3; 3, 22, 3; 3, 12, 3 for 0.5, 1.0, 2.0, 4.0, 8.0 time between frames, for Nellie, MitoTNT, and Mitometer, respectively. **h**, left: n = 1, or 0 if none detected. right: n = 33, 31; 17, 15; 9, 7; 5, 3; 3, 0, for 0.5, 1.0, 2.0, 4.0, 8.0 time between frames for Nellie and MitoTNT, respectively. Scale bars are 5 um in length.

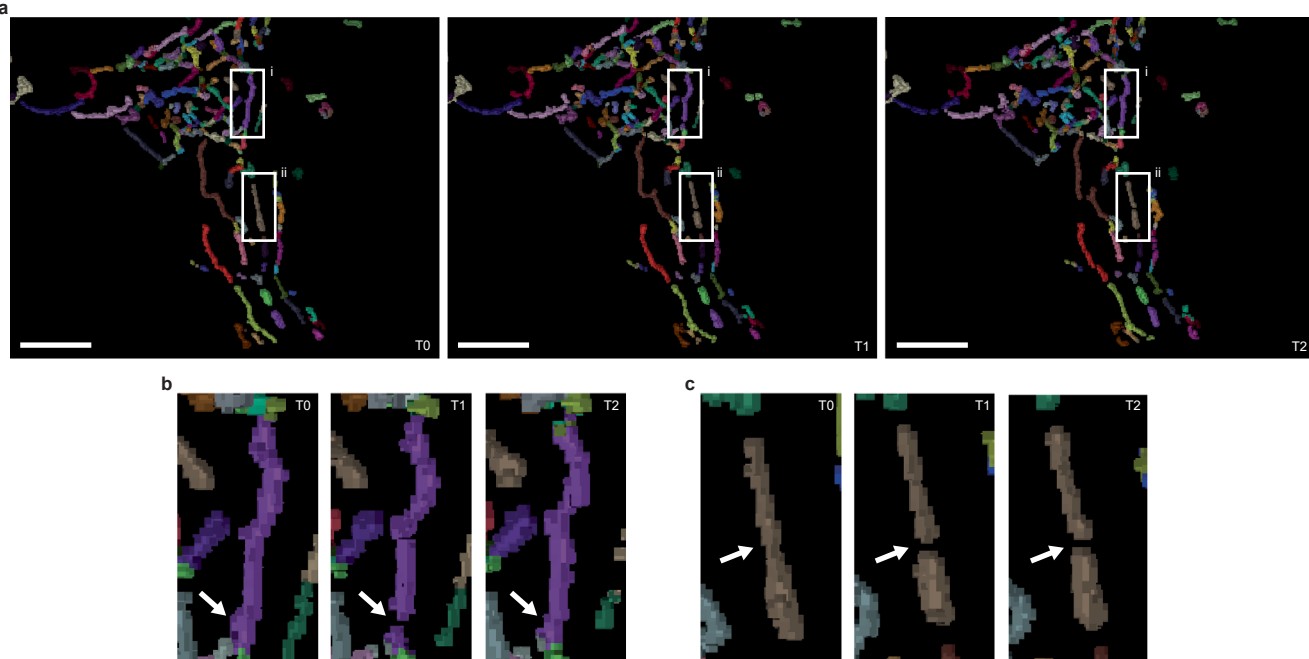

**Extended Data Fig. 5 | Temporal continuity in organelle tracking via forward and backward interpolation of semantic segmentations across frames.** **a**, Three consecutive frames of labeled semantic segmentations of mitochondrial branch objects after undergoing forward and backward flow interpolation for voxel label reassignment based on T0's object labels. **b**, An example of an object (purple, a i.) undergoing a fission event in T1, and fusion event in T2, while keeping track of original T0 labeled voxels, despite the object's connectivity discontinuity between frames. **c**, Another example of an object (brown, a ii.) undergoing a fission event in T1, while keeping track of original T0 labeled voxels, despite the object's connectivity discontinuity in T1 and T2. Scale bars are 10 um in length for the full images, and 2 um in length for the zoomed in images.

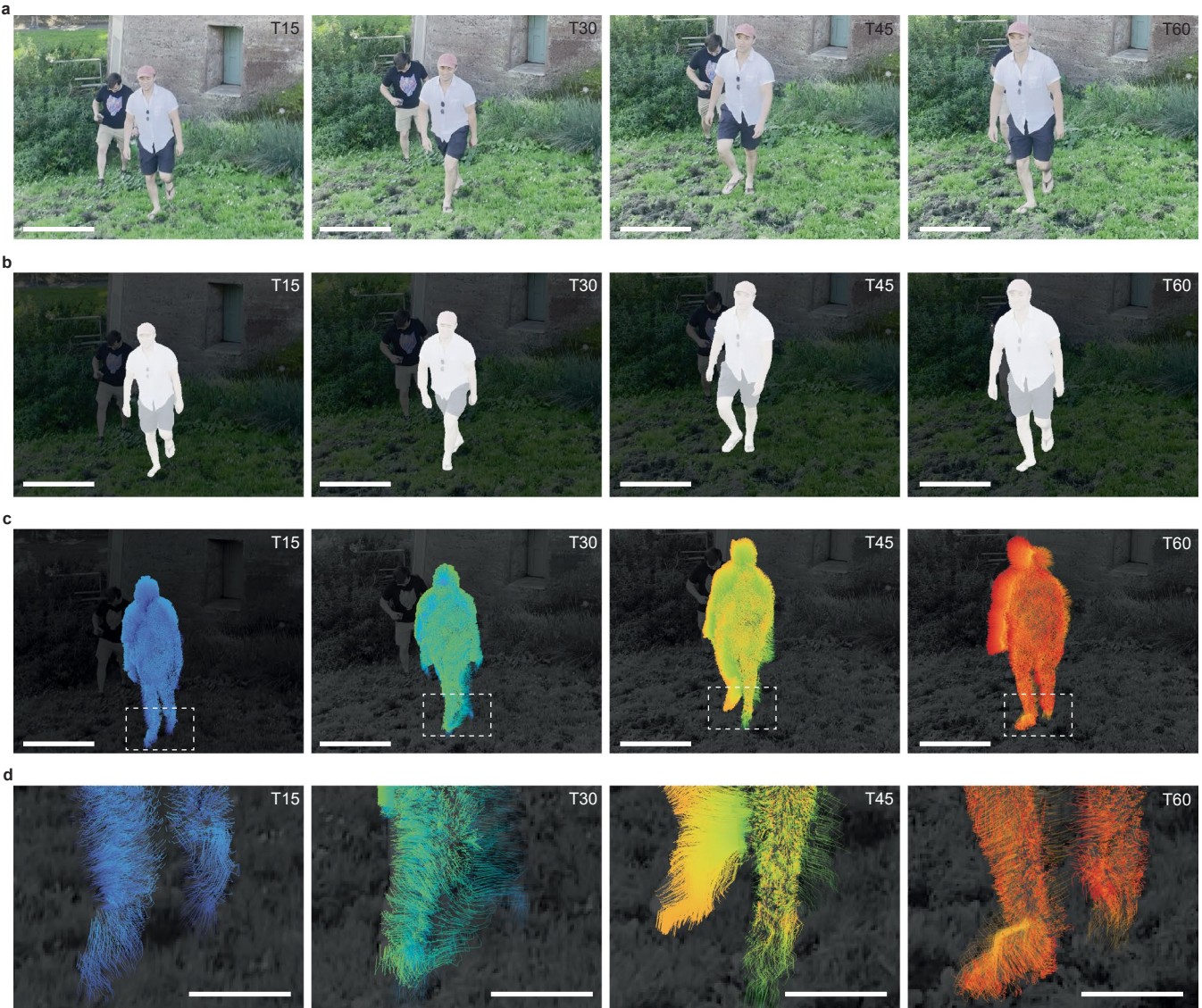

**Extended Data Fig. 6 | Automated tracking and flow interpolation from iPhone camera video frames and corresponding semantic segmentation masks. a**, Four frames at T15, T30, T45, and T60 (left to right) of a 66 frame iPhone video taken at 29.98 fps using an iPhone 12 Pro, with a resolution of 1920×1080 pixels. **b**, The same corresponding frames as (**a**) after segmentation with Meta's Segment Anything model, and selection of semantic segmentations corresponding to the highlighted subject. **c**, The same corresponding frames as (**b**) after running the original video's frames and segmentation mask through Nellie's mocap marking, tracking, and flow interpolation pipelines, generating tracks for 5% of all pixels in the 3D (TYX) mask. **d**, A zoom in portion near a region of relatively abundant movement, the subject's feet, in the corresponding frames of (**c**). Tracks are colored by frame number, where T0 is dark blue and T66 is dark red based on a 'turbo' colormap, where each track is capped to a 5-frame tail. Scale bars are 3 ft in length for **a-c**, and 1 ft in length for **d**.

# Reporting Summary

## Statistics

For all statistical analyses, confirm that the following items are present in the figure legend, table legend, main text, or Methods section.

| n/a | Confirmed | |
|---|---|---|
| ☐ | ☒ | The exact sample size (*n*) for each experimental group/condition, given as a discrete number and unit of measurement |
| ☐ | ☒ | A statement on whether measurements were taken from distinct samples or whether the same sample was measured repeatedly |
| ☐ | ☒ | The statistical test(s) used AND whether they are one- or two-sided *Only common tests should be described solely by name; describe more complex techniques in the Methods section.* |
| ☒ | ☐ | A description of all covariates tested |
| ☐ | ☒ | A description of any assumptions or corrections, such as tests of normality and adjustment for multiple comparisons |
| ☐ | ☒ | A full description of the statistical parameters including central tendency (e.g. means) or other basic estimates (e.g. regression coefficient) AND variation (e.g. standard deviation) or associated estimates of uncertainty (e.g. confidence intervals) |
| ☐ | ☒ | For null hypothesis testing, the test statistic (e.g. *F*, *t*, *r*) with confidence intervals, effect sizes, degrees of freedom and *P* value noted *Give P values as exact values whenever suitable.* |
| ☒ | ☐ | For Bayesian analysis, information on the choice of priors and Markov chain Monte Carlo settings |
| ☒ | ☐ | For hierarchical and complex designs, identification of the appropriate level for tests and full reporting of outcomes |
| ☒ | ☐ | Estimates of effect sizes (e.g. Cohen's *d*, Pearson's *r*), indicating how they were calculated |

*Our web collection on statistics for biologists contains articles on many of the points above.*

## Software and code

Policy information about availability of computer code

Data collection   The Nellie pipeline and its Napari-based plugin is fully written in Python. The Python code and plugin are freely available online via Github at https://github.com/aelefebv/nellie. Supplemental materials for non pipeline-related code can be found via Github at https://github.com/aelefebv/nellie-supplemental. A template for creating Nellie plugins can be found via GitHub at https://github.com/aelefebv/nellie-plugin-example.

Main:
 numpy==1.26.4
 scipy==1.12.0
 scikit-image==0.22.0
 nd2==0.9.0
 ome-types==0.5.2
 pandas==2.2.1
 matplotlib==3.8.3
 napari[all]==0.4.19.post1
 imagecodecs==2024.9.22
 pydantic==2.9.2
 pydantic-core==2.23.4

Supplementary:
alabaster==0.7.16
annotated-types==0.6.0

```
app-model==0.2.6
appdirs==1.4.4
appnope==0.1.4
asttokens==2.4.1
attrs==23.2.0
Babel==2.14.0
build==1.2.1
cachey==0.2.1
certifi==2024.2.2
charset-normalizer==3.3.2
click==8.1.7
cloudpickle==3.0.0
comm==0.2.2
contourpy==1.2.1
cycler==0.12.1
czifile==2019.7.2
dask==2024.4.1
debugpy==1.8.1
decorator==5.1.1
docstring_parser==0.16
docutils==0.20.1
exceptiongroup==1.2.0
executing==2.0.1
fastdist==1.1.6
fonttools==4.53.1
freetype-py==2.4.0
fsspec==2024.3.1
HeapDict==1.0.1
hsluv==5.0.4
idna==3.7
igraph==0.11.6
imageio==2.34.0
imagesize==1.4.1
importlib_metadata==7.1.0
in-n-out==0.2.0
ipykernel==6.29.4
ipython==8.23.0
jedi==0.19.1
Jinja2==3.1.3
joblib==1.4.2
jsonschema==4.21.1
jsonschema-specifications==2023.12.1
jupyter_client==8.6.1
jupyter_core==5.7.2
kiwisolver==1.4.5
lazy_loader==0.4
llvmlite==0.42.0
locket==1.0.0
magicgui==0.8.2
markdown-it-py==3.0.0
MarkupSafe==2.1.5
matplotlib==3.9.2
matplotlib-inline==0.1.6
mdurl==0.1.2
napari==0.4.19.post1
napari-console==0.0.9
napari-plugin-engine==0.2.0
napari-plugin-manager==0.1.0a2
napari-svg==0.1.10
nest-asyncio==1.6.0
networkx==3.3
npe2==0.7.5
numba==0.59.1
numpy==1.26.4
numpydoc==1.7.0
ome-types==0.5.1.post1
packaging==24.0
pandas==2.2.2
parso==0.8.4
partd==1.4.1
pexpect==4.9.0
pillow==10.3.0
Pint==0.23
platformdirs==4.2.0
pooch==1.8.1
prompt-toolkit==3.0.43
```

psutil==5.9.8
psygnal==0.11.0
ptyprocess==0.7.0
pure-eval==0.2.2
pyconify==0.1.6
pydantic==2.7.0
pydantic-compat==0.1.2
pydantic_core==2.18.1
Pygments==2.17.2
PyOpenGL==3.1.7
pyparsing==3.1.2
pyproject_hooks==1.0.0
PyQt5==5.15.10
PyQt5-Qt5==5.15.13
PyQt5-sip==12.13.0
python-dateutil==2.9.0.post0
pytz==2024.1
PyYAML==6.0.1
pyzmq==25.1.2
qtconsole==5.5.1
QtPy==2.4.1
referencing==0.34.0
requests==2.31.0
rich==13.7.1
rpds-py==0.18.0
scikit-image==0.23.1
scikit-learn==1.5.1
scipy==1.13.0
seaborn==0.13.2
shellingham==1.5.4
six==1.16.0
snowballstemmer==2.2.0
Sphinx==7.2.6
sphinxcontrib-applehelp==1.0.8
sphinxcontrib-devhelp==1.0.6
sphinxcontrib-htmlhelp==2.0.5
sphinxcontrib-jsmath==1.0.1
sphinxcontrib-qthelp==1.0.7
sphinxcontrib-serializinghtml==1.1.10
stack-data==0.6.3
superqt==0.6.3
tabulate==0.9.0
tensorly==0.8.1
texttable==1.7.0
threadpoolctl==3.5.0
tifffile==2024.2.12
tomli==2.0.1
tomli_w==1.0.0
toolz==0.12.1
tornado==6.4
tqdm==4.66.2
traitlets==5.14.2
typer==0.12.3
typing_extensions==4.11.0
tzdata==2024.1
urllib3==2.2.1
vispy==0.14.2
wcwidth==0.2.13
wrapt==1.16.0
xsdata==24.3.1
zipp==3.18.1

Data analysis

The Nellie pipeline and its Napari-based plugin is fully written in Python. The Python code and plugin are freely available online via Github at https://github.com/aelefebv/nellie. Supplemental materials for non pipeline-related code can be found via Github at https://github.com/aelefebv/nellie-supplemental. A template for creating Nellie plugins can be found via GitHub at https://github.com/aelefebv/nellie-plugin-example.

For manuscripts utilizing custom algorithms or software that are central to the research but not yet described in published literature, software must be made available to editors and reviewers. We strongly encourage code deposition in a community repository (e.g. GitHub). See the Nature Portfolio guidelines for submitting code & software for further information.

## Data

Policy information about availability of data

All manuscripts must include a data availability statement. This statement should provide the following information, where applicable:
- Accession codes, unique identifiers, or web links for publicly available datasets
- A description of any restrictions on data availability
- For clinical datasets or third party data, please ensure that the statement adheres to our policy

The authors declare that all data supporting the findings of this study are available in the article and its supplementary information files, and raw .tif files are available upon request only, due to the large size and number of tif files present throughout the manuscript. Example images are provided within the GitHub repo for testing at https://github.com/aelefebv/nellie/tree/main/sample_data. Source data for figures are provided with this paper.

## Human research participants

Policy information about studies involving human research participants and Sex and Gender in Research.

| | |
|---|---|
| Reporting on sex and gender | No data from human research participants have been collected for this study. |
| Population characteristics | N/A |
| Recruitment | N/A |
| Ethics oversight | N/A |

Note that full information on the approval of the study protocol must also be provided in the manuscript.

# Field-specific reporting

Please select the one below that is the best fit for your research. If you are not sure, read the appropriate sections before making your selection.

☒ Life sciences        ☐ Behavioural & social sciences        ☐ Ecological, evolutionary & environmental sciences

For a reference copy of the document with all sections, see nature.com/documents/nr-reporting-summary-flat.pdf

# Life sciences study design

All studies must disclose on these points even when the disclosure is negative.

| | |
|---|---|
| Sample size | Sample sizes were chosen to give a reasonable visualization of Nellie's pipeline (Figs. 1-3).<br>Sample sizes were chosen to replicate a typical single-cell imaging experiment (Fig. 4).<br>Sample sizes were chosen to replicate a typical long 3D lightsheet timelapse imaging experiment (Fig. 5)<br>Sample sizes were chosen to showcase exactly 2 cell types in 3 timepoints (Fig. 6) |
| Data exclusions | No data were excluded from analyses. |
| Replication | For multi-organelle unmixing, an 11-fold leave one out cross validation was used as replicates. For multi-mesh graph experiments, the graph neural net was retrained multiple times with similar results for each. The experiments were replicated twice. For simulations, one run was performed as the methods used are deterministic (Ext. Fig. 1), and one simulation set of all conditions listed were performed for each method (Ext. Fig. 2 and Ext. Fig. 4) |
| Randomization | Randomization was not relevant to our studies as there were no inherently randomizable conditions. |
| Blinding | Blinding was not relevant to our studies as no manual or user-biased selection of datasets or analysis was applicable to the presented figures or results. |

# Reporting for specific materials, systems and methods

We require information from authors about some types of materials, experimental systems and methods used in many studies. Here, indicate whether each material, system or method listed is relevant to your study. If you are not sure if a list item applies to your research, read the appropriate section before selecting a response.

## Materials & experimental systems

| n/a | Involved in the study |
|-----|----------------------|
| ☒ | ☐ Antibodies |
| ☐ | ☒ Eukaryotic cell lines |
| ☒ | ☐ Palaeontology and archaeology |
| ☒ | ☐ Animals and other organisms |
| ☒ | ☐ Clinical data |
| ☒ | ☐ Dual use research of concern |

## Methods

| n/a | Involved in the study |
|-----|----------------------|
| ☒ | ☐ ChIP-seq |
| ☒ | ☐ Flow cytometry |
| ☒ | ☐ MRI-based neuroimaging |

# Eukaryotic cell lines

Policy information about cell lines and Sex and Gender in Research

| | |
|---|---|
| Cell line source(s) | U-2 OS cell line was purchased through ATCC (#HTB96), hFB cell line was purchased through LifeLine Cell Technologies (FC-0024) |
| Authentication | The cell lines were not authenticated |
| Mycoplasma contamination | The cell lines were not tested for mycoplasma contamination |
| Commonly misidentified lines (See ICLAC register) | No commonly misidentified cell lines were used in this study. |

