## [Peer Review File · Nature Methods]

Nellie: Automated organelle segmentation, tracking, and hierarchical feature extraction in 2D/3D live-cell microscopy

Corresponding Author: Dr Austin Lefebvre

A version of this paper was originally rejected for publication by Nature Methods, however that decision was reconsidered after appeal by the authors.

Version 0:

Decision Letter:

9th Jul 2024

Dear Austin,

Your Article entitled "Nellie: Automated organelle segmentation, tracking, and hierarchical feature extraction in 2D/3D live-cell microscopy" has now been seen by three reviewers, whose comments are attached. While they find your work of potential interest, they have raised serious concerns which in our view are sufficiently important that they preclude publication of the work in Nature Methods, at least in its present form.

As you will see, the reviewers raise concerns about Nellie's ease-of-use, performance benefits over other (DL-based) tools, and general applicability to diverse organelles/subcellular structures.

Should further experimental data allow you to fully address these criticisms we would be willing to look at a revised manuscript (unless, of course, something similar has by then been accepted at Nature Methods or appeared elsewhere). This includes submission or publication of a portion of this work somewhere else. We hope you understand that until we have read the revised paper in its entirety we cannot promise that it will be sent back for peer-review.

If you are interested in revising this manuscript for submission to Nature Methods in the future, please contact me to discuss your appeal before making any revisions. Otherwise, we hope that you find the reviewers' comments helpful when preparing your paper for submission elsewhere.

Sincerely,
Rita

Rita Strack, Ph.D.
Senior Editor
Nature Methods

Although we cannot publish your paper, it may be appropriate for another journal in the Nature Portfolio. If you wish to explore the journals and transfer your manuscript please use our manuscript transfer portal. You will not have to re-supply manuscript metadata and files, unless you wish to make modifications. For more information, please see our [manuscript transfer FAQ](http://www.nature.com/authors/author_resources/transfer_manuscripts.html?WT.mc_id=EMI_NPG_1511_AUTHORTRANSF&WT.ec_id=AUTHOR) page.

Reviewers' Comments:

Reviewer #1:

Remarks to the Author:

The manuscript introduces Nellie, an automated pipeline for the segmentation, tracking, and feature extraction of various intracellular structures in microscopy images, especially mitochondria. Nellie adapts to image metadata, enhances structural

contrast, and allows for hierarchical segmentation and feature extraction. Internal motion capture markers are generated and tracked using a radius-adaptive pattern matching scheme, facilitating sub-voxel flow interpolation. The paper demonstrates the pipeline's capabilities through two case studies: feature-based classification for organelle unmixing from a single channel and training an unsupervised graph autoencoder to quantify changes in mitochondrial networks following ionomycin treatment. The paper also compares its performance with other organelle segmentation and tracking algorithms (MitoTNT, Mitometer, MitoGraph). Nellie appears applicable to a wide range of images, including other types of organelle segmentation (peroxisome, ER, microtubule, etc.).

Nellie includes a Napari-based GUI and is compatible with all major operating systems. The documentation and code are comprehensive.

I am generally impressed by Nellie's performance. However, there are a few concerns that need to be addressed before the paper is published:

Major Points:

1. Provide a figure for summary or flowchart of the Nellie pipeline.
2. Please provide more details on the training and validation processes, including training and validation loss curves, to check for overfitting or data leakage.
3. During our testing, we found that processing larger images could take several hours. Add a computer runtime comparison of Nellie, MitoMeter, and MitoGraph for Extended Data Fig. 2 for 2D, 3D or time-lapse images at different pixel resolutions.
4. In the text and supplementary notes, the authors have explained the methods for image preprocessing in detail, including multi-scale Gaussian filtering, Minotri thresholding, Frangi filter, LoG filter, hole filling, etc. However, the performance of various steps in Nellie, such as the Frangi filter and mcap marker generation, heavily depends on the choice of parameters. Although the authors have provided initial/default values for these parameters, further details on parameter tuning might be needed. Please discuss the parameter choice in detail. Test these methods on different types of images and add a discussion about finding parameters for different types of microscopy images. Adding these default parameters in the supplementary table may be helpful, too.

Minor Points:

1. There are many algorithms for image processing and calculations. Since this is a methods paper, readers would benefit from some mathematical explanations or pseudocode algorithms illustration. Adding equations when explaining the methods for filtering, segmentation, feature extraction, and graph-based analysis would be helpful.
2. Besides providing the Python codes, Nellie is developed as a plugin for Napari. Napari seems less commonly used. Does the research team plan to develop plugins or APIs to integrate Nellie with tools like ImageJ or CellProfiler? This would enhance its utility in broader research contexts. Lack of integration with other popular bioinformatics and image analysis tools can limit Nellie's utility in multi-step workflows.
3. Including some files in the GitHub repository that explains the purpose of each file in the folder would be helpful to understand and finding the codes.
4. The paper includes an example of tracking mitochondrial fission and fusion in Supplementary Note 7, Extended Data Fig. 5, and Fig. 8. Could Nellie provide the fission and fusion frequency or rate (https://github.com/aelefebv/nellie-supplemental/tree/main/comparisons/simulations/motion/fission_fusion) as one of the outputs in Supplementary Table 1?
5. Please check Extended Data Fig. 2g for any potential mislabels on the pixel size of representative images of generated objects at high and low pixel resolutions.

Reviewer #2:

Remarks to the Author:

The current manuscript introduces Nellie, an automated tool designed for organelle segmentation, tracking, and hierarchical feature extraction in 2D/3D live-cell microscopy images. In this manuscript, Nellie leverages a multi-scale modified Frangi filter for preprocessing, enhancing structural contrast to facilitate robust hierarchical segmentation of organelles. The tool integrates motion capture markers and sub-voxel flow interpolation, providing a comprehensive framework for detailed organelle analysis, aiming to address the challenges associated with the dynamic analysis of organelles. While Nellie represents a sophisticated and potentially powerful tool for organelle analysis, its complexity and the requirement for parameter tuning pose challenges for implementation and consistency. Simplifying the process, automating parameters, and integrating more adaptable machine-learning (ML) methods could enhance its usability and generalisability, making it more accessible to a broader range of researchers. Specifically, the reviewer has the following concerns:

1. The novelty of Nellie is somewhat limited by its dependence on traditional filtering techniques rather than leveraging state-of-the-art ML approaches for segmentation. The key component, Frangi filtering, is a multi-scale vessel enhancement filter primarily used to enhance tubular structures in medical images by analysing the eigenvalues of the Hessian matrix at multiple scales.
2. While Frangi filtering is a powerful traditional method for enhancing and segmenting tubular structures, it differs significantly from modern ML-based segmentation methods. Frangi filtering relies on predefined mathematical operations and eigenvalue analysis, with generally fixed parameters that do not automatically adapt to specific image contexts beyond scale variations. Fine-tuning for optimal performance on different datasets can be time-consuming and requires expertise, making it less user-friendly for non-specialists. The need for parameter tuning means that different users might achieve different results based on their specific settings and adjustments.
3. Frangi filtering is highly effective for enhancing tubular structures such as blood vessels, nerve fibres, and similar organelles, however, it may not perform well on non-tubular structures or highly complex and irregular shapes, making it susceptible to noise and artefacts in images. This is particularly crucial in the study of mitochondria, which exhibit a variety of structures and organisations, from tubular networks to single short sausage-like structures, and even globular shapes under strong stress.

Such structural diversity is not adequately considered in this work, yet it is crucial in biomedical research, especially in evaluating mitochondrial stress and drug effects. The reliance on Frangi filtering may limit the generalisability to non-tubular organelles or structures with more complex geometries.

4. Compared to traditional segmentation based on filtering or thresholding, state-of-the-art models based on deep neural networks, such as U-net, Vision-Transformer, and Swin-Transformer, have been extensively applied to perform image segmentation from cells to organelles. These models not only generate robust segmentation results with high precision and accuracy but also produce multi-parametric measurements informing the structural and dynamic features of the objects with biological significance. The current work faces challenges in demonstrating its technical advancement and precision in results compared to these studies.

5. Although the introduction of motion capture markers is technically sound and could be interesting for mitochondria studies, it is not clear how this can be interpreted within a biological framework. What is its biological significance? How is it correlated with the structure, state, and functions of mitochondria? How can researchers implement this in their studies?

6. Another question regarding motion capture markers is their application to only tubular mitochondria. Given the great structural diversity of mitochondria under different conditions, from tubular to globular shapes, how can this method analyse the non-tubular shapes of mitochondria?

7. The current work lacks versatility testing of its methods. First, only U2OS cells are used throughout the study. How about other non-flat cells, such as HEK293, neurons, or other cell models? Second, only one microscopy mode is used to acquire data. How about other microscopy techniques, particularly Structured Illumination Microscopy (SIM), which is widely used in the study of mitochondrial dynamics? Third, the method is tested only on tubular mitochondria. How about other shapes?

Reviewer #3:

Remarks to the Author:

The manuscript by Lefebvre and colleagues describes an automatic image analysis and tracking solution for the quantification of organelles and their movement based on fluorescence image data. The majority of the examples presented relate to mitochondria (U-2 OS cells stably expressing a fluorescently-tagged mitochondrial matrix protein, COX8A), and the segmentation and quantification of these organelles is impressive. However, the wider applicability of the system to other organelles is unclear, as is the imaging modality that is compatible with the software.

1. According to the Methods section, all images were acquired using a light-sheet microscope. This imaging modality is well-known to produce a superior signal-to-noise ratio in fluorescence imaging, and therefore generate images from which subcellular structures can be readily segmented and analysed. This therefore raises the crucial question of whether 'Nellie' is also able to successfully work with regular confocal images, or images from wide-field microscopes, or whether its functionality is limited to images with high signal-to-noise. Given that probably less than 1% of the cell biology labs in the world have easy access to a light sheet microscope, it is unclear as to the wider applicability of Nellie for day-to-day use by cell biology researchers.

2. The vast majority of the data presented analyse mitochondrial dynamics. This raises the question of the adaptability of 'Nellie' to analyse other complex membrane organelles. These might include the endoplasmic reticulum, dynamic endosomes/lysosomes, and membrane traffic events emanating from the Golgi apparatus. Figure 4 does show some data using a Golgi marker (a truncated variant of the glycosylation enzyme B4GALT1), however this organelle appears highly unusual as a series of punctate structure, not the characteristic juxta-nuclear membranes. As such, the sparse examples from other important membrane compartments in the cell suggest that 'Nellie' is probably only effective at analysis of mitochondrial dynamics. The endomembrane system of cells has many highly dynamic organelles, and further examples need to be provided to address the wider applicability of 'Nellie'.

3. It is unclear who this manuscript is targeted at in its current form. The vast majority of the text is written in a way that is highly specialised and directed towards experts who develop image analysis tools. However, the end-point users will inevitably be cell biologists, and for that community, the manuscript is not particularly 'digestible'. In its current form, the manuscript is far more suited to a specialised informatics journal.

Other comments:

1. Supplementary Figure 1 seems to show the software interface. This is a little difficult to appreciate, and yet is a critically important feature of the work. More emphasis needs to be given to this.

2. It is unclear what the output files are that are generated by 'Nellie'. Are these all tabulated data and can they then be interrogated by downstream image data analytics software. Is specialised software needed to work with .pkl and .npy file formats?

3. What level of coding skills are needed for the installation and use of 'Nellie'? Relating to my point above; the manuscript is highly technical with respect to image analysis methodology, but at the end of the day it will be cell biologists who will use the tool.

4. What computational power is needed to run 'Nellie', and importantly what is the throughput? Similarly, what do typical output

data sizes look like?

5. Can the user choose specific features to extract, or is it default that all feature information is extracted?
6. Can 'Nellie' be used to segment and quantify organelles in fixed cells, or is it only compatible with live cell / time-lapse data?
7. In Supplementary Table 1, in the Branches and Organelle features, the feature "area_raw" is used. For this feature, for 2D images, the unit is μm^2 and for 3D images it is μm^3 . How is μm^3 a measure of area, is this not a volumetric measurement?
8. Why is simulated data used in Supplementary Notes 2 and 7? Please see my comment earlier with respect to more examples of organelle dynamics within the endomembrane system.
9. The Methods section mentions an ECFP-H2B construct, but I do not see any data in the manuscript using this construct.
The Methods sections mentions 'drug injections' – what cells were injected with drugs?

** For Nature Portfolio general information and news for authors, see <http://npg.nature.com/authors>.

Version 1:

Decision Letter:

29th Jul 2024

Dear Austin,

Thank you for your letter asking us to reconsider our decision on your Article, "Nellie: Automated organelle segmentation, tracking, and hierarchical feature extraction in 2D/3D live-cell microscopy". After careful consideration we have decided that we are willing to consider a revised version of your manuscript that is updated mostly as you've outlined.

During your revision, we ask that you compare to one custom, fine-tuned method (like U-nets) on at least two organelles. The point of this exercise is to show what, if any, performance gains a user will have with Nellie compared to a deep learning model that is well-suited to analyzing their particular data (vs a generalist algorithm like Nellie).

We also ask that you make video tutorials for software installation and for a single-use case.

Finally, we don't require that you do as much for another organelle as you've done for mitos, but I think it would strengthen the paper if you chose one less regular structure (like ER or Golgi) and did a little more analysis to showcase general applicability. Otherwise, we think the additional compartments you describe will be great to add to the paper to emphasize versatility.

- * include a point-by-point response to our referees and to any editorial suggestions
- * please underline/highlight any additions to the text or areas with other significant changes to facilitate review of the revised manuscript
- * address the points listed described below to conform to our open science requirements
- * ensure it complies with our general format requirements as set out in our guide to authors at www.nature.com/naturemethods
- * resubmit all the necessary files electronically by using the link below to access your home page

Link Redacted

We hope to receive your revised paper within three months. If you cannot send it within this time, please let us know. In this event, we will still be happy to reconsider your paper at a later date so long as nothing similar has been accepted for publication at Nature Methods or published elsewhere.

OPEN SCIENCE REQUIREMENTS

REPORTING SUMMARY AND EDITORIAL POLICY CHECKLISTS

When revising your manuscript, please submit reporting summary and editorial policy checklists.

DATA AVAILABILITY

CODE AVAILABILITY

Please include a "Code Availability" subsection in the Online Methods which details how your custom code is made available. Only in rare cases (where code is not central to the main conclusions of the paper) is the statement "available upon request" allowed (and reasons should be specified).

MATERIALS AVAILABILITY

ORCID

Nature Methods is committed to improving transparency in authorship. As part of our efforts in this direction, we are now requesting that all authors identified as 'corresponding author' on published papers create and link their Open Researcher and Contributor Identifier (ORCID) with their account on the Manuscript Tracking System (MTS), prior to acceptance. This applies to primary research papers only. ORCID helps the scientific community achieve unambiguous attribution of all scholarly contributions. You can create and link your ORCID from the home page of the MTS by clicking on 'Modify my Springer Nature account'. For more information please visit <http://www.springernature.com/orcid>.

Sincerely,

Rita

Rita Strack, Ph.D.
Senior Editor
Nature Methods

Version 2:

Decision Letter:

Our ref: NMETH-A56495B

11th Nov 2024

Dear Austin,

Thank you for submitting your revised manuscript "Nellie: Automated organelle segmentation, tracking, and hierarchical feature extraction in 2D/3D live-cell microscopy" (NMETH-A56495B). It has now been seen by the original referees and their comments are below. The reviewers find that the paper has improved in revision, and therefore we'll be happy in principle to publish it in Nature Methods, pending minor revisions to satisfy the referees' final requests and to comply with our editorial and formatting guidelines.

With regards to the remaining concerns, please update your discussion of limitations of the methods, especially if it underperforms on certain organelles, like the ER, relative to models dedicated to handling a specific organelle. In addition, we ask that you please briefly discuss existing methods that could not be compared (like ERNet, MitoSegNet, and AICS) and why in the discussion and not only the rebuttal. Please provide a point-by-point rebuttal summarizing the changes upon resubmission.

TRANSPARENT PEER REVIEW

ORCID

Sincerely,
Rita

Rita Strack, Ph.D.
Senior Editor
Nature Methods

Reviewer #1 (Remarks to the Author):

I appreciate the authors' efforts in answering the questions and concerns. All the issues were solved. The manuscript is largely improved.

As a method paper, the assisted documentation, videos, and annotations are now more accessible to readers. Runtime and performance comparisons were added. A case study on the less regular structure (ER) was added. Comparisons with fine-tuned deep learning models (Swin UNETR) were added.

Reviewer #1 (Remarks on code availability):

The assisted documentation, videos, and annotations of codes are more accessible to the readers now.

Reviewer #2 (Remarks to the Author):

Fig1. Need to improve the presentation, some subfigs are hard to see.

Fig. 6. The authors claim to analyse 3D volumetric images. But no volumetric structure of ER is presented. The skeletons are just 2D. Clearly the ER segmentation of tubular network in the perispherical region is overestimated, which should be largely ER sheet. This is likely due to the poor segmentation in the dense tubular region and the sheet. Existing methods including Pain et al., 2019 and Lu et al., 2023, have already achieved much better classification of ER tubules and sheets. Therefore the authors need to discuss their method's limitation, such as overestimation of tubular network, at this point to avoid misleading.

There is still a lack of biological significance discussion. Why are the features or dynamics analysed by this method important?

Reviewer #3 (Remarks to the Author):

Many thanks to the authors who have taken my comments and suggestions seriously. The rebuttal and associated revisions are comprehensive. The revised manuscript is significantly improved from the original version. Of particular note is demonstration that 'Nellie' can indeed be applied to analyze multiple organelles beyond mitochondria; this is a key addition. The additional documentation for use of Nellie is also an extremely valuable addition to the revised manuscript, and will be appreciated by users.

My only remaining comment - which does need to be addressed given the nature of the work - is that all the figures showing images need to have appropriate scale bars added. On review of the revised manuscript (containing additional, and very welcome examples), it is clear that different 'zooms' are used across the various data presented, and as such a scale bar is vital as a reference for readers of the work to understand the images that they are looking at. This comment applies to the figures in the main body text, and also in the supporting data.

Professor Jeremy C Simpson

Version 3:

Decision Letter:

21st Jan 2025

Dear Austin,

I am pleased to inform you that your Article, "Nellie: Automated organelle segmentation, tracking, and hierarchical feature extraction in 2D/3D live-cell microscopy", has now been accepted for publication in Nature Methods. The received and accepted dates will be May 20, 2024 and Jan 21, 2025. This note is intended to let you know what to expect from us over the next month or so, and to let you know where to address any further questions.

Over the next few weeks, your paper will be copyedited to ensure that it conforms to Nature Methods style. Once your paper is typeset, you will receive an email with a link to choose the appropriate publishing options for your paper and our Author Services team will be in touch regarding any additional information that may be required. It is extremely important that you let us know now whether you will be difficult to contact over the next month. If this is the case, we ask that you send us the contact information (email, phone and fax) of someone who will be able to check the proofs and deal with any last-minute problems.

Authors may need to take specific actions to achieve [compliance](https://www.springernature.com/gp/open-research/funding/policy-compliance-faqs) with funder and institutional open access mandates. If your research is supported by a funder that requires immediate open access (e.g. according to [Plan S principles](https://www.springernature.com/gp/open-research/plan-s-compliance)) then you should select the

gold OA route, and we will direct you to the compliant route where possible. For authors selecting the subscription publication route, the journal's standard licensing terms will need to be accepted, including [self-archiving policies](https://www.springernature.com/gp/open-research/policies/journal-policies). Those licensing terms will supersede any other terms that the author or any third party may assert apply to any version of the manuscript.

If you are active on Twitter/X, please e-mail me your and your coauthors' handles so that we may tag you when the paper is published.

Best regards,
Rita

Rita Strack, Ph.D.
Senior Editor
Nature Methods

Visit the Springer Nature Editorial and Publishing website at http://editorial-jobs.springernature.com?utm_source=ejP_NMeth_email&utm_medium=ejP_NMeth_email&utm_campaign=ejp_Nmeth for more information about our career opportunities. If you have any questions please click [here](mailto:editorial.publishing.jobs@springernature.com).

Open Access This Peer Review File is licensed under a Creative Commons Attribution 4.0 International License, which permits use, sharing, adaptation, distribution and reproduction in any medium or format, as long as you give appropriate credit to the original author(s) and the source, provide a link to the Creative Commons license, and indicate if changes were made. In cases where reviewers are anonymous, credit should be given to 'Anonymous Referee' and the source. The images or other third party material in this Peer Review File are included in the article's Creative Commons license, unless indicated otherwise in a credit line to the material. If material is not included in the article's Creative Commons license and your intended use is not permitted by statutory regulation or exceeds the permitted use, you will need to obtain permission directly from the copyright holder.

We expect these concerns to be addressable within 2 weeks.

Major concerns

Reviewer 1:

Reviewer 1 presents four major concerns that we summarize below, two of which (2, 4) we believe stem from a misunderstanding of the paper, and the other two (1, 3) which we believe can be easily addressed and do not point at flaws within our paper or code.

1. “Provide a figure for summary or flowchart of the Nellie pipeline”
 - a. This is a great idea and we agree including a flowchart figure for Nellie’s pipeline would help readability.
2. “Please provide more details on the training and validation processes, including training and validation loss curves, to check for overfitting or data leakage”
 - a. We are somewhat confused by this comment, as Nellie does not have a training and validation loss since no part of the pipeline is ML-based.
 - b. We believe the reviewer could also be referring to the graph neural network (GNN) in our second biological application, which does undergo training. If this is the case, we originally included the loss curves in the preprint, but removed it in lieu of other extended data figures that we believed were more helpful. We would be happy to reinclude such a figure after thorough checking of overfitting and data leakage. We would also be happy to include more details in our supplementary section about our GNN’s model architecture, and training and validation process.
3. “Add a computer runtime comparison of Nellie, MitoMeter, and MitoGraph for Extended Data Fig. 2 for 2D ,3D or time-lapse images at different pixel resolutions”
 - a. We are more than happy to provide a runtime comparisons graph. Within the graph, we can include the run time for each portion of the pipeline in the case that the user only wants to perform segmentation, for example. We can include these metrics for both 2D and 3D timelapses, across different pixel resolutions, and varying systems (e.g. Linux CPU, Linux GPU, Mac CPU, Windows CPU, Windows GPU).
4. “Please discuss the parameter choice in detail. Test these methods on different types of images and add a discussion about finding parameters for different types of microscopy images”
 - a. We believe that the reviewer has overlooked several important sections within our paper and, by consequence, misunderstood key features of our pipeline that make Nellie generalizable across datasets without the need for manual parameter tuning by the end user. The reviewer asks for a discussion on parameter choice for both Frangi filtering and mocap marker generation. These discussions are indeed within our paper. Within these sections we discuss

parameter choice in detail, include the automation of parameter adjustment, and test these methods within our simulations. We list the following sections for reference:

- i. Sections on our modified Frangi filter in the main paper ("Multi-scale adaptive filters enhances structural features of organelles" L95) dedicated to describing the very automation that the reviewer claims is not present, and in our supplementary materials ("Supplementary Note 1: Optimization of multi-scale Gaussian filtering for anisotropic images in cellular fluorescence microscopy datasets" sL20, "Supplementary Note 3: The Minotri threshold" sL181, "Supplementary Note 4: Frangi filter parameter selection" sL222, "Supplementary Note 5: Preprocessing refinement techniques for structural contrast enhancement" sL283) dedicated to describing the automation of this parameter selection in great detail.
 - ii. Sections on our motion capture marker generation ("Motion capture markers are generated for downstream tracking" L189) dedicated to describing the higher level automation of motion capture marking, and in our supplementary materials ("Supplementary Note 8: Efficient distance transformation using k-dimensional trees" sL494, "Supplementary Note 9: Methodology for multi-scale and adaptive local maxima detection" sL521) dedicated to the detailed description of this automation process.
- b. We can add a part within our discussion section to emphasize that these processes are indeed automated.

Reviewer 2:

Reviewer 2 also presents four major points of concern listed as seven different sections, which we summarize below. We have serious concerns about the reviewers' responses. We believe 3 of the four concerns (1, 2, 3) are unwarranted, and that one concern (4) is valid and can be readily addressed.

1. "The novelty of Nellie is somewhat limited by its dependence on traditional filtering techniques... Frangi filtering relies on predefined mathematical operations and eigenvalue analysis, with generally fixed parameters that do not automatically adapt to specific image contexts beyond scale variations... The need for parameter tuning means that different users might achieve different results based on their specific settings and adjustments... it may not perform well on non-tubular structures or highly complex and irregular shapes... Such structural diversity is not adequately considered in this work, yet it is crucial in biomedical research..."
 - a. The reviewer claims our novelty is limited by our pipeline's dependence on the Frangi filter's limitations, which has parameters that "do not automatically adapt to specific image contexts beyond scale variations", when in fact, we explicitly and specifically modify the filter to allow for automatic adaptation to specific image contexts, including and beyond scale variations. The details of these modifications are discussed in great detail both within the paper and within the supplementary materials in the same sections as listed in reviewer 1's concern 4:

("Multi-scale adaptive filters enhances structural features of organelles" L95) dedicated to describing the very automation that the reviewer claims is not present, and in our supplementary materials ("Supplementary Note 1: Optimization of multi-scale Gaussian filtering for anisotropic images in cellular fluorescence microscopy datasets" sL20, "Supplementary Note 3: The Minotri threshold" sL181, "Supplementary Note 4: Frangi filter parameter selection" sL222, "Supplementary Note 5: Preprocessing refinement techniques for structural contrast enhancement" sL283) dedicated to describing the automation of this parameter selection in great detail.

- b. The reviewer also claims that the Frangi filter parameter “fine-tuning for optimal performance on different datasets can be time-consuming and requires expertise, making it less user-friendly for non-specialists” and that “users might achieve different results based on their specific settings and adjustments.” We intentionally remove from the user the ability to fine-tune and manually adjust any settings within the GUI, as this process is completely automated. *This quote makes it clear to us that the reviewer did not attempt to use Nellie or attempt to read our paper in full.*
 - c. Furthermore, the reviewer states that, as Frangi filtering is only “highly effective for enhancing tubular structures”, it may not perform well on mitochondria, which has a diversity of shapes. The reviewer claims that “such structural diversity is not adequately considered in this work.” Not only do we explicitly discuss our modifications to the Frangi filter to make it generalizable to non-tubular structures, we quantitatively test our modifications in a large panel of diversely structured simulated datasets ranging from small round objects to large round objects to small tubular objects to large tubular objects and everything in between across a variety of noise levels (see “Supplementary Note 2: Segmentation comparison and benchmarking against state-of-the-art organelle segmentation algorithms” sL57). *We worry that the reviewer’s preconceived notions about traditional Frangi filters in other contexts has negatively biased their review of our novel approach, and caused them to not read the paper in full.* We are unsure how to make these points more clear as they are heavily detailed throughout the paper and supplementary materials. As with reviewer 1, we can ensure to include another part within our discussion section to emphasize all of these points, and another part within our discussion of the Frangi filter to emphasize our methods’ differences compared to the traditional filter.
2. “Compared to traditional segmentation based on filtering or thresholding, state-of-the-art models based on deep neural networks, such as U-net, Vision-Transformer, and Swin-Transformer, have been extensively applied to perform image segmentation from cells to organelles... The current work faces challenges in demonstrating its technical advancement and precision in results compared to these studies.”
 - a. We have conducted extensive literature searches both before starting this project and during the writing of the manuscript for DL-based tools that perform automated generalized segmentation of 3D organelles across modalities and

datasets, without the need for manually generating ground-truth labels for model finetuning, and *we are confident in stating that no such tools currently exist.* Since receiving the reviewer's comments, we have attempted to find the tools mentioned by the reviewer based on the higher-level architecture names provided (U-net, Vision-Transformer, and Swin-Transformer). We also would like to point out that the Swin-Transformer is itself a type of Vision-Transformer. We could only identify 3D DL-based segmentation models for organelles in *electron microscopy images*, cellular tomography images, or *cell or nucleus (not pan-organelle) segmentation*:

- i. U-Nets: <https://doi.org/10.1101/2024.02.19.580954>,
<https://doi.org/10.1038/s41592-018-0261-2>,
<https://doi.org/10.1002/jemt.24548>,
<https://doi.org/10.48550/arXiv.2303.03876>,
<https://doi.org/10.1101/2021.05.23.445351>
- ii. Vision and Swin-Transformers:
<https://doi.org/10.1016/j.jplph.2024.154236>,
<https://doi.org/10.1038/s41477-023-01527-5>,
<https://doi.org/10.1101/2024.04.05.588365>,
<https://doi.org/10.1109/ICSIP55141.2022.9886148>.
- iii. Comprehensive review of existing tools:
<https://doi.org/10.1016/j.tcb.2023.10.010>

- b. Furthermore, we could not find implementations even for 2D datasets that do not require a substantial amount of manual annotation and fine-tuning of the model for specific organelles and imaging modalities, a complexity which we remove from the user with Nellie. We can make it a point to reemphasize this gap within our discussion section. If we are mistaken, and such methods do in fact exist, we would kindly also ask the reviewer to send them our way as we would very much like to compare our method against these models.
3. "Although the introduction of motion capture markers is technically sound and could be interesting for mitochondria studies, it is not clear how this can be interpreted within a biological framework. What is its biological significance? How is it correlated with the structure, state, and functions of mitochondria? How can researchers implement this in their studies?... how can this method analyse the non-tubular shapes of mitochondria?"
 - a. The reviewer mentions that our motion capture markers are technically sound. However, the reviewer also suggests that we use them to generate insights about mitochondria, or that it contains meaningful biological significance. The reviewer also has concerns about generalizability of these motion capture markers to non-tubular objects, and claims we have only shown "application to only tubular mitochondria." We are confused by these comments as we use, and explicitly describe, these motion capture markers as linkage points for motion interpolation between frames. *We do not claim to use them to generate insights on organelles or that they intrinsically contain biological significance.* We can reinforce this point within our motion capture marker section.

- b. Additionally, we explicitly show motion capture markers overlaid on tubular and round mitochondria, as well as round lysosomes in the first figure of our main paper (L118). We also show quantitatively that tracking of objects using these motion capture markers performs objectively well on a range of simulated object widths and lengths, across various noise levels (see “Supplementary Note 7: Tracking comparison and benchmarking against state-of-the-art organelle tracking algorithms” sL337). *These comments again suggest a lack of proper revision of the paper by the reviewer.*
4. “The current work lacks versatility testing of its methods... other cell models... only one microscopy mode... only tubular mitochondria.”
 - a. Although we have included real examples of lysosomes, mitochondria, and Golgi, we do agree this is a valid concern which we have only fully addressed via simulations rather than real data. To this point, however, we have already generated much data across multiple cell types, organelles, and imaging modalities, and can generate more that we can include as a figure as part of the manuscript to visually demonstrate Nellie’s generalizability across datasets.

Reviewer 3:

Reviewer 3 presents two major concerns across 3 paragraphs, as summarized below. We believe the concerns are valid (1, 2) but can be readily addressed.

1. “... all images were acquired using a light-sheet microscope... This therefore raises the crucial question of whether ‘Nellie’ is also able to successfully work with regular confocal images, or images from wide-field microscopes, or whether its functionality is limited to images with high signal-to-noise... The vast majority of the data presented analyse mitochondrial dynamics. This raises the question of the adaptability of ‘Nellie’ to analyse other complex membrane organelles... further examples need to be provided to address the wider applicability of ‘Nellie’.”
 - a. The reviewer mentions, in two paragraphs, concerns on the versatility of Nellie’s use across imaging modalities, as our included data came from only light sheet datasets, and across organelles, as our included data came from only mitochondria, Golgi, and lysosomes. As with reviewer 2’s 4th concern, we can address this concern quite readily, as we have already generated much of this data across imaging modalities, cell types, and organelles (including those within the endomembrane such as the endoplasmic reticulum), and could apply Nellie to available open-source datasets from varying groups. We can include these data as a figure as part of the manuscript to visually demonstrate Nellie’s generalizability across datasets.
2. “It is unclear who this manuscript is targeted at in its current form... the end-point users will inevitably cell biologists, and for that community, the manuscript is not particularly ‘digestible’.”

- a. The reviewer mentions their concern for the suitability of our manuscript to Nature Methods due to its specialized image analysis-oriented text, despite its end-point use being largely driven by cell biologists. We acknowledge that we include detailed descriptions of Nellie’s methodology that can be quite sophisticated and build upon or introduce a variety of complex image analysis techniques. However we ensure to include digestible figures, and exciting, biologically relevant use cases to strongly emphasize immediate relevance of our tool. We believe the flowchart mentioned by reviewer 1 will also aid cell biologists in digesting the content. Importantly, our goal was to write a comprehensive methods paper to empirically justify our methodology and allow more code-savvy cell biologists (like ourselves) to extend and enhance Nellie, while keeping the user-facing application as simple as possible for users without coding experience. As an example, authors of novel microscope builds who publish in Nature Methods typically do not expect cell biologists themselves to recreate these microscopes by way of their detailed methods sections, but rather aim to 1. inform the cell biologist of their method, 2. empirically justify the tool’s benefits and 3. showcase exciting novel biological use cases made possible by the tool’s novelty, ones other cell biologists can replicate or extend upon when using the same microscope. Indeed, our goal is the same. Users can easily install Nellie via point-and-click methods. Nellie is run via an intuitive point-and-click GUI with only 2 clicks of the mouse to run a sample or folder of samples. Nellie’s quantification outputs are .csv files that can be opened in, for example, Excel. Nellie’s intermediate images (segmentation masks, filtered image, etc.), should the user want them, are .tif files that can be opened in, for example ImageJ or Napari via drag and drop, but are there mainly to serve images to the GUI upon the user’s click. Thus, we believe that the comprehensive methods section, the simplicity of Nellie’s user experience and user interface, and the clear and impactful biological use cases throughout the paper make our manuscript well suited for Nature Methods.

Minor concerns

Reviewer 1:

Reviewer 1’s minor concerns can be readily addressed and we believe the concerns do not invalidate our results or code. Thus, we only briefly address the points.

1. “Adding equations when explaining the methods for filtering, segmentation, feature extraction, and graph-based analysis would be helpful”
 - a. We can certainly provide equations and pseudocode behind our algorithms within Nellie’s pipeline within the supplementary materials.
2. “Does the research team plan to develop plugins or APIs to integrate Nellie with tools like ImageJ or CellProfiler?”
 - a. We have already developed a command line interface (CLI) based method of running Nellie, both on individual files and batch folders. This method avoids using Napari. All intermediates can still be opened via ImageJ, for example.

Nellie is usable with an API already, with encapsulating function calls for each part of the pipeline. We can make this more clear within the documentation and code itself.

3. "Including some files in the GitHub repository that explains the purpose of each file in the folder would be helpful to understand and finding the codes."
 - a. We can include additional details on each section of the codebase within our GitHub repository.
4. "Could Nellie provide the fission and fusion frequency or rate... as one of the outputs in Supplementary Table 1?"
 - a. The code for fission and fusion detection is already written (as pointed out by the reviewer) and we can certainly make that available within the list of outputs.
5. "Please check Extended Data Fig. 2g for any potential mislabels on the pixel size of representative images of generated objects at high and low pixel resolutions."
 - a. The labeling is correct. Smaller pixel sizes create larger objects, and vice versa if the objects remain a constant size.

Reviewer 2:

No minor concerns listed.

Reviewer 3:

As with reviewer 1, reviewer 3's minor concerns can also readily be addressed, and we again believe the concerns do not invalidate our results or code. Thus, we briefly address the points in bulk.

1. "Supplementary Figure 1 seems to show the software interface. This is a little difficult to appreciate, and yet is a critically important feature of the work. More emphasis needs to be given to this."
 - a. A more detailed figure of the software interface can be provided.
2. "It is unclear what the output files are that are generated by 'Nellie'. Are these all tabulated data and can they then be interrogated by downstream image data analytics software. Is specialised software needed to work with .pkl and .npy file formats?"
 - a. All output files for end users are indeed tabulated .csv files and can be opened with, for example, Excel, R, or any other data analytics software. Intermediate images that are used by the software are stored as .tif files. Intermediate objects and matrices for back end flow calculation and object linkages are stored as .pkl and .npy files, both commonly used file types for Python-based information storage and retrieval. We do not expect end users to use these files. We can make it clear, via additional folders, which outputs are provided for the end user (the .csv files) and those provided for Nellie (everything else).
3. "What level of coding skills are needed for the installation and use of 'Nellie'? Relating to my point above; the manuscript is highly technical with respect to image analysis methodology, but at the end of the day it will be cell biologists who will use the tool."
 - a. No coding skills are required. In fact we detail the entire installation process within our github readme. It entails a single "pip install" to install Nellie, and requires only point-and-click for usage.

4. “What computational power is needed to run ‘Nellie’, and importantly what is the throughput? Similarly, what do typical output data sizes look like?”
 - a. All requirements for Nellie’s usage is listed in the readme, as well as what Nellie has been tested on. Typical output data sizes depend on the input data. Nellie is also adaptable to the system, and is able to run over the CPU, but can take advantage of NVIDIA GPUs if they are available. This can be added to the readme.
5. “Can the user choose specific features to extract, or is it default that all feature information is extracted?”
 - a. Currently, all features get extracted by default. But we can change this to have the user specify up front which features get extracted. Users can also export individual metrics via the “Analyze” tab within Nellie, which we detail in the repo’s comprehensive readme.
6. “Can ‘Nellie’ be used to segment and quantify organelles in fixed cells, or is it only compatible with live cell / time-lapse data?”
 - a. Nellie can be used to segment datasets without a temporal dimension (such as with fixed cells), meaning even single frames can be segmented. We can make this more clear within the readme and paper.
7. “In Supplementary Table 1, in the Branches and Organelle features, the feature “area_raw” is used. For this feature, for 2D images, the unit is μm^2 and for 3D images it is μm^3 . How is μm^3 a measure of area, is this not a volumetric measurement?”
 - a. μm^3 is indeed a measure of volume. We retain the name “area_raw” since the measurement process is the same within the code. We will make this more clear within our supplementary table 1.
8. “Why is simulated data used in Supplementary Notes 2 and 7? Please see my comment earlier with respect to more examples of organelle dynamics within the endomembrane system.”
 - a. Simulated data is used in Supplementary Notes 2 and 7 in order to generate ground truth data to objectively quantify performance of our, and other softwares. Manual segmentation and annotation is intrinsically biased. Also there does not exist an easy way to manually segment organelles in 3D. We can include additional real dataset examples of organelle dynamics within the endomembrane system as mentioned above.
9. “The Methods section mentions an ECFP-H2B construct, but I do not see any data in the manuscript using this construct.”
 - a. Although the cell line contained an ECFP-H2B construct, which we felt should be mentioned, that marker was not used in any part of our manuscript. Only the other two markers (mEGFP targeted to the mitochondrial matrix with a human COX8 and the first 82 residues of B4GALT1 tagged with mScarlet) were used.
10. “The Methods sections mentions ‘drug injections’ – what cells were injected with drugs?”
 - a. Cells were not injected with drugs. Instead, drugs (Ionomycin) were injected into the media as a treatment procedure to induce fission-like pearling events in mitochondria (L451) in the GNN case study. We can make this more clear within the methods section.

Dear Rita,

We are pleased to resubmit our revised manuscript entitled “Nellie: Automated organelle segmentation, tracking, and hierarchical feature extraction in 2D/3D live-cell microscopy” for consideration in Nature Methods. We are grateful for the thoughtful and constructive feedback provided by you, the other editors, and the reviewers, which has greatly enhanced the quality and clarity of our work.

In response to the comments and suggestions, we have made substantial revisions to the manuscript. The significant changes are highlighted in red in both the main text and the supplementary materials. Below is a summary of the major revisions:

1. **Comparison with Custom Fine-Tuned Deep Learning Models:** To address the request for comparison with custom, fine-tuned deep learning methods, we developed and trained nine custom Swin UNETR segmentation models, each specialized for a specific organelle (mitochondria, desmosomes, actin, actomyosin, ER, Golgi apparatus, nucleoli, tight junctions, and tubulin), as well as a combined model trained on all organelles. Our findings underscore Nellie’s robust generalizability across diverse organelles and microscopy modalities without the need for fine-tuning, highlighting its strength over specialized deep learning models that require extensive training data and parameter adjustments.
2. **Creation of Video Tutorials:** We have produced comprehensive video tutorials that guide users through the installation process and demonstrate full use cases of the Nellie pipeline for different data types, including the use of custom plugins. These resources aim to enhance the usability and accessibility of Nellie for the broader research community.
3. **Expanded Analysis on Less Regular Structures:** We conducted an in-depth case study on the endoplasmic reticulum (ER) to showcase Nellie’s applicability to complex and dynamic organelle networks beyond mitochondria. This analysis includes detailed network characterization, comparative analysis across different cell types (primary human fibroblasts and U2OS cells), and validation using machine learning techniques.
4. **Addition of a Workflow Summary Figure:** A new figure (now Figure 1) has been included, providing a comprehensive flowchart of both the user workflow and Nellie’s internal pipeline. This visual summary enhances the clarity of the manuscript and aids readers in understanding the overall workflow and functionalities of Nellie.
5. **Detailed Training and Validation Processes:** We expanded our description of the training and validation procedures for the graph neural network (GNN) used in our analyses. We included training and validation loss curves (Supplementary Figure 8) to demonstrate the robustness of our models and to ensure that overfitting and data leakage have been adequately addressed.

6. **Optimized Runtime and Performance Comparisons:** We optimized Nellie's code for faster processing and conducted comprehensive runtime comparisons with existing tools (MitoGraph, Mitometer, and MitoTNT) for both segmentation and tracking tasks. The results, presented in Extended Data Figure 1 and Supplementary Note 1, demonstrate Nellie's efficiency and scalability across datasets of varying sizes.
7. **Enhanced Discussion on Parameter Selection:** We provided a more detailed explanation of Nellie's automated parameter selection processes, emphasizing how the pipeline adapts parameters based on image metadata to optimize performance across various types of microscopy images without the need for manual tuning.
8. **Extended Validation Across Various Organelles and Microscopy Techniques:** We expanded our validation to include multiple cell types (including but not limited to primary human fibroblasts and HEK293 cells), organelles (such as desmosomes, peroxisomes, actin, ER, Golgi apparatus, nucleoli, tight junctions, and tubulin structures), and microscopy techniques (including confocal, spinning disk, and widefield microscopy). This extensive validation, presented in Extended Data Figure 3 and Supplementary Data Figure 4, demonstrates Nellie's versatility and robustness across diverse imaging conditions.
9. **Improved Accessibility and Usability:** We have made significant efforts to make the manuscript more accessible to a broader audience, particularly cell biologists. Technical details and algorithmic discussions have been moved to the supplementary materials, and the main text now focuses on the practical applications and biological significance of Nellie. We also enhanced user guidance with detailed documentation, code comments, and tutorials, including over 3,000 lines of code documentation and step-by-step user guides.
10. **Minor Revisions and Clarifications:** We addressed all minor concerns raised by the reviewers, including adding equations and pseudocode for algorithms in the supplementary notes, updating the GitHub repository with detailed explanations of each file, and clarifying the outputs generated by Nellie.

We have provided detailed point-by-point responses to all the comments from the editor and reviewers, outlining how each concern was addressed in the revised manuscript, including line numbers for relevant sections in both the main manuscript and supplementary materials. We believe that these substantial revisions have significantly strengthened our manuscript and have addressed all the concerns raised.

We hope that the revised manuscript meets the high standards of Nature Methods and will be of interest to the journal's readership. We look forward to your response.

Sincerely,
Austin E. Y. T. Lefebvre

Table of Contents

Editor Revisions.....	4
Major Revisions.....	14
Reviewer 1.....	14
Reviewer 2.....	22
Reviewer 3.....	34
Minor Revisions.....	45
Reviewer 1.....	45
Reviewer 2.....	53
Reviewer 3.....	54

Editor Revisions

Concern: “During your revision, we ask that you compare to one custom, fine-tuned method (like U-nets) on at least two organelles. The point of this exercise is to show what, if any, performance gains a user will have with Nellie compared to a deep learning model that is well-suited to analyzing their particular data (vs a generalist algorithm like Nellie).”

Response:

We appreciate the editor’s suggestion to compare Nellie with a custom, fine-tuned deep learning method to evaluate potential performance gains. In response, we initially attempted to compare Nellie with several existing fine-tuned deep learning models, including ERnet (2D only), MitoSegNet (2D only), and the Allen Institute for Cell Science (AICS) segmenter (2D and 3D models). However, we encountered significant challenges:

- ERnet: Installation issues prevented successful deployment, and numerous unresolved problems reported on GitHub suggest this is a widespread concern.
- MitoSegNet: The model failed to detect any objects in our test images, resulting in segmentation outputs with no detected structures.
- AICS Segmenter: While it provides 2D/3D models for specific structures, it required extensive manual fine-tuning, making it impractical for our comparative purposes.

Due to these limitations, we proceeded to develop our own custom Swin UNETR segmentation models for a fair and rigorous comparison. We trained nine different models, each specialized for a specific organelle—mitochondria, desmosomes, actin, actomyosin, endoplasmic reticulum (ER), Golgi apparatus, nucleoli, tight junctions, and tubulin. Additionally, we created a combined model trained on data from all organelles.

These models were trained and validated with Nellie's segmentation outputs from the Allen Institute for Cell Science (AICS) label-free prediction dataset (<https://www.nature.com/articles/s41592-018-0111-2>). While the models performed well on the specific organelles and microscopy modalities they were trained on, their performance declined significantly when applied to different microscopes, organelles, or imaging modalities outside their training set. This observation underscores a known limitation of deep learning models—their generalizability is often constrained by the diversity of their training data.

In contrast, Nellie consistently demonstrated robust performance across a wide range of organelles and microscopy modalities without the need for fine-tuning. This highlights Nellie's strength as a non-deep-learning pipeline that can generalize effectively across diverse datasets.

We acknowledge that future deep learning foundation models may achieve greater generalization capabilities. However, such models currently do not exist, particularly for organelle microscopy where large-scale 3D manual annotations are scarce. We believe

Nellie could even serve as a valuable tool for providing initial 3D segmentations (as it did for our Swin UNETR models) that can aid in training future deep learning models.

Relevant Manuscript Sections:

Figures Added:

- Added a new figure (Supplementary Data Fig. 3) showing the training and validation loss curves for the models.
 - sL 733-740
- Included a new figure (Supplementary Data Fig. 4) displaying the predictions of the Swin UNETR models.
 - sL 741-754

Supplementary Material:

- Added a supplementary note (Supplementary Note 5) detailing the model architectures, training processes, and validation results for three of the models (mitochondria, desmosomes, and combined model).
 - sL 515-732
 - Uploaded the training and validation curve data for the three models.

Main Text Revisions:

- Expanded the Introduction and Discussion sections to address the advancements in deep learning methods for microscopy and to position Nellie within this context

- L 66-73, 164-167

Concern: *“We also ask that you make video tutorials for software installation and for a single-use case.”*

Response:

We appreciate the editor’s suggestion to enhance the usability of Nellie through video tutorials. In response, we have created a comprehensive series of video tutorials that cover both the installation process and a complete single-use case (for 3 different modalities) of the Nellie pipeline. These videos are designed to assist users in easily installing the software and utilizing its full functionality, including the use of custom plugins.

We believe these additions will significantly enhance the user experience and accessibility of Nellie, allowing users to fully leverage its capabilities with ease.

Modifications:

Installation Videos:

- Python Installation Video: A step-by-step guide for users who need to install Python prior to installing Nellie (<https://github.com/aelefebv/nellie?tab=readme-ov-file#installation--1-minute>).
- Method 1 - Napari-based Installation Video: Demonstrates how to install Nellie through the Napari graphical user interface (GUI) (<https://github.com/aelefebv/nellie?tab=readme-ov-file#option-1-via-napari-plugin-manager>).

- Method 2 - Pip-based Installation Video: Provides instructions for installing Nellie via the command line using pip

(<https://github.com/aelefebv/nellie?tab=readme-ov-file#option-2-via-pip>).

Usage Videos:

- Workflow Tutorials: Detailed walkthroughs of the full Nellie pipeline for different data types, including 2D + time, 3D + time, and datasets without time components. These videos cover file selection, slice selection, processing, and visualization of intermediate results, tracks, and outputs

(<https://github.com/aelefebv/nellie?tab=readme-ov-file#usage>).

- Nellie-Napari Plugins Tutorial: A guide on using the example Nellie-Napari plugin we developed, which outputs metrics for fission and fusion events

(<https://github.com/aelefebv/nellie-plugin-fission-fusion?tab=readme-ov-file#tutorial>).

Additional Resources:

- Repository Updates: The videos have been uploaded to the relevant README files in both the main Nellie repository and the newly created plugin repository.
- Plugin Development Template: We have added a new plugin development template repository to assist users in extending Nellie's functionality through custom plugins (<https://github.com/aelefebv/nellie-plugin-example>).
- Plugin System Documentation: A full tutorial on plugin creation, packaging, and distribution via PyPI has been included in the repository's README. This

includes a video tutorial demonstrating how to use the example plugin for fission and fusion event detection.

- Nellie Plugin System: Utilizes Python's entrypoint mechanism to dynamically discover and load plugins, enabling seamless integration within Nellie's ecosystem through the Napari interface.
 - L 104, 120-121

Concern: “We don’t require that you do as much for another organelle as you’ve done for mitos, but I think it would strengthen the paper if you chose one less regular structure (like ER or Golgi) and did a little more analysis to showcase general applicability. Otherwise, we think the additional compartments you describe will be great to add to the paper to emphasize versatility.”

Response:

We appreciate the editor’s suggestion to further demonstrate Nellie’s general applicability by analyzing a less regular organelle structure. In response, we have conducted an in-depth case study on the endoplasmic reticulum (ER) to showcase Nellie’s capabilities in analyzing complex organelle networks.

Using Nellie’s segmentation, skeletonization, and tracking functionalities, we performed a comprehensive characterization and comparison of ER networks across different cell types—primary human fibroblasts (hFB) and U2OS cells—and temporal frames. Our analysis focused on ER morphology, motility, and network topology.

This new case study highlights Nellie’s ability to:

- Construct Network Graphs: Generated from ER skeletons to enable detailed analysis of network properties, such as node degree and betweenness centrality.
- Perform Comparative Analyses: Assessed ER network topology and branch features between different cell types and across temporal sequences.

- Apply Tensor Decomposition Techniques: Employed to extract feature weights that distinguish between the two cell types while maintaining consistency within frames of the same cell type.
- Validate Findings with Machine Learning: Utilized random forest classifiers to validate the discriminative features, achieving high accuracy in classifying ER branches between the two cell types.

Our results demonstrate Nellie's strength in analyzing complex and dynamic organelle networks like the ER, providing meaningful insights into structural differences between cell types while ensuring consistency within temporal sequences. This analysis reinforces Nellie's general applicability and versatility across diverse organelle structures.

Additionally, we have included a figure showcasing Nellie's segmentation and tracking outputs across a variety of organelles and imaging modalities to further illustrate its generalizability.

We believe that this additional analysis significantly strengthens the manuscript by demonstrating Nellie's versatility and effectiveness in analyzing less regular organelle structures, thereby showcasing its general applicability.

Relevant Manuscript Sections:

Abstract and Main Text:

- Updated to include the new ER analysis as a demonstration of Nellie's broad range of applications, including a discussion, and materials and methods related to the case study.
 - L 33-35, 130-138, 495-587, 629-638, 658-661, 689-695, 707-711

Figures Added:

- Figure 6: Showcasing Nellie's detailed ER network characterization and comparative analysis across cell types and temporal frames.
 - L 523-546
- Extended Data Figure 3: Highlighting Nellie's versatility across multiple organelles, structures, and imaging modalities.
 - L 780-793

Supplementary Material:

- Included supplementary videos providing close-up visualizations of interpolated voxel tracks for all tracking examples presented in the Extended Data Figure 3.
- Included datasets for Figure 6's plots

Major Revisions

Reviewer 1

Concern: “Provide a figure for summary or flowchart of the Nellie pipeline.”

Response:

We thank the reviewer for this valuable suggestion. In response, we have created a new figure (now Figure 1) that provides a comprehensive flowchart summarizing both the user workflow and the internal pipeline of Nellie. This visual overview outlines the required and optional steps for users interacting with Nellie through the Napari GUI, as well as the stages of the internal pipeline once data is confirmed for processing. We believe that this addition will significantly enhance the clarity of the manuscript and aid readers in understanding the overall workflow and functionalities of Nellie.

Relevant Manuscript Sections:

Figure Changes:

- Replaced the original Figure 1 with the new flowchart figure (Figure 1) and updated the caption accordingly.
 - L 105-119
- Figure 1 Panels:
 - a–d: Depict the user workflow from file selection through feature visualization.

- e-m: Illustrate Nellie's internal pipeline, including segmentation, tracking, and feature extraction steps.

Text Revisions:

- Updated the Introduction section to incorporate an overview of the new flowchart
 - L 90-104, 120-121

Concern: “Please provide more details on the training and validation processes, including training and validation loss curves, to check for overfitting or data leakage.”

Response:

We thank the reviewer for highlighting the need for additional details regarding our training and validation processes. In response, we have expanded our description of the training and validation procedures for the graph neural network (GNN) used in our second biological application. We agree that providing the training and validation loss curves is essential for demonstrating the model’s performance and ensuring that overfitting and data leakage have been adequately addressed.

We utilized a 70/30 split for the training and validation datasets, implementing early stopping to prevent overfitting. Data leakage was avoided by strictly separating the training and validation datasets and performing independent feature scaling on each set. Training was halted when no improvement in validation loss was observed for 10 consecutive epochs, with a minimum improvement threshold of 0.001. The loss curves, now included as Supplementary Figure 8, show that the model trained for a total of 47 epochs, with the lowest validation loss achieved at epoch 37. The close alignment between the training and validation loss curves indicates that overfitting did not occur.

Relevant Manuscript Sections:

Supplementary Material:

- Added Supplementary Figure 8 displaying the training and validation loss curves for the GNN.
 - sL 1432-1437
- Expanded the supplementary note 14 on our GNN model to provide detailed information on the training process, model evaluation metrics, and measures taken to prevent overfitting and data leakage.
 - sL 1418-1431
- Uploaded training and validation loss curve raw data.

Concern: “During our testing, we found that processing larger images could take several hours. Add a computer runtime comparison of Nellie, MitoMeter, and MitoGraph for Extended Data Fig. 2 for 2D, 3D or time-lapse images at different pixel resolutions.”

Response:

We appreciate the reviewer bringing this issue to our attention. To address this concern, we have made several optimizations to our pipeline, including vectorization of slow portions of the code, made non-essential portions of Nellie disabled by default, and finally conducted a comprehensive runtime comparison between Nellie, MitoGraph, Mitometer, and MitoTNT for both segmentation and tracking tasks. These comparisons were performed on datasets of varying sizes to evaluate the scalability and efficiency of each tool. The tests were executed on both CPU and GPU platforms to provide a thorough assessment under different computational conditions.

Our results demonstrate that Nellie offers superior performance in segmentation speed compared to MitoGraph and Mitometer, especially when utilizing GPU acceleration. For tracking tasks, Nellie also outperforms MitoTNT and Mitometer, particularly with larger datasets. These findings highlight Nellie’s efficiency and scalability for processing large and complex imaging data.

Relevant Manuscript Sections:

Extended Data Figure:

- Added Extended Data Figure 1 presenting the runtime comparisons:

- L 755-763
- Figure 1a: Segmentation time comparisons for MitoGraph, Mitometer, and Nellie (CPU and GPU) across datasets of increasing size.
- Figure 1b: Tracking time comparisons for MitoTNT, Mitometer, and Nellie (CPU and GPU) across datasets of increasing size.
- Figure 1c–d: Visualizations of the datasets used for comparison, ranging from the smallest dataset (3.7 MB) to the largest dataset (1.87 GB).

Supplementary Material:

- Added Supplementary Note 1 detailing the runtime comparison methodology, including testing procedures, dataset preparation, and a comprehensive analysis of the performance of each tool.
 - sL 20-88

Concern: “In the text and supplementary notes, the authors have explained the methods for image preprocessing in detail, including multi-scale Gaussian filtering, Minotri thresholding, Frangi filter, LoG filter, hole filling, etc. However, the performance of various steps in Nellie, such as the Frangi filter and mocap marker generation, heavily depends on the choice of parameters. Although the authors have provided initial/default values for these parameters, further details on parameter tuning might be needed. Please discuss the parameter choice in detail. Test these methods on different types of images and add a discussion about finding parameters for different types of microscopy images. Adding these default parameters in the supplementary table may be helpful, too.”

Response:

We thank the reviewer for this insightful suggestion. We would like to reemphasize that Nellie is designed to function without the need for manual parameter tuning. All critical parameters, particularly those related to the Frangi filter and motion capture marker generation, are automatically determined based on the input image’s characteristics. This automated parameter selection allows Nellie to generalize across different datasets and imaging modalities without user intervention.

To address the reviewer’s concern, we have expanded the manuscript to provide a more detailed explanation of our automated parameter selection processes. Specifically, we have elaborated on how Nellie adapts parameters based on image metadata, such as voxel dimensions and intensity distributions, to optimize performance across various

types of microscopy images. We have also included discussions on the robustness of our approach when applied to different imaging conditions and organelles. We believe that these additions will clarify how Nellie manages parameter selection and will provide users with a better understanding of how the pipeline can be applied to various imaging modalities without the need for parameter adjustments.

Relevant Manuscript Sections:

Main Text Revisions:

- Expanded the Results section to provide a detailed explanation of the automated parameter selection mechanisms in Nellie, emphasizing how parameters are calculated based on image metadata.
 - L 20-22, 111, 147-153, 590, 599-601
- Provided examples and results of testing Nellie on various types of microscopy images in a new Extended Data Fig. 3, demonstrating the effectiveness of the automated parameter selection across different imaging conditions.
 - L 780-781, 164-167, 206-209

Supplementary Material:

- Expanded the supplementary notes to include detailed descriptions of the parameter selection algorithms for the Frangi filter, segmentation, motion capture marker generation, and tracking scheme.
 - sL 89-305, 320-390, 791-842, 867-911, 1069-1103

Reviewer 2

Concern: “The novelty of Nellie is somewhat limited by its dependence on traditional filtering techniques... Frangi filtering relies on predefined mathematical operations and eigenvalue analysis, with generally fixed parameters that do not automatically adapt to specific image contexts beyond scale variations... The need for parameter tuning means that different users might achieve different results based on their specific settings and adjustments... it may not perform well on non-tubular structures or highly complex and irregular shapes... Such structural diversity is not adequately considered in this work, yet it is crucial in biomedical research.”

Response:

We thank the reviewer for their thoughtful critique and for highlighting important considerations regarding the Frangi filter and its application within Nellie. We would like to address each point raised and provide clarifications to demonstrate how our work addresses these concerns.

We acknowledge that traditional Frangi filtering techniques have limitations, particularly regarding parameter adaptability beyond scale variations. However, Nellie introduces significant modifications to the traditional Frangi filter to allow for automatic adaptation to specific image contexts, including but not limited to scale variations. Our approach adjusts key parameters based on the inherent characteristics of the input image, such

as voxel dimensions and intensity distributions. This ensures that the filter adapts to various imaging conditions without manual intervention.

We designed Nellie to be fully automated, eliminating the need for manual parameter tuning. The pipeline calculates all necessary parameters internally, preventing discrepancies that might arise from different user settings. This design choice enhances user-friendliness and ensures consistent results across different users and datasets. Users interact with Nellie through a graphical user interface (GUI) that does not expose parameter settings, thus streamlining the process and making it accessible to non-specialists.

We recognize the importance of accommodating structural diversity in biomedical research. Nellie is explicitly engineered to handle a wide range of organelle shapes, including non-tubular and highly irregular structures. We have modified the traditional Frangi filter to generalize beyond tubular structures by adapting the filter's response to the local image context. This modification allows Nellie to enhance and segment various organelle morphologies effectively. To validate our approach, we conducted extensive quantitative testing using a large set of simulated datasets that encompass diverse structural morphologies—ranging from small and large round objects to small and large tubular structures—across various noise levels. The results demonstrate Nellie's robust performance in accurately segmenting and analyzing these diverse structures.

Relevant Manuscript Sections:

Main Text Revisions:

- Results Section: Expanded the explanation of our modifications to the Frangi filter, detailing how parameters are automatically adjusted based on image metadata to adapt to various structures beyond tubular forms.
 - L 147-157, 592-594
- Extended Data Figure 2: Illustrated the performance of Nellie on simulated datasets with various shapes and noise levels.
 - L 764-779
- Extended Data Figure 3: Showcased Nellie's segmentation outputs across a variety of organelles and microscopy methods, including non-tubular and highly irregular structures.
 - L 780-793

Supplementary Material:

- Supplementary Note 2, 3: Provided detailed descriptions of the automated parameter selection algorithms for the Frangi filter and other preprocessing steps.
 - sL 89-305, 320-390
- Supplementary Note 5: Provided comparisons to DL-based models to showcase generalizability of the modified Frangi filter with automated parameter tuning to a wide variety of intracellular structures and imaging modalities.
 - sL 515-619

Concern: “Compared to traditional segmentation based on filtering or thresholding, state-of-the-art models based on deep neural networks, such as U-net, Vision-Transformer, and Swin-Transformer, have been extensively applied to perform image segmentation from cells to organelles. These models not only generate robust segmentation results with high precision and accuracy but also produce multi-parametric measurements informing the structural and dynamic features of the objects with biological significance. The current work faces challenges in demonstrating its technical advancement and precision in results compared to these studies.”

Response:

We appreciate the reviewer’s perspective on the advancements in deep learning (DL) models for image segmentation and the importance of benchmarking Nellie against such methods. We have carefully considered this point and undertaken substantial efforts to address it.

Our extensive literature review indicates that while DL models like U-Net, Vision Transformers, and Swin Transformers have significantly advanced image segmentation, their application for generalized 3D organelle segmentation across various fluorescence microscopy modalities is currently non-existent. Most of the relevant existing 3D DL-based segmentation tools are specialized for cell segmentation, nuclear segmentation, or electron microscopy datasets.

To provide a fair comparison, we attempted to evaluate Nellie against existing fine-tuned deep learning models

- ERnet (2D Only): Installation issues prevented successful deployment, and numerous unresolved problems reported on GitHub suggest this is a widespread concern.
- MitoSegNet (2D Only): The model failed to detect any objects in our test images, resulting in segmentation outputs with no detected structures.
- AICS Segmenter (2D and 3D Models): While it provides models for specific structures, it required extensive manual fine-tuning and parameter adjustments, making it impractical for a generalized comparison.

Due to these limitations, we developed and pre-trained our own custom Swin UNETR models for a rigorous comparison:

Model Training:

- Trained nine models, each specialized for a specific organelle (mitochondria, desmosomes, actin, actomyosin, ER, Golgi apparatus, nucleoli, tight junctions, and tubulin).
- Created a combined model trained on data from all organelles.
- Utilized the Allen Institute for Cell Science (AICS) label-free prediction dataset for training and validation.

Performance Evaluation:

- The models performed well on the specific modality they were trained on.
- Performance declined significantly when applied to different microscopes, organelles, or imaging modalities outside their training set.
- This limitation highlights the challenge of generalizability in DL models due to the need for extensive and diverse training data.

In contrast, Nellie consistently demonstrated robust performance across a wide range of organelles and microscopy modalities without the need for manual annotations or fine-tuning. Nellie's non-DL approach allows it to generalize effectively across diverse datasets, filling a critical gap where DL models may struggle due to limitations in training data availability and generalization capabilities. Of note, we hope Nellie's outputs can be useful in creating generalized segmentation models, either directly, or following manual refinement of its outputs for use as ground truth training and validation datasets.

Relevant Manuscript Sections:

Supplementary Material:

- Supplementary Note 5: Detailed the architectures, training processes, and validation results for three of the models (mitochondria, desmosomes, and combined model).
 - sL 515-732
- Supplementary Data Figure 3: Showed the training and validation loss curves for the custom DL models.
 - sL 733-740

- Supplementary Data Figure 4: Displayed the predictions of the custom Swin UNETR models alongside Nellie's outputs for direct comparison.
 - sL 741-754
- Supplementary Data: Provided the training and validation data for all three models.

Main Text Revisions:

- Expanded to discuss advancements in DL methods and position Nellie within this context and Highlighted Nellie's superior generalizability and utility in contexts where DL models may face limitations.
 - L 66-73, 164-167

Concern: “Although the introduction of motion capture markers is technically sound and could be interesting for mitochondria studies, it is not clear how this can be interpreted within a biological framework. What is its biological significance? How is it correlated with the structure, state, and functions of mitochondria? How can researchers implement this in their studies?... how can this method analyse the non-tubular shapes of mitochondria?”

Response:

We thank the reviewer for acknowledging the technical soundness of our motion capture (mocap) markers and for raising important questions regarding their biological interpretation and applicability.

The mocap markers in Nellie are designed as computational tools to facilitate accurate motion interpolation during the tracking of organelles across time frames. They serve as linkage points that guide flow estimation algorithms, enhancing the precision of temporal analyses within the pipeline. We want to clarify that these mocap markers are not intended to have direct biological significance or to be interpreted within a biological framework. Their primary function is to improve the technical accuracy of tracking by providing reference points for motion estimation. As such, they are not used for interpretation of specific structural, state, or functional aspects of the organelles themselves.

Regarding the analysis of non-tubular shapes, Nellie's tracking method, aided by the mocap markers, is designed to handle a wide range of organelle morphologies, including non-tubular and highly irregular structures. The mocap markers enhance tracking performance by adapting to the local structural context of the organelles, regardless of their shape.

Relevant Manuscript Sections:

Main Text:

- Added clarification and additional details on how the mocap markers function within the tracking algorithm without attributing biological meaning to them, as well as its generalizability.
 - L 82-85, 195-209, 594-597

Figures Added:

- Extended Data Figure 3: Showcased Nellie's tracking across various organelles and microscopy methods, including non-tubular and complex structures, demonstrating the effectiveness of the mocap markers.
 - L 780-792, 164-167

Supplementary Materials:

- Added tracking visualization videos for all examples in Extended Data Figure 3.
- Expanded on Supplementary Note 7, describing the mocap marker generation pipeline

- sL 791-842
- Expanded on Supplementary Note 10, describing the features used specifically for linking mocap markers between frames.
 - sL 1069-1103
- Expanded on Supplementary Note 11, describing the usage of mocap markers to link voxels between frames, allowing for temporal continuity of semantic segmentation labels across frames.
 - sL 1104-1162

Concern: “The current work lacks versatility testing of its methods. First, only U2OS cells are used throughout the study. How about other non-flat cells, such as HEK293, neurons, or other cell models? Second, only one microscopy mode is used to acquire data. How about other microscopy techniques, particularly Structured Illumination Microscopy (SIM), which is widely used in the study of mitochondrial dynamics? Third, the method is tested only on tubular mitochondria. How about other shapes?”

Response:

We appreciate the reviewer’s insightful feedback and agree that demonstrating Nellie’s versatility across different cell types, microscopy techniques, and organelle shapes is crucial. To address these concerns, we have expanded our study to include a broader range of datasets, including our in-house rounded Jurkat cells and flat primary human fibroblast cells, as well as examples from the AICS dataset which includes human embryonic kidney cells (HEK293), human fibrosarcoma cells (HT-1080), and genome-edited (hiPSC) lines. Unfortunately, we did not have access to neuronal datasets during the revision.

We have also added a variety of microscopy techniques, including 2D and 3D confocal datasets, 2D spinning disk datasets, 2D widefield datasets, and additional 3D lightsheet datasets. Although SIM is indeed widely used for studying mitochondrial dynamics, we did not have access to SIM datasets for this study. However, the included microscopy methods cover a wide range of imaging conditions and resolutions. Finally, we have added numerous organelle types to our manuscript, including mitochondria,

desmosomes, peroxisomes, actin, actomyosin, ER, Golgi apparatus, nucleoli, tight junctions, and tubulin structures. The included organelles exhibit a variety of shapes, from tubular to spherical to highly irregular structures, demonstrating Nellie's capability to handle different morphologies.

Relevant Manuscript Sections:

Main Text:

- Expanded to emphasize Nellie's versatility and robustness in handling various cell models, microscopy modalities, and organelle shapes.
 - L 164-167, 205-209
- Figure 6: Showcased Nellie's segmentation and analysis on ER datasets
 - L 33-35, 495-587, 629-638
- Extended Data Figure 3: Showcased Nellie's segmentation and tracking outputs across a wide array of organelles, cell types, and microscopy methods.
 - L 780-793

Supplementary:

- Supplementary Data Figure 4: Additional segmentation examples from Nellie against the AICS dataset.
 - sL 741-754
- Provided videos of close-up visualizations of interpolated voxel tracks for all tracking examples in Extended Data Figure 3, highlighting Nellie's performance in dynamic contexts.

Reviewer 3

Concern: “According to the Methods section, all images were acquired using a light-sheet microscope. This imaging modality is well-known to produce a superior signal-to-noise ratio in fluorescence imaging, and therefore generate images from which subcellular structures can be readily segmented and analysed. This therefore raises the crucial question of whether ‘Nellie’ is also able to successfully work with regular confocal images, or images from wide-field microscopes, or whether its functionality is limited to images with high signal-to-noise. Given that probably less than 1% of the cell biology labs in the world have easy access to a light sheet microscope, it is unclear as to the wider applicability of Nellie for day-to-day use by cell biology researchers.”

Response:

We thank the reviewer for this important observation regarding the applicability of Nellie to images acquired using different microscopy modalities. We agree that demonstrating Nellie’s versatility across various imaging techniques is crucial for its broader adoption by the cell biology community.

To address this concern, we have expanded our study to include datasets acquired using a range of common microscopy techniques beyond light-sheet microscopy.

Specifically, we have tested Nellie on images obtained from:

- Point scanning confocal microscopy (both 2D and 3D datasets)

- Spinning disk microscopy
- Widefield microscopy
- Additional light-sheet microscopy datasets for organelle diversity

These additional datasets encompass images with varying signal-to-noise ratios and resolution levels, much more representative of those commonly encountered in day-to-day cell biology research.

Our results demonstrate that Nellie effectively processes and analyzes images from these modalities, maintaining robust performance even on data with lower signal-to-noise ratios typical of confocal and widefield microscopes.

Relevant Manuscript Sections:

Main Text:

- Expanded to emphasize Nellie's versatility and robustness in handling various cell models, microscopy modalities, and organelle shapes.
 - L 164-167, 205-209
- Extended Data Figure 3: Showcased Nellie's segmentation and tracking outputs across a wide array of organelles, cell types, and microscopy methods.
 - L 780-793

Supplementary:

- Supplementary Data Figure 4: Additional segmentation examples from Nellie against the AICS dataset.
 - sL 741-754
- Provided videos of close-up visualizations of interpolated voxel tracks for all tracking examples in Extended Data Figure 3, highlighting Nellie's performance in dynamic contexts.

Concern: “The vast majority of the data presented analyse mitochondrial dynamics.

This raises the question of the adaptability of ‘Nellie’ to analyse other complex membrane organelles. These might include the endoplasmic reticulum, dynamic endosomes/lysosomes, and membrane traffic events emanating from the Golgi apparatus. Figure 4 does show some data using a Golgi marker (a truncated variant of the glycosylation enzyme B4GALT1), however this organelle appears highly unusual as a series of punctate structure, not the characteristic juxta-nuclear membranes. As such, the sparse examples from other important membrane compartments in the cell suggest that ‘Nellie’ is probably only effective at analysis of mitochondrial dynamics. The endomembrane system of cells has many highly dynamic organelles, and further examples need to be provided to address the wider applicability of ‘Nellie’.”

Response:

We appreciate the reviewer’s insightful feedback regarding the applicability of Nellie to organelles beyond mitochondria. We agree that demonstrating Nellie’s versatility in analyzing various complex membrane organelles is essential to establish its broader utility.

In response to this concern, we have conducted an in-depth case study focusing on the endoplasmic reticulum (ER), a highly dynamic and structurally complex organelle within the endomembrane system. Using Nellie’s segmentation, skeletonization, and tracking functionalities, we performed comprehensive analyses of ER morphology, motility, and

network topology in different cell types—specifically primary human fibroblasts (hFB) and U2OS cells.

Our analysis demonstrates Nellie’s capability to:

- Construct Network Graphs: Generated from ER skeletons, enabling detailed examination of network properties such as node degree and betweenness centrality.
- Perform Comparative Analyses: Assessed differences in ER network topology and branch features between cell types and over time.
- Extract Discriminative Features: Utilized tensor decomposition to identify feature weights that distinguish between cell types while maintaining temporal consistency.
- Validate Findings with Machine Learning: Employed random forest classifiers to confirm the discriminative power of the extracted features, achieving high accuracy in classifying ER branches between cell types.

Beyond the ER, we have expanded our dataset to include various other organelles and structures, such as mitochondria, desmosomes, peroxisomes, actin, actomyosin, ER, Golgi (these with more typical juxta-nuclear membranes), nucleoli, tight junctions, and tubulin structures. These organelles exhibit diverse morphologies—from tubular to spherical to highly irregular shapes—demonstrating Nellie’s ability to handle a wide range of structural complexities.

Relevant Manuscript Sections:

Main Text:

- Figure 6: Showcasing Nellie's detailed ER network characterization and comparative analysis across cell types and temporal frames.
 - L 523-546
- Updated to include the new ER analysis as a demonstration of Nellie's broad range of applications, including a discussion, and materials and methods related to the case study.
 - L 33-35, 130-138, 495-587, 629-638, 658-661, 689-695, 707-711
- Expanded to emphasize Nellie's versatility and robustness in handling various cell models, microscopy modalities, and organelle shapes.
 - L 164-167, 205-209
- Extended Data Figure 3: Showcased Nellie's segmentation and tracking outputs across a wide array of organelles, cell types, and microscopy methods.
 - L 780-793

Supplementary:

- Supplementary Data Figure 4: Additional segmentation examples from Nellie against the AICS dataset.
 - sL 741-754
- Provided videos of close-up visualizations of interpolated voxel tracks for all tracking examples in Extended Data Figure 3, highlighting Nellie's performance in dynamic contexts.

- Included datasets for Figure 6's plots

Concern: “It is unclear who this manuscript is targeted at in its current form. The vast majority of the text is written in a way that is highly specialised and directed towards experts who develop image analysis tools. However, the end-point users will inevitably be cell biologists, and for that community, the manuscript is not particularly ‘digestible’. In its current form, the manuscript is far more suited to a specialised informatics journal.”

Response:

We appreciate the reviewer’s thoughtful observation regarding the accessibility of our manuscript to the broader cell biology community. Our intention is to present Nellie as a valuable tool for cell biologists while providing sufficient methodological detail for those interested in the technical aspects.

To address this concern, we have made several revisions to make the manuscript more digestible for cell biologists:

Simplified Presentation:

- Shifted detailed technical descriptions and algorithmic discussions to the supplementary materials, allowing the main text to focus on the practical application and biological relevance of Nellie.
- Ensured that figures and explanations in the main text are presented in a clear and accessible manner, highlighting the biological insights that Nellie enables.

Flowchart for Workflow Overview:

- Included a flowchart that outlines both the user workflow and the internal pipeline of Nellie. This visual aid helps users understand the process without needing to read into technical complexities, while also emphasizing the minimal input required by the user.

Emphasis on Biological Applications:

- Expanded sections that demonstrate Nellie's application to biologically relevant cases, emphasizing the insights gained and how they can be replicated or extended by other researchers.

User-Friendly Implementation. We have designed and updated Nellie with ease of use in mind for cell biologists without coding experience:

- More intuitive GUI:
 - Nellie operates through a user-friendly, point-and-click graphical user interface (GUI) with straightforward options for running analyses.
 - Users can process their data by simply selecting files or folders and initiating the pipeline with minimal steps.
- Accessible Outputs:
 - Quantitative results are provided in common formats (e.g., .csv files) that can be easily opened and analyzed using standard software like Microsoft Excel.

- Intermediate images are saved in widely compatible formats (e.g., .tif) that can be viewed in common image analysis tools such as ImageJ or Napari, and are openable via Nellie's visualization tab via point-and-click usage.

Comprehensive Documentation and Tutorials:

- Added extensive documentation, including over 3,000 lines of comments, and user guides, to assist users in installation and operation.
- Created video tutorials that guide users through installation and typical use cases, enhancing accessibility for non-specialists.

We believe that these revisions significantly improve the manuscript's accessibility to cell biologists, making it more suitable for publication in Nature Methods. Our goal is to empower researchers to utilize Nellie effectively in their studies, gaining valuable biological insights without requiring expertise in image analysis tool development.

Relevant Manuscript Sections:

Main Text Revisions:

- Refined the language to reduce technical jargon and focus on the practical benefits of Nellie for cell biology research.
- Reorganized content to prioritize biological applications and results over technical details.
 - L 141-330

Supplementary Material:

- Moved in-depth technical explanations, algorithmic details, and computational considerations to the supplementary notes.
 - sL 89-390, 755-911, 1069-1260

Enhanced User Guidance:

- Clarified within the repo which files and outputs are intended for the user's direct interaction versus those necessary for the pipeline's internal functioning.
- Organized output directories to separate user-facing results from pipeline necessities (e.g., placing internal files in a subfolder named `nellie_necessities`).

Additional Resources:

- Updated the README files and online documentation to provide clearer instructions for installation and use.

Minor Revisions

Reviewer 1

Concern: “Adding equations when explaining the methods for filtering, segmentation, feature extraction, and graph-based analysis would be helpful”

Response:

We thank the reviewer for this valuable suggestion. In response, we have included detailed equations and pseudocode for the algorithms used in Nellie’s pipeline within the supplementary notes. These additions provide a comprehensive mathematical explanation of the methods for filtering, segmentation, feature extraction, mocap marker generation, mocap marker feature matrix construction, mocap marker linkage, feature extraction, and graph-based analysis, which we believe enhances the clarity and reproducibility of our work.

Relevant Manuscript Sections:

Supplementary Material:

- Filtering Techniques: Detailed the mathematical formulations of the multi-scale Gaussian filtering, modified Frangi filter and other pre-processing related algorithms.
- sL 89-305

- Segmentation Algorithms: Provided equations describing the Minotri thresholding.
 - sL 320-390
- Mocap Marker Generation: Provided equations describing the generation of mocap markers, construction of feature matrices of mocap markers, and linkage of mocap markers.
 - sL 791-842, 867-911, 1069-1103
- Temporal Continuity of label voxels across frames: Provided equations describing the linkage of voxels between adjacent temporal frames.
 - sL 1104-1162
- Feature Extraction Methods: Included mathematical descriptions of the hierarchical feature extraction techniques.
 - sL 1163-1260
- Graph-Based Analyses: Presented the algorithms and equations used for constructing and analyzing organelle network graphs, and its downstream graph neural network.
 - sL 1281-1431

Concern: “Does the research team plan to develop plugins or APIs to integrate Nellie with tools like ImageJ or CellProfiler?”

Response:

We appreciate the reviewer’s interest in integrating Nellie with other popular image analysis tools. Currently, we do not have plans to develop plugins or APIs specifically for ImageJ or CellProfiler. However, we have designed Nellie with modularity and extensibility in mind. We have included substantial documentation within our codebase and have provided easy-to-reference wrappers that allow users and developers to run individual sections of the pipeline. This modular design facilitates potential integration with other platforms. Developers proficient in Java or familiar with CellProfiler plugin development should find it straightforward to adapt Nellie’s functionalities for integration with these tools.

Relevant Manuscript Sections:

Developer Documentation:

- Added substantial documentation to our codebase (>3000 lines)

Concern: “Including some files in the GitHub repository that explains the purpose of each file in the folder would be helpful to understand and finding the codes.”

Response:

We thank the reviewer for this helpful suggestion. In response, we have updated the README file in our GitHub repository to include a detailed description of the codebase. This section explains the purpose of each file and folder within the repository, making it easier for users to navigate and understand the code. Additionally, we have added over 3,000 lines of documentation within the code, including tutorials and examples, to further assist users and developers in utilizing and extending Nellie.

Relevant Manuscript Sections:

GitHub Repository Updates:

- README File:
 - Added a Code Contents section detailing the purpose of each file and folder.
 - Organized the description under headings for the Nellie Pipeline and Nellie Napari Plugin.

- Inline Documentation:
 - Enhanced code comments and docstrings throughout the codebase.
 - Included usage examples and explanations of key functions.

- Tutorials:
 - Added step-by-step tutorials guiding users through installation, running analyses, and interpreting results.
 - Included example datasets for practice.

Concern: “Could Nellie provide the fission and fusion frequency or rate... as one of the outputs in Supplementary Table 1?”

Response:

We appreciate the reviewer’s interest in including fission and fusion frequency metrics in Nellie’s outputs. While we recognize the importance of these metrics in mitochondrial studies, we have chosen not to integrate them directly into the main codebase, as calculations of fission and fusion rates can be mitochondria-specific and may not generalize well to other organelles. However, to address this request and showcase Nellie’s extensibility, we have developed an example Nellie-Napari plugin (<https://github.com/aelefebv/nellie-plugin-fission-fusion>) that calculates label count differences between frames—a proxy for fission and fusion events, as described in the supplementary note on tracking comparisons.

We have also created a plugin development template (<https://github.com/aelefebv/nellie-plugin-example>) to assist users in creating their own custom plugins for Nellie. Additionally, we have produced a video tutorial demonstrating how to use the fission and fusion plugin. This approach allows users to tailor Nellie’s functionalities to their specific research needs without overcomplicating the core pipeline for all users.

Relevant Manuscript Sections:

Plugin Development Template:

- Provided a link to the GitHub repository containing the plugin template: Nellie Plugin Example.
 - L 834-841

Main Text Revisions:

- Mentioned the availability of the plugin system and how users can extend Nellie's functionality for specific applications.
 - L 29, 104, 120-121

Concern: “Please check Extended Data Fig. 2g for any potential mislabels on the pixel size of representative images of generated objects at high and low pixel resolutions.”

Response:

We thank the reviewer for pointing out this potential issue. We have carefully re-examined Extended Data Fig. 2g and confirm that the labeling is correct. In the context of our analysis, smaller pixel sizes correspond to larger objects when the physical size of the objects remains constant. This is because a smaller pixel size results in a higher resolution image, capturing more detail and thus representing the object with more pixels, making it appear larger in the pixel-based image. Conversely, a larger pixel size results in fewer pixels representing the object, making it appear smaller in the image.

Relevant Manuscript Sections:

Main Text:

- Extended Data Fig. 2g
 - L 764-779

Reviewer 2

No minor concerns listed.

Reviewer 3

Concern: “Supplementary Figure 1 seems to show the software interface. This is a little difficult to appreciate, and yet is a critically important feature of the work. More emphasis needs to be given to this.”

Response:

We appreciate the reviewer’s observation regarding the importance of the software interface and the need for clearer representation. In response, we have created a new figure (now Figure 1) that provides a comprehensive flowchart summarizing both the user workflow and the internal pipeline of Nellie. This visual overview highlights the critical features of the software interface and outlines the required and optional steps for users interacting with Nellie through the Napari GUI. We believe that this addition enhances the clarity of the manuscript and underscores the practical usability of Nellie for the research community.

Relevant Manuscript Sections:

Figure Changes:

- Replaced the original Figure 1 with the new flowchart figure (Figure 1) and updated the caption accordingly.
 - L 105-119
- Figure 1 Panels:

- a–d: Depict the user workflow from file selection through feature visualization.
- e–m: Illustrate Nellie’s internal pipeline, including segmentation, tracking, and feature extraction steps.

Text Revisions:

- Updated the Introduction section to incorporate an overview of the new flowchart
 - L 90-104, 120-121

Concern: “It is unclear what the output files are that are generated by ‘Nellie’. Are these all tabulated data and can they then be interrogated by downstream image data analytics software. Is specialised software needed to work with .pkl and .npy file formats?”

Response:

We appreciate the reviewer bringing this to our attention. Nellie generates output files in widely accessible formats to facilitate ease of use and compatibility with common data analysis tools:

Quantitative results are provided as comma-delimited text in .csv files, which can be opened and analyzed using standard software such as Microsoft Excel or any statistical analysis software. Intermediate images are saved as .tif files, compatible with popular image analysis tools like ImageJ or Napari. Users can view these images directly through Nellie’s visualization tab via a point-and-click interface. Internal pipeline files such as .pkl and .npy are used internally by Nellie for processing and are not intended for user interaction. These, and the intermediate images are now stored in a separate subdirectory named nellie_necessities to avoid confusion.

We hope that these clarifications will help users understand and effectively utilize the outputs generated by Nellie without the need for specialized software.

Relevant Manuscript Sections:

GitHub Repository Updates:

- Clarified within the repo which files and outputs are intended for the user's direct interaction versus those necessary for the pipeline's internal functioning.
- Organized output directories to separate user-facing results from pipeline necessities (e.g., placing internal files in a subfolder named `nellie_necessities`).

Concern: “What level of coding skills are needed for the installation and use of ‘Nellie’?”

Relating to my point above; the manuscript is highly technical with respect to image analysis methodology, but at the end of the day it will be cell biologists who will use the tool.”

Response:

We appreciate the reviewer’s concern regarding the accessibility of Nellie to users without coding expertise. Nellie is designed to be user-friendly and accessible to cell biologists with minimal technical background. Installation and use require no programming skills.

Installation can be performed through straightforward methods, including a point-and-click installation via the Napari plugin manager or using simple commands in the command line interface. We have created video tutorials that guide users step-by-step through the installation process.

Nellie operates through an intuitive graphical user interface (GUI) with clear instructions and minimal steps required to run analyses. Users can perform analyses by selecting files or folders and initiating the pipeline with simple button clicks.

We’ve also provided comprehensive documentation, including user guides and extensive code comments. Additionally, we’ve created a comprehensive series of video tutorials that cover both the installation process and a complete single-use case (for 3

different modalities) of the Nellie pipeline. These videos are designed to assist users in easily installing the software and utilizing its full functionality, including the use of custom plugins.

We believe these efforts make Nellie highly accessible to cell biologists and other researchers without requiring coding skills.

Relevant Manuscript Sections:

Main Text Revisions:

- Highlighted the user-friendly aspects of Nellie in the Introduction and Discussion sections.
 - L 21, 28-29, 78-80, 90-121, 602-604
- Replaced the original Figure 1 with the new flowchart figure (Figure 1) and updated the caption and main text accordingly.
 - L 90-121
- Figure 1 Panels:
 - a–d: Depict the user workflow from file selection through feature visualization.
 - e–m: Illustrate Nellie’s internal pipeline, including segmentation, tracking, and feature extraction steps.

Additional Resources:

- Enhanced the README files and online documentation to provide clearer installation and usage instructions.

Installation Videos:

- Python Installation Video: A step-by-step guide for users who need to install Python prior to installing Nellie (<https://github.com/aelefebv/nellie?tab=readme-ov-file#installation--1-minute>).
- Method 1 - Napari-based Installation Video: Demonstrates how to install Nellie through the Napari graphical user interface (GUI) (<https://github.com/aelefebv/nellie?tab=readme-ov-file#option-1-via-napari-plugin-manager>).
- Method 2 - Pip-based Installation Video: Provides instructions for installing Nellie via the command line using pip (<https://github.com/aelefebv/nellie?tab=readme-ov-file#option-2-via-pip>).

Usage Videos:

- Workflow Tutorials: Detailed walkthroughs of the full Nellie pipeline for different data types, including 2D + time, 3D + time, and datasets without time components. These videos cover file selection, slice selection, processing, and visualization of intermediate results, tracks, and outputs (<https://github.com/aelefebv/nellie?tab=readme-ov-file#usage>).
- Nellie-Napari Plugins Tutorial: A guide on using the example Nellie-Napari plugin we developed, which outputs metrics for fission and fusion events

(<https://github.com/aelefebv/nellie-plugin-fission-fusion?tab=readme-ov-file#tutorial>).

Additional Resources:

- Repository Updates: The videos have been uploaded to the relevant README files in both the main Nellie repository and the newly created plugin repository.
- Plugin Development Template: We have added a new plugin development template repository to assist users in extending Nellie's functionality through custom plugins (<https://github.com/aelefebv/nellie-plugin-example>).
- Plugin System Documentation: A full tutorial on plugin creation, packaging, and distribution via PyPI has been included in the repository's README. This includes a video tutorial demonstrating how to use the example plugin for fission and fusion event detection.
- Nellie Plugin System: Utilizes Python's entrypoint mechanism to dynamically discover and load plugins, enabling seamless integration within Nellie's ecosystem through the Napari interface.
 - L 104, 120-121

Concern: “What computational power is needed to run ‘Nellie’, and importantly what is the throughput? Similarly, what do typical output data sizes look like?”

Response:

Thank you for raising these important practical considerations. Nellie is designed to be adaptable to various computational environments. Nellie can run on standard desktop computers equipped with a CPU. For enhanced performance, particularly with large datasets, Nellie can utilize NVIDIA GPUs if available. Software dependencies are detailed in the README file, with installation instructions and videos provided.

In terms of throughput, we have conducted runtime comparisons, which are included in the manuscript (see Extended Data Figure 1 and Supplementary Note 1). Nellie performs efficiently across datasets of varying sizes, with GPU acceleration significantly improving processing times. In terms of output data sizes, the csv outputs heavily depend on how much of the data is taken up by actual organelle voxels, but even then are usually much smaller than the input dataset. However, for intermediate images and necessary files for Nellie to run processing or analysis, for a 16-bit dataset, the output-to-input data size ratio is approximately 15x. To alleviate this, users have the option to toggle on a setting within the GUI to automatically delete intermediate files after processing, retaining only the essential .csv files containing extracted features.

Relevant Manuscript Sections:

Manuscript text:

- Provided a comprehensive runtime comparison in Supplementary Note 1 and Extended Data Figure 1 including testing procedures.
 - L 90-92, 754-763
 - sL 20-88

GitHub Repository Updates:

- Updated the README file to include the computational requirements, performance considerations, and data size management options.

GUI Updates:

- Provide an optional checkbox to automatically remove intermediate files during processing.

Concern: “Can the user choose specific features to extract, or is it default that all feature information is extracted?”

Response:

We appreciate this insightful question. To enhance flexibility and efficiency, we have updated Nellie to allow users to select specific features for extraction. Notably, extraction of node features—which can be time-consuming and is usually not necessary for all analyses—is now opt-in. Users can enable or disable this feature in the Settings tab of the GUI. Voxel relabeling is also made optional, providing users control over processing time and outputs. We have made it so that users can export individual metrics through the Analyze tab within Nellie. This feature allows users to focus on specific data relevant to their research questions.

Relevant Manuscript Sections:

Methods Section:

- Updated to describe the new options for feature selection and how users can customize their analyses.

User Interface Updates:

- Revised the GUI to include settings where users can opt-in or opt-out of certain feature extractions.
- Created a feature export button in the analysis tab to create a new csv with only the feature of interest.

Documentation:

- Updated the README file and user guides to reflect these changes and provide instructions on selecting features.

Concern: “Can ‘Nellie’ be used to segment and quantify organelles in fixed cells, or is it only compatible with live cell / time-lapse data?”

Response:

Yes, Nellie can be used to segment and quantify organelles in fixed cells. It is compatible with datasets that have no temporal dimension, allowing for morphology-only analysis. Nellie can process single-frame datasets (both spatially 2D and spatially 3D) to extract morphological features of organelles in fixed cells. We have updated the manuscript to emphasize Nellie’s applicability to both live-cell and fixed-cell imaging data. We’ve also added an example use-case video of this scenario (3D dataset without time dimension) in the repository to guide users.

Relevant Manuscript Sections:

Main Text Revisions:

- Included the following statement: “Notably, Nellie allows the user to run single frame datasets (without a temporal dimension) for morphology-only analysis, or multi frame datasets for additional motility quantification, and works for both 2D (single-plane) datasets and 3D (volumetric) datasets.”
 - L 101-104
- We have also provided examples of 3D datasets without a temporal dimension segmented with Nellie within Extended Data Figure 3.

GitHub Repository:

- We have also provided an example usage video of running and analyzing a dataset without a temporal dimension.

Concern: “In Supplementary Table 1, in the Branches and Organelle features, the feature “area_raw” is used. For this feature, for 2D images, the unit is μm^2 and for 3D images it is μm^3 . How is μm^3 a measure of area, is this not a volumetric measurement?”

Response:

Thank you for highlighting this point. You are correct that in 3D images, the measurement represents volume. The feature “area_raw” in 2D images represents the area in square micrometers (μm^2). In 3D images, “area_raw” represents the volume in cubic micrometers (μm^3), effectively measuring the number of voxels scaled by the image resolution. We retained the name “area_raw” for consistency within the codebase, although the calculation differs between 2D and 3D datasets. However, we’ve updated Supplementary Table 1 to clarify this distinction.

Relevant Manuscript Sections:

Supplementary Table:

- Added a note for each area measurement specifying “The number of voxels in the <hierarchy level>, scaled by the image resolution (2D: μm^2 3D: μm^3). In 3D this is the volume.”
 - sL 1261-1273

Concern: “Why is simulated data used in Supplementary Notes 2 and 7? Please see my comment earlier with respect to more examples of organelle dynamics within the endomembrane system.”

Response:

We understand the importance of using real datasets for demonstrating practical applicability. However, simulated data plays a crucial role in our study. Simulated data allows us to create datasets with known properties and ground truth labels, which are essential for objectively evaluating and benchmarking the performance of Nellie and other software tools. Manual segmentation of 3D microscopy data is challenging, as no reliable tool exists, and the process is subject to user bias. Simulations provide a controlled environment for fair assessments.

In response to your earlier comment and to enhance the manuscript, we have included additional real dataset examples of organelle dynamics within the endomembrane system and other structures. These examples demonstrate Nellie’s applicability to various organelles and validate its performance in practical scenarios.

Relevant Manuscript Sections:

Abstract and Main Text:

- Updated to include the new ER analysis as a demonstration of Nellie’s broad range of applications, including a discussion, and materials and methods related to the case study.

- L 33-35, 130-138, 495-587, 629-638, 658-661, 689-695, 707-711

Figures Added:

- Figure 6: Showcasing Nellie's detailed ER network characterization and comparative analysis across cell types and temporal frames.
 - L 523-546
- Extended Data Figure 3: Highlighting Nellie's versatility across multiple organelles, structures, and imaging modalities.
 - L 780-793

Supplementary Material:

- Included supplementary videos providing close-up visualizations of interpolated voxel tracks for all tracking examples presented in the Extended Data Figure 3.
- Included datasets for Figure 6's plots
- Explanations in Supplementary Notes 4 and 9 regarding the use of simulated data.
 - sL 450-453, 970-976

Concern: “The Methods section mentions an ECFP-H2B construct, but I do not see any data in the manuscript using this construct.”

Response:

Thank you for pointing out this oversight. While the ECFP-H2B construct was included in the cell line for potential future applications, it was not utilized in the analyses presented in this manuscript, only the other two fluorescent markers were used in our study, namely, mEGFP targeted to the mitochondrial matrix with a human COX8 presequence and the first 82 residues of B4GALT1 tagged with mScarlet.

Relevant Manuscript Sections:

Updated the Methods section to clarify this point:

- “...ECFP-tagged H2B (not used in this paper)...”
 - L 682

Concern: “The Methods sections mentions ‘drug injections’ – what cells were injected with drugs?”

Response:

We appreciate the opportunity to clarify this point. We meant the term “drug injections” as referring to the addition of drugs to the cell culture medium, not to injections into the cells themselves:

Relevant Manuscript Sections:

Updated the Methods section to provide a clearer description:

- “For calcium-ionophore treatment experiments, Ionomycin (4 μ M, Thermo #I24222) in DMSO was manually injected into the media of the cell imaging dish.”
- L 713-715

Editor:**Remarks to the author:**

With regards to the remaining concerns, please update your discussion of limitations of the methods, especially if it underperforms on certain organelles, like the ER, relative to models dedicated to handling a specific organelle. In addition, we ask that you please briefly discuss existing methods that could not be compared (like ERNet, MitoSegNet, and AICS) and why in the discussion and not only the rebuttal. Please provide a point-by-point rebuttal summarizing the changes upon resubmission.

Response:

Dear Rita,

We have expanded the discussion section to include a more detailed examination of the limitations of our methods, especially regarding underperformance on certain organelles like the endoplasmic reticulum (ER) relative to models dedicated to handling specific organelles. We acknowledge that while Nellie is versatile and generalizable, specialized tools may offer superior performance for their target organelles. Specifically, we have added the following to the discussion:

“While Nellie performs well across diverse datasets without parameter tuning, it may underperform compared to specialized software when analyzing specific organelles. For example, dedicated tools like ERNet and AnalyzER for the ER discriminate between tubular and sheet-like structures^{29,33}. Similarly, MitoSegNet specializes in mitochondrial analysis in 2D images, and the Allen Institute’s segmentation pipeline offers specialized tools for each organelle^{28,32}. We attempted to compare Nellie with these models but faced challenges: ERNet had installation issues, and is limited to 2D images; MitoSegNet failed to detect structures in our images and also operates only on 2D images; the AICS Segmenter, while providing 2D and 3D models, requires extensive manual fine-tuning, making it impractical for comparison.

While such tools may offer superior performance for their target organelles, Nellie’s strength lies in its generalizability across organelles, without any parameter tuning or annotated datasets. However, Nellie does not perform optimally for organelles with large minor axes, such as nuclei, as it tends to segment the edges rather than the entire structure. This limitation suggests that for certain applications, integration with specialized methods may be necessary. As deep learning advances, generalized foundation models trained on extensive ground truth data may eventually surpass Nellie, but until then, Nellie remains a valuable tool for researchers needing a flexible, organelle-agnostic pipeline for spatial and temporal analysis.”

In addition to acknowledging these limitations, we have briefly discussed existing methods that we could not compare directly—such as ERNet, MitoSegNet, and the AICS Segmenter—and the reasons why.

We believe these revisions address your concerns and enhance the manuscript by offering a balanced perspective on Nellie's capabilities and limitations within the current landscape of organelle analysis tools.

Thank you for your time and effort throughout the revision process. Your guidance has significantly improved the manuscript, making it more accessible and informative for our readers.

Sincerely,
Austin Lefebvre

Reviewer #1:**Remarks to the Author:**

I appreciate the authors' efforts in answering the questions and concerns. All the issues were solved. The manuscript is largely improved.

As a method paper, the assisted documentation, videos, and annotations are now more accessible to readers. Runtime and performance comparisons were added. A case study on the less regular structure (ER) was added. Comparisons with fine-tuned deep learning models (Swin UNETR) were added.

Remarks on code availability:

The assisted documentation, videos, and annotations of codes are more accessible to the readers now.

Response:

Dear Reviewer,

Thank you for your thoughtful and constructive feedback throughout the revision process. We are delighted to hear that you find the manuscript substantially improved. Your suggestions have been invaluable in enhancing the accessibility, clarity, and overall quality of our work. We appreciate your time and effort in helping us refine our paper.

Sincerely,
Austin Lefebvre

Reviewer #2:**Remarks to the Author:**

Fig1. Need to improve the presentation, some subfigs are hard to see.

Fig. 6. The authors claim to analyse 3D volumetric images. But no volumetric structure of ER is presented. The skeletons are just 2D. Clearly the ER segmentation of tubular network in the perispherical region is overestimated, which should be largely ER sheet. This is likely due to the poor segmentation in the dense tubular region and the sheet. Existing methods including Pain et al., 2019 and Lu et al., 2023, have already achieved much better classification of ER tubules and sheets. Therefore the authors need to discuss their method's limitation, such as overestimation of tubular network, at this point to avoid misleading.

There is still a lack of biological significance discussion. Why are the features or dynamics analysed by this method important?

Response:

Dear Reviewer,

Thank you for your valuable feedback and constructive suggestions throughout the revision process. We have addressed your comments to improve the clarity and quality of our manuscript.

Regarding Figure 1, we have replaced the screenshots of the GUI with higher-resolution versions to enhance visibility and ensure that all subfigures are clear and easy to interpret.

In response to your comments on Figure 6, we would like to clarify that all images shown are of 3D ER structures, including the skeletons. We have included this clarification in the figure caption. We acknowledge that Nellie does not perform classification between ER tubules and sheets, which may lead to an overestimation of the tubular network in regions where sheets are prevalent. Recognizing this limitation, we have included a discussion in the manuscript to address the potential for overestimation and to avoid any misleading interpretations.

Regarding the biological significance of the features and dynamics analyzed by our method, we appreciate your insight. While our paper primarily focuses on the development and validation of the Nellie pipeline as a methodological tool, we understand the importance of contextualizing its applications within biological research. To this end, our first paragraph in the introduction section highlights the relevance of studying organellar morphology and motility. However, we have intentionally kept this section concise to maintain the focus on method development and to avoid overlapping with the extensive discussions present in specialized literature on specific organelles and use cases.

Once again, thank you for your helpful suggestions. We believe that these revisions have made the manuscript more accessible and improved its overall quality.

Sincerely,
Austin Lefebvre

Reviewer #3:**Remarks to the Author:**

Many thanks to the authors who have taken my comments and suggestions seriously. The rebuttal and associated revisions are comprehensive. The revised manuscript is significantly improved from the original version. Of particular note is demonstration that 'Nellie' can indeed be applied to analyze multiple organelles beyond mitochondria; this is a key addition. The additional documentation for use of Nellie is also an extremely valuable addition to the revised manuscript, and will be appreciated by users.

My only remaining comment - which does need to be addressed given the nature of the work - is that all the figures showing images need to have appropriate scale bars added. On review of the revised manuscript (containing additional, and very welcome examples), it is clear that different 'zooms' are used across the various data presented, and as such a scale bar is vital as a reference for readers of the work to understand the images that they are looking at. This comment applies to the figures in the main body text, and also in the supporting data.

Professor Jeremy C Simpson

Response:

Dear Professor Simpson,

Thank you for your thorough review and for your kind remarks regarding the improvements in our revised manuscript. We are pleased to hear that the additional demonstrations of Nellie's applicability to multiple organelles and the enhanced documentation have been well received.

We appreciate your insightful comment about the necessity of including appropriate scale bars in all figures containing images. You are absolutely correct that scale bars are vital for readers to accurately interpret the images, especially given the varying magnifications used throughout our data presentations. In response to your suggestion, we have added scale bars to all relevant figures in the main text, specifically Figures 1 through 6, and have updated the corresponding figure legends to include detailed scale information. Additionally, we have incorporated scale bars into the supplementary figures (Supplementary Figures 1, 2, 4, 5, and 6) and extended data figures (Extended Data Figures 1, 2, 4, 5, and 6), along with updates to their legends.

We believe these additions will enhance the clarity and usefulness of the figures for our readers. Once again, thank you for your valuable feedback and constructive suggestions throughout the revision process. Your input has significantly contributed to making the manuscript more accessible and has overall improved the quality of our work.

Sincerely,
Austin Lefebvre